# Adaptive Reduced Rank Regression

**Qiong Wu**[*]
William & Mary

**Felix M. F. Wong**
Independent Researcher[†]

**Yanhua Li**
Worcester Polytechnic Institute

**Zhenming Liu**
William & Mary

**Varun Kanade**
University of Oxford

## Abstract

We study the low rank regression problem $\mathbf{y} = M\mathbf{x} + \epsilon$, where $\mathbf{x}$ and $\mathbf{y}$ are $d_1$ and $d_2$ dimensional vectors respectively. We consider the extreme high-dimensional setting where the number of observations $n$ is less than $d_1 + d_2$. Existing algorithms are designed for settings where $n$ is typically as large as $\mathrm{rank}(M)(d_1 + d_2)$. This work provides an efficient algorithm which only involves two SVD, and establishes statistical guarantees on its performance. The algorithm decouples the problem by first estimating the precision matrix of the features, and then solving the matrix denoising problem. To complement the upper bound, we introduce new techniques for establishing lower bounds on the performance of any algorithm for this problem. Our preliminary experiments confirm that our algorithm often out-performs existing baselines, and is always at least competitive.

## 1 Introduction

We consider the regression problem $\mathbf{y} = M\mathbf{x} + \epsilon$ in the high dimensional setting, where $\mathbf{x} \in \mathbf{R}^{d_1}$ is the vector of features, $\mathbf{y} \in \mathbf{R}^{d_2}$ is a vector of responses, $M \in \mathbf{R}^{d_2 \times d_1}$ are the learnable parameters, and $\epsilon \sim N(0, \sigma_\epsilon^2 I_{d_2 \times d_2})$ is a noise term. High-dimensional setting refers to the case where the number of observations $n$ is insufficient for recovery and hence regularization for estimation is necessary [26, 30, 12]. This high-dimensional model is widely used in practice, such as identifying biomarkers [48], understanding risks associated with various diseases [18, 7], image recognition [34, 17], forecasting equity returns in financial markets [33, 39, 28, 8], and analyzing social networks [46, 35].

We consider the "large feature size" setting, in which the number of features $d_1$ is excessively large and can be even larger than the number of observations $n$. This setting frequently arises in practice because it is often straightforward to perform feature-engineering and produce a large number of potentially useful features in many machine learning problems. For example, in a typical equity forecasting model, $n$ is around 3,000 (i.e., using 10 years of market data), whereas the number of potentially relevant features can be in the order of thousands [33, 22, 25, 13]. In predicting the popularity of a user in an online social network, $n$ is in the order of hundreds (each day is an observation and a typical dataset contains less than three years of data) whereas the feature size can easily be more than 10k [36, 6, 38].

Existing low-rank regularization techniques (e.g., [3, 23, 26, 30, 27]) are not optimized for the large feature size setting. These results assume that either the features possess the so-called restricted isometry property [10], or their covariance matrix can be accurately estimated [30]. Therefore, their sample complexity $n$ depends on either $d_1$ or the smallest eigenvalue value $\lambda_{\min}$ of $\mathbf{x}$'s covariance matrix. For example, a mean-squared error (MSE) result that appeared in [30] is of the form

---

[*] Correspondence to: Qiong Wu <qwu05@email.wm.edu>.

[†] Currently at Google.

$O\left(\frac{r(d_1+d_2)}{n\lambda_{\min}^2}\right)$. When $n \leq d_1/\lambda_{\min}^2$, this result becomes trivial because the forecast $\hat{\mathbf{y}} = \mathbf{0}$ produces a comparable MSE. We design an efficient algorithm for the large feature size setting. Our algorithm is a simple two-stage algorithm. Let $\mathbf{X} \in \mathbf{R}^{n \times d_1}$ be a matrix that stacks together all features and $\mathbf{Y} \in \mathbf{R}^{n \times d_2}$ be the one that stacks the responses. In the first stage, we run a principal component analysis (PCA) on $\mathbf{X}$ to obtain a set of uncorrelated features $\hat{\mathbf{Z}}$. In the second stage, we run another PCA to obtain a low rank approximation of $\hat{\mathbf{Z}}^{\mathrm{T}}\mathbf{Y}$ and use it to construct an output.

While the algorithm is operationally simple, we show a powerful and generic result on using PCA to process features, a widely used practice for "dimensionality reduction" [11, 21, 19]. PCA is known to be effective to orthogonalize features by keeping only the subspace explaining large variations. But its performance can only be analyzed under the so-called factor model [40, 39]. We show the efficacy of PCA *without* the factor model assumption. Instead, PCA should be interpreted as a robust estimator of $\mathbf{x}$'s covariance matrix. The empirical estimator $C = \frac{1}{n}\mathbf{X}\mathbf{X}^{\mathrm{T}}$ in the high-dimensional setting cannot be directly used because $n \ll d_1 \times d_2$, but it exhibits an interesting regularity: the leading eigenvectors of $C$ are closer to ground truth than the remaining ones. In addition, the number of reliable eigenvectors grows as the sample size grows, so our PCA procedure projects the features along reliable eigenvectors and *dynamically* adjusts $\hat{\mathbf{Z}}$'s rank to maximally utilize the raw features. Under mild conditions on the ground-truth covariance matrix $C^*$ of $\mathbf{x}$, we show that it is always possible to decompose $\mathbf{x}$ into a set of near-independent features and a set of (discarded) features that have an inconsequential impact on a model's MSE.

When features $\mathbf{x}$ are transformed into uncorrelated ones $\mathbf{z}$, our original problem becomes $\mathbf{y} = N\mathbf{z} + \epsilon$, which can be reduced to a matrix denoising problem [16] and be solved by the second stage. Our algorithm guarantees that we can recover all singular vectors of $N$ whose associated singular values are larger than a certain threshold $\tau$. The performance guarantee can be translated into MSE bounds parametrized by commonly used variables (though, these translations usually lead to looser bounds). For example, when $N$'s rank is $r$, our result reduces the MSE from $O(\frac{r(d_1+d_2)}{n\lambda_{\min}^2})$ to $O(\frac{rd_2}{n} + n^{-c})$ for a suitably small constant $c$. The improvement is most pronounced when $n \ll d_1$.

We also provide a new matching lower bound. Our lower bound asserts that *no algorithm* can recover a fraction of singular vectors of $N$ whose associated singular values are smaller than $\rho\tau$, where $\rho$ is a "gap parameter". Our lower bound contribution is twofold. First, we introduce a notion of "local minimax", which enables us to define a lower bound parametrized by the singular values of $N$. This is a stronger lower bound than those delivered by the standard minimax framework, which are often parametrized by the rank $r$ of $N$ [26]. Second, we develop a new probabilistic technique for establishing lower bounds under the new local minimax framework. Roughly speaking, our techniques assemble a large collection of matrices that share the same singular values of $N$ but are far from each other, so no algorithm can successfully distinguish these matrices with identical spectra.

## 2  Preliminaries

**Notation.** Let $\mathbf{X} \in \mathbf{R}^{n \times d_1}$ and $\mathbf{Y} \in \mathbf{R}^{n \times d_2}$ be data matrices with their $i$-th rows representing the $i$-th observation. For matrix $A$, we denote its singular value decomposition as $A = U^A\Sigma^A(V^A)^{\mathrm{T}}$ and $\mathbf{P}_r(A) \triangleq U_r^A\Sigma_r^A V_r^{A\mathrm{T}}$ is the rank $r$ approximation obtained by keeping the top $r$ singular values and the corresponding singular vectors. When the context is clear, we drop the superscript $A$ and use $U, \Sigma$, and $V$ ($U_r, \Sigma_r$, and $V_r$) instead. Both $\sigma_i(A)$ and $\sigma_i^A$ are used to refer to $i$-th singular value of $A$. We use MATLAB notation when we refer to a specific row or column, e.g., $V_{1,:}$ is the first row of $V$ and $V_{:,1}$ is the first column. $\|A\|_F$, $\|A\|_2$, and $\|A\|_*$ are Frobenius, spectral, and nuclear norms of $A$. In general, we use boldface upper case (e.g., $\mathbf{X}$) to denote data matrices and boldface lower case (e.g., $\mathbf{x}$) to denote one sample. Regular fonts denote other matrices. Let $C^* = \mathbb{E}[\mathbf{x}\mathbf{x}^{\mathrm{T}}]$ and $C = \frac{1}{n}\mathbf{X}^{\mathrm{T}}\mathbf{X}$ be the empirical estimate of $C^*$. Let $C^* = V^*\Lambda^*(V^*)^{\mathrm{T}}$ be the eigen-decomposition of the matrix $C^*$, and $\lambda_1^* \geq \lambda_2^*, \ldots, \geq \lambda_{d_1}^* \geq 0$ be the diagonal entries of $\Lambda^*$. Let $\{\mathbf{u}_1, \mathbf{u}_2, \ldots \mathbf{u}_\ell\}$ be an arbitrary set of column vectors, and $\mathrm{Span}(\{\mathbf{u}_1, \mathbf{u}_2, \ldots, \mathbf{u}_\ell\})$ be the subspace spanned by it. An event happens with high probability means that it happens with probability $\geq 1 - n^{-5}$, where 5 is an arbitrarily chosen large constant and is not optimized.

**Our model.**  We consider the model $\mathbf{y} = M\mathbf{x} + \epsilon$, where $\mathbf{x} \in \mathbf{R}^{d_1}$ is a multivariate Gaussian, $\mathbf{y} \in \mathbf{R}^{d_2}$, $M \in \mathbf{R}^{d_2 \times d_1}$, and $\epsilon \sim N(0, \sigma_\epsilon^2 I_{d_2 \times d_2})$. We can relax the Gaussian assumptions on $\mathbf{x}$ and

STEP-1-PCA-X($\mathbf{X}$)

1    $[U, \Sigma, V] = \text{svd}(\mathbf{X})$
2    $\Lambda = \frac{1}{n}(\Sigma^2); \lambda_i = \Lambda_{i,i}.$
3    ▷ **Gap thresholding.**
4    ▷ $\delta = n^{-O(1)}$ is a tunable parameter.
5    $k_1 = \max\{k_1 : \lambda_{k_1} - \lambda_{k_1+1} \geq \delta\},$
6    $\Lambda_{k_1}$: diagonal matrix comprised of $\{\lambda_i\}_{i \leq k_1}.$
7    $U_{k_1}, V_{k_1}$: $k_1$ leading columns of $U$ and $V$.
8    $\hat{\Pi} = (\Lambda_{k_1})^{-\frac{1}{2}} V_{k_1}^{\mathrm{T}}$
9    $\hat{\mathbf{Z}}_+ = \sqrt{n} U_{k_1} (= X\hat{\Pi}^{\mathrm{T}}).$
10   **return** $\{\hat{\mathbf{Z}}_+, \hat{\Pi}\}.$

STEP-2-PCA-DENOISE($\hat{\mathbf{Z}}_+, \mathbf{Y}$)

1    $\hat{N}_+^{\mathrm{T}} \leftarrow \frac{1}{n}\hat{\mathbf{Z}}_+^{\mathrm{T}}\mathbf{Y}.$
2    ▷ **Absolute value thresholding.**
3    ▷ $\theta$ is a suitable constant; $\sigma_\epsilon$ is std. of the noise.
4    $k_2 = \max\left\{k_2 : \sigma_{k_2}(\hat{N}_+) \geq \theta\sigma_\epsilon\sqrt{\frac{d_2}{n}}\right\}.$
5    **return** $\mathbf{P}_{k_2}(\hat{N}_+)$

ADAPTIVE-RRR($\mathbf{X}, \mathbf{Y}$)

1    $[\hat{Z}_+, \hat{\Pi}] = \text{STEP-1-PCA-A}(\mathbf{X}).$
2    $\mathbf{P}_{k_2}(\hat{N}_+) = \text{STEP-2-PCA-DENOISE}(\hat{Z}_+, \mathbf{Y}).$
3    **return** $\hat{M} = \mathbf{P}_{k_2}(\hat{N}_+)\hat{\Pi}$

**Figure 1:** Our algorithm (ADAPTIVE-RRR) for solving the regression $\mathbf{y} = M\mathbf{x} + \epsilon$.

$\epsilon$ for most results we develop. We assume a PAC learning framework, i.e., we observe a sequence $\{(\mathbf{x}_i, \mathbf{y}_i)\}_{i \leq n}$ of independent samples and our goal is to find an $\hat{M}$ that minimizes the test error $\mathbb{E}_{\mathbf{x},\mathbf{y}}[\|\hat{M}\mathbf{x} - M\mathbf{x}\|_2^2]$. We are specifically interested in the setting in which $d_2 \approx n \leq d_1$.

The key assumption we make to circumvent the $d_1 \geq n$ issue is that the features are correlated. This assumption can be justified for the following reasons: *(i)* In practice, it is difficult, if not impossible, to construct completely uncorrelated features. *(ii)* When $n \ll d_1$, it is not even possible to *test* whether the features are uncorrelated [5]. *(iii)* When we indeed know that the features are independent, there are significantly simpler methods to design models. For example, we can build multiple models such that each model regresses on an individual feature of $\mathbf{x}$, and then use a boosting/bagging method [19, 37] to consolidate the predictions.

The correlatedness assumption implies that the eigenvalues of $C^*$ *decays*. The only (full rank) positive semidefinite matrices that have non-decaying (uniform) eigenvalues are the identity matrix (up to some scaling). In other words, when $C^*$ has uniform eigenvalues, $\mathbf{x}$ has to be uncorrelated.

We aim to design an algorithm that works *even when* the decay is slow, such as when $\lambda_i(C^*)$ has a heavy tail. Specifically, our algorithm assumes $\lambda_i$'s are bounded by a heavy-tail power law series:

**Assumption 2.1.** *The $\lambda_i(C^*)$ series satisfies $\lambda_i(C^*) \leq c \cdot i^{-\omega}$ for a constant $c$ and $\omega \geq 2$.*

We *do not* make functional form assumptions on $\lambda_i$'s. This assumption also covers many benign cases, such as when $C^*$ has low rank or its eigenvalues decay exponentially. Many empirical studies report power law distributions of data covariance matrices [2, 31, 44, 14]. Next, we make standard normalization assumptions. $\mathbb{E}\|\mathbf{x}\|_2^2 = 1$, $\|M\|_2 \leq \Upsilon = O(1)$, and $\sigma_\epsilon \geq 1$. Remark that we assume only the spectral norm of $M$ is bounded, while its Frobenius norm can be unbounded. Also, we assume the noise $\sigma_\epsilon \geq 1$ is sufficiently large, which is more important in practice. The case when $\sigma_\epsilon$ is small can be tackled in a similar fashion. Finally, our studies avoid examining excessively unrealistic cases, so we assume $d_1 \leq d_2^3$. We examine the setting where existing algorithms fail to deliver non-trivial MSE, so we assume that $n \leq rd_1 \leq d_2^4$.

## 3   Upper bound

Our algorithm (see Fig. 1) consists of two steps. **Step 1. Producing uncorrelated features.** We run a PCA to obtain a total number of $k_1$ orthogonalized features. See STEP-1-PCA-X in Fig. 1. Let the SVD of $\mathbf{X}$ be $\mathbf{X} = U\Sigma(V)^{\mathrm{T}}$. Let $k_1$ be a suitable rank chosen by inspecting the gaps of $\mathbf{X}$'s singular values (Line 5 in STEP-1-PCA-X). $\hat{\mathbf{Z}}_+ = \sqrt{n}U_{k_1}$ is the set of transformed features output by this step. The subscript $+$ in $\hat{\mathbf{Z}}_+$ reflects that a dimension reduction happens so the number of columns in $\hat{\mathbf{Z}}_+$ is smaller than that in $\mathbf{X}$. Compared to standard PCA dimension reduction, there are two differences: *(i)* We use the left leading singular vectors of $\mathbf{X}$ (with a re-scaling factor $\sqrt{n}$) as the output, whereas the PCA reduction outputs $\mathbf{P}_{k_1}(\mathbf{X})$. *(ii)* We design a specialized rule to choose $k_1$ whereas PCA usually uses a hard thresholding or other ad-hoc rules. **Step 2. Matrix denoising.** We run a second PCA on the matrix $(\hat{N}_+)^{\mathrm{T}} \triangleq \frac{1}{n}\hat{\mathbf{Z}}_+^{\mathrm{T}}\mathbf{Y}$. The rank $k_2$ is chosen by a hard thresholding rule (Line 4 in STEP-2-PCA-DENOISE). Our final estimator is $\mathbf{P}_{k_2}(\hat{N}_+)\hat{\Pi}$, where $\hat{\Pi} = (\Lambda_{k_1})^{-\frac{1}{2}}V_{k_1}^{\mathrm{T}}$ is computed in STEP-1-PCA-X($\mathbf{X}$).

## 3.1 Intuition of the design

While the algorithm is operationally simple, its design is motivated by carefully unfolding the statistical structure of the problem. We shall realize that applying PCA on the features *should not* be viewed as removing noise from a factor model, or finding subspaces that maximize variations explained by the subspaces as suggested in the standard literature [19, 40, 41]. Instead, it implicitly implements a robust estimator for $\mathbf{x}$'s precision matrix, and the design of the estimator needs to be coupled with our objective of forecasting $\mathbf{y}$, thus resulting in a new way of choosing the rank.

**Design motivation: warm up.** We first examine a simplified problem $\mathbf{y} = N\mathbf{z} + \epsilon$, where variables in $\mathbf{z}$ are assumed to be uncorrelated. Assume $d = d_1 = d_2$ in this simplified setting. Observe that

$$\frac{1}{n}\mathbf{Z}^{\mathrm{T}}\mathbf{Y} = \frac{1}{n}\mathbf{Z}^{\mathrm{T}}(\mathbf{Z}N^{\mathrm{T}} + E) = (\frac{1}{n}\mathbf{Z}^{\mathrm{T}}\mathbf{Z})N^{\mathrm{T}} + \frac{1}{n}\mathbf{Z}^{\mathrm{T}}E \approx I_{d_1 \times d_1}N^{\mathrm{T}} + \frac{1}{n}\mathbf{Z}^{\mathrm{T}}E = N^{\mathrm{T}} + \mathcal{E}, \quad (1)$$

where $E$ is the noise term and $\mathcal{E}$ can be approximated by a matrix with independent zero-mean noises. *Solving the matrix denoising problem.* Eq. 1 implies that when we compute $\mathbf{Z}^{\mathrm{T}}\mathbf{Y}$, the problem reduces to an extensively studied matrix denoising problem [16, 20]. We include the intuition for solving this problem for completeness. The signal $N^{\mathrm{T}}$ is overlaid with a noise matrix $\mathcal{E}$. $\mathcal{E}$ will elevate all the singular values of $N^{\mathrm{T}}$ by an order of $\sigma_\epsilon\sqrt{d/n}$. We run a PCA to extract reliable signals: when the singular value of a subspace is $\gg \sigma_\epsilon\sqrt{d/n}$, the subspace contains significantly more signal than noise and thus we keep the subspace. Similarly, a subspace associated a singular value $\lesssim \sigma_\epsilon\sqrt{d/n}$ mostly contains noise. This leads to a hard thresholding algorithm that sets $\hat{N}^{\mathrm{T}} = \mathbf{P}_r(N^{\mathrm{T}} + \mathcal{E})$, where $r$ is the maximum index such that $\sigma_r(N^{\mathrm{T}} + \mathcal{E}) \geq c\sqrt{d/n}$ for some constant $c$. In the general setting $\mathbf{y} = M\mathbf{x} + \epsilon$, $\mathbf{x}$ may not be uncorrelated. But when we set $\mathbf{z} = (\Lambda^*)^{-\frac{1}{2}}(V^*)^{\mathrm{T}}\mathbf{x}$, we see that $\mathbb{E}[\mathbf{z}\mathbf{z}^{\mathrm{T}}] = I$. This means knowing $C^*$ suffices to reduce the original problem to a simplified one. Therefore, our algorithm uses Step 1 to estimate $C^*$ and $\mathbf{Z}$, and uses Step 2 to reduce the problem to a matrix denoising one and solve it by standard thresholding techniques.

**Relationship between PCA and precision matrix estimation.** In step 1, while we plan to estimate $C^*$, our algorithm runs a PCA on $\mathbf{X}$. We observe that empirical covariance matrix $C = \frac{1}{n}\mathbf{X}^{\mathrm{T}}\mathbf{X} = \frac{1}{n}V(\Sigma)^2(V)^{\mathrm{T}}$, i.e., $C$'s eigenvectors coincide with $\mathbf{X}$'s right singular vectors. When we use the empirical estimator to construct $\hat{\mathbf{z}}$, we obtain $\hat{\mathbf{z}} = \sqrt{n}(\Sigma)^{-1}(V)^{\mathrm{T}}\mathbf{x}$. When we apply this map to every training point and assemble the new feature matrix, we exactly get $\hat{\mathbf{Z}} = \sqrt{n}\mathbf{X}V(\Sigma)^{-1} = \sqrt{n}U$. It means that using $C$ to construct $\hat{\mathbf{z}}$ is the same as running a PCA in STEP-1-PCA-X with $k_1 = d_1$.

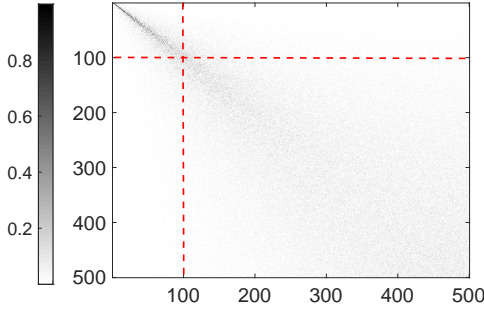

When $k_1 < d_1$, PCA uses a low rank approximation of $C$ as an estimator for $C^*$. We now explain why this is effective. First, note that $C$ is *very far* from $C^*$ when $n \ll d_1$, therefore it is dangerous to directly plug in $C$ to find $\hat{\mathbf{z}}$. Second, an interesting regularity of $C$ exists and can be best explained by a picture. In Fig. 2, we plot the pairwise angles between eigenvectors of $C$ and those of $C^*$ from a synthetic dataset. Columns are sorted by the $C^*$'s eigenvalues in decreasing order. When $C^*$ and $C$ coincide, this plot would look like an identity matrix. When $C$ and $C^*$ are unrelated, then the plot behaves like a block of white Gaussian noise. We

**Figure 2:** The angle matrix between $C$ and $C^*$.

observe a pronounced pattern: the angle matrix can be roughly divided into two sub-blocks (see the red lines in Fig. 2). The upper left sub-block behaves like an identity matrix, suggesting that the leading eigenvectors of $C$ are close to those of $C^*$. The lower right block behaves like a white noise matrix, suggesting that the "small" eigenvectors of $C$ are far from those of $C^*$. When $n$ grows, one can observe the upper left block becomes larger and this the eigenvectors of $C$ will sequentially get stabilized. Leading eigenvectors are first stabilized, followed by smaller ones. Our algorithm leverages this regularity by keeping only a suitable number of reliable eigenvectors from $C$ while ensuring not much information is lost when we throw away those "small" eigenvectors.

**Implementing the rank selection.** We rely on three interacting building blocks:

*1. Dimension-free matrix concentration.* First, we need to find a concentration behavior of $C$ for $n \leq d_1$ to decouple $d_1$ from the MSE bound. We utilize a dimension-free matrix concentration inequality [32]. Roughly speaking, the concentration behaves as $\|C - C^*\|_2 \approx n^{-\frac{1}{2}}$. This guarantees that $|\lambda_i(C) - \lambda_i(C^*)| \leq n^{-\frac{1}{2}}$ by standard matrix perturbation results [24].

*2. Davis-Kahan perturbation result.* However, the pairwise closeness of the $\lambda_i$'s does not imply the eigenvectors are also close. When $\lambda_i(C^*)$ and $\lambda_{i+1}(C^*)$ are close, the corresponding eigenvectors in $C$ can be "jammed" together. Thus, we need to identify an index $i$, at which $\lambda_i(C^*) - \lambda_{i+1}(C^*)$ exhibits significant gap, and use a Davis-Kahan result to show that $\mathbf{P}_i(C)$ is close to $\mathbf{P}_i(C^*)$. On the other hand, the map $\Pi^*(\triangleq (\Lambda^*)^{-\frac{1}{2}}(V^*)^{\mathrm{T}})$ we aim to find depends on the square root of inverse $(\Lambda^*)^{-\frac{1}{2}}$, so we need additional manipulation to argue our estimate is close to $(\Lambda^*)^{-\frac{1}{2}}(V^*)^{\mathrm{T}}$.

*3. The connection between gap and tail.* Finally, the performance of our procedure is also characterized by the total volume of signals that are discarded, i.e., $\sum_{i>k_1} \lambda_i(C^*)$, where $k_1$ is the location that exhibits the gap. The question becomes whether it is possible to identify a $k_1$ that simultaneously exhibits a large gap and ensures the tail after it is well-controlled, e.g., the sum of the tail is $O(n^{-c})$ for a constant $c$. We develop a combinatorial analysis to show that it is *always possible* to find such a gap under the assumption that $\lambda_i(C^*)$ is bounded by a power law distribution with exponent $\omega \geq 2$. Combining all these three building blocks, we have:

**Proposition 1.** *Let $\xi$ and $\delta$ be two tunable parameters such that $\xi = \omega(\log^3 n/\sqrt{n})$ and $\delta^3 = \omega(\xi)$. Assume that $\lambda_i^* \leq c \cdot i^{-\omega}$. Consider running STEP-1-PCA-X in Fig. 1, with high probability, we have (i) Leading eigenvectors/values are close: there exists a unitary matrix $W$ and a constant $c_1$ such that $\|V_{k_1}(\Lambda_{k_1})^{-\frac{1}{2}} - V_{k_1}^*(\Lambda_{k_1}^*)^{-\frac{1}{2}}W\| \leq \frac{c_1 \xi}{\delta^3}$. (ii) Small tail: $\sum_{i \geq k_1} \lambda_i^* \leq c_2 \delta^{\frac{\omega-1}{\omega+1}}$ for a constant $c_2$.*

Prop. 1 implies that our estimate $\hat{\mathbf{z}}_+ = \hat{\Pi}(\mathbf{x})$ is sufficiently close to $\mathbf{z} = \Pi^*(\mathbf{x})$, up to a unitary transform. We then execute STEP-2-PCA-DENOISE to reduce the problem to a matrix denoising one and solve it by hard-thresholding. Let us refer to $\mathbf{y} = N\mathbf{z} + \epsilon$, where $\mathbf{z}$ is a standard multivariate Gaussian and $N = MV^*(\Lambda^*)^{\frac{1}{2}}$ as the *orthogonalized form* of the problem. While we do not directly observe $\mathbf{z}$, our performance is characterized by spectra structure of $N$.

**Theorem 1.** *Consider running ADAPTIVE-RRR in Fig. 1 on $n$ independent samples $(\mathbf{x}, \mathbf{y})$ from the model $\mathbf{y} = M\mathbf{x} + \epsilon$, where $\mathbf{x} \in \mathbf{R}^{d_1}$ and $\mathbf{y} \in \mathbf{R}^{d_2}$. Let $C^* = \mathbb{E}[\mathbf{x}\mathbf{x}^{\mathrm{T}}]$. Assume that (i) $\|M\|_2 \leq \Upsilon = O(1)$, and (ii) $\mathbf{x}$ is a multivariate Gaussian with $\|\mathbf{x}\|_2 = 1$. In addition, $\lambda_1(C^*) < 1$ and for all $i$, $\lambda_i(C^*) \leq c/i^\omega$ for a constant $c$, and (iii) $\epsilon \sim N(0, \sigma_\epsilon^2 I_{d_1})$, where $\sigma_\epsilon \geq \min\{\Upsilon, 1\}$.*

*Let $\xi = \omega(\log^3 n/\sqrt{n})$, $\delta^3 = \omega(\xi)$, and $\theta$ be a suitably large constant. Let $\mathbf{y} = N\mathbf{z} + \epsilon$ be the orthogonalized form of the problem. Let $\ell^*$ be the largest index such that $\sigma_{\ell^*}^N > \theta\sigma_\epsilon\sqrt{\frac{d_2}{n}}$. Let $\hat{\mathbf{y}}$ be our testing forecast. With high probability over the training data:*

$$\mathbb{E}[\|\hat{\mathbf{y}} - \mathbf{y}\|_2^2] \leq \sum_{i>\ell^*}(\sigma_i^N)^2 + O\left(\frac{\ell^* d_2 \theta^2 \sigma_\epsilon^2}{n}\right) + O\left(\sqrt{\frac{\xi}{\delta^3}}\right) + O\left(\delta^{\frac{\omega-1}{4(\omega+1)}}\right) \qquad (2)$$

*The expectation is over the randomness of the test data.*

Theorem 1 also implies that there exists a way to parametrize $\xi$ and $\delta$ such that $\mathbb{E}[\|\hat{\mathbf{y}} - \mathbf{y}\|_2^2] \leq \sum_{i>\ell^*}(\sigma_i^N)^2 + O\left(\frac{\ell^* d_2 \theta^2 \sigma_\epsilon^2}{n}\right) + O(n^{-c_0})$ for some constant $c_0$. We next interpret each term in (2).

*Terms $\sum_{i>\ell^*}(\sigma_i^N)^2 + O\left(\frac{\ell^* d_2 \theta^2 \sigma_\epsilon^2}{n}\right)$* are typical for solving a matrix denoising problem $\hat{N}_+^{\mathrm{T}} + \mathcal{E} (\approx N^{\mathrm{T}} + \mathcal{E})$: we can extract signals associated with $\ell^*$ leading singular vectors of $N$, so $\sum_{i>\ell^*}(\sigma_i^N)^2$ starts at $i > \ell^*$. For each direction we extract, we need to pay a noise term of order $\theta^2 \sigma_\epsilon^2 \frac{d_2}{n}$, leading to the term $O\left(\frac{\ell^* d_2 \theta^2 \sigma_\epsilon^2}{n}\right)$. *Terms $O\left(\sqrt{\frac{\xi}{\delta^3}}\right) + O\left(\delta^{\frac{\omega-1}{4(\omega+1)}}\right)$* come from the estimations error of $\hat{\mathbf{z}}_+$ produced from Prop. 1, consisting of both estimation errors of $C^*$'s leading eigenvectors and the error of cutting out a tail. We pay an exponent of $\frac{1}{4}$ on both terms (e.g., $\delta^{\frac{\omega-1}{\omega+1}}$ in Prop. 1 becomes $\delta^{\frac{\omega-1}{4(\omega+1)}}$) because we used Cauchy-Schwarz (CS) twice. One is used in running matrix denoising algorithm with inaccurate $\mathbf{z}_+$; the other one is used to bound the impact of cutting a tail. It remains open whether two CS is can be circumvented.

Sec. 4 explains how Thm 1 and the lower bound imply the algorithm is near-optimal. Sec. 5 compares our result with existing ones under other parametrizations, e.g. $\text{rank}(M)$.

# 4   Lower bound

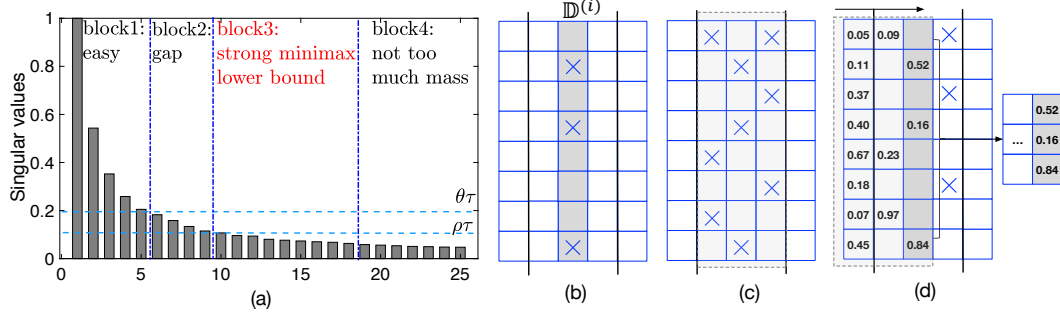

**Figure 3:** (a) **Major result:** signals in $N$ are partitioned into four blocks. All signals in block 1 can be estimated (Thm 1). All signals in block 3 cannot be estimated (Prop 2). Our lower bound techniques does not handle a small tail in Block 4. A gap in block 2 exists between upper and lower bounds. (b)-(d) **Constructing** $\mathbb{N}$: Step 1 and 2 belong to the first stage; step 3 belongs to the second stage. (b) Step 1. Generate a random subset $\mathbb{D}^{(i)}$ for each row $i$, representing its non-zero positions. (c) Step 2. Randomly sample from $\mathbb{D}$, where $\mathbb{D}$ is the Cartesian product of $\mathbb{D}^{(i)}$. (d) Step 3. Fill in non-zero entries sequentially from left to right.

Our algorithm accurately estimates the singular vectors of $N$ that correspond to singular values above the threshold $\tau = \theta \sigma_\epsilon \sqrt{\frac{d_2}{n}}$. However, it may well happen that most of the spectral 'mass' of $N$ lies only slightly below this threshold $\tau$. In this section, we establish that *no algorithm* can do better than us, in a bi-criteria sense, i.e. we show that any algorithm that has a slightly smaller sample than ours can only minimally outperform ours in terms of MSE.

We establish 'instance dependent' lower bounds: When there is more 'spectral mass' below the threshold, the performance of our algorithm will be worse, and we will need to establish that no algorithm can do much better. This departs from the standard minimax framework, in which one examines the entire parameter space of $N$, e.g. all rank $r$ matrices, and produces a large set of statistically indistinguishable 'bad' instances [43]. These lower bounds are not sensitive to instance-specific quantities such as the spectrum of $N$, and in particular, if prior knowledge suggests that the unknown parameter $N$ is far from these bad instances, the minimax lower bound cannot be applied.

We introduce the notion of *local minimax*. We partition the space into parts so that *similar* matrices are together. Similar matrices are those $N$ that have the same singular values and right singular vectors; we establish strong lower bounds even against algorithms that know the singular values and right singular vectors of $N$. An equivalent view is to assume that the algorithm has oracle access to $C^*$, $M$'s singular values, and $M$'s right singular vectors. This algorithm can solve the orthogonalized form as $N$'s singular values and right singular vectors can easily be deduced. Thus, the only reason why the algorithm needs data is to *learn* the left singular vectors of $N$. The lower bound we establish is the minimax bound for this 'unfair' comparison, where the competing algorithm is given more information. In fact, this can be reduced further, i.e., even if the algorithm 'knows' that the left singular vectors of $N$ are sparse, identifying the locations of the non-zero entries is the key difficulty that leads to the lower bound.

**Definition 1** (Local minimax bound). *Consider a model* $\mathbf{y} = M\mathbf{x} + \epsilon$*, where* $\mathbf{x}$ *is a random vector, so* $C^*(\mathbf{x}) = \mathbb{E}[\mathbf{x}\mathbf{x}^{\mathrm{T}}]$ *represents the co-variance matrix of the data distribution, and* $M = U^M \Sigma^M (V^M)^{\mathrm{T}}$*. The relation* $(M, \mathbf{x}) \sim (M', \mathbf{x}') \Leftrightarrow (\Sigma^M = \Sigma^{M'} \wedge V^M = V^{M'} \wedge C^*(\mathbf{x}) = C^*(\mathbf{x}'))$ *is an equivalence relation and let the equivalence class of* $(M, \mathbf{x})$ *be* $\mathcal{R}(M, \mathbf{x}) = \{(M', \mathbf{x}') : \Sigma^{M'} = \Sigma^M, V^{M'} = V^M, \text{ and } C^*(\mathbf{x}') = C^*(\mathbf{x})\}$*. The local minimax bound for* $\mathbf{y} = M\mathbf{x} + \epsilon$ *with* $n$ *independent samples and* $\epsilon \sim N(0, \sigma_\epsilon^2 I_{d_2 \times d_2})$ *is*

$$\mathbf{r}(\mathbf{x}, M, n, \sigma_\epsilon) = \min_{\hat{M}} \max_{(M', \mathbf{x}') \in \mathcal{R}(M, \mathbf{x})} E_{\substack{\mathbf{X}, \mathbf{Y} \text{ from} \\ \mathbf{y} \sim M'\mathbf{x}' + \epsilon}} \left[ \mathbb{E}_{\mathbf{x}'}[\|\hat{M}(\mathbf{X}, \mathbf{Y})\mathbf{x}' - M'\mathbf{x}'\|_2^2 \mid \mathbf{X}, \mathbf{Y}] \right]. \quad (3)$$

It is worth interpreting (3) in some detail. For any two $(M, \mathbf{x})$, $(M', \mathbf{x}')$ in $\mathcal{R}(M, \mathbf{x})$, the algorithm has the same 'prior knowledge', so it can only distinguish between the two instances by using the *observed data*, in particular $\hat{M}$ is a function only of $\mathbf{X}$ and $\mathbf{Y}$, and we denote it as $\hat{M}(\mathbf{X}, \mathbf{Y})$ to emphasize this. Thus, we can evaluate the performance of $\hat{M}$ by looking at the worst possible $(M', \mathbf{x}')$ and considering the MSE $\mathbb{E}\|\hat{M}(\mathbf{X}, \mathbf{Y})\mathbf{x}' - M'\mathbf{x}'\|^2$.

**Proposition 2.** *Consider the problem* $\mathbf{y} = M\mathbf{x} + \epsilon$ *with normalized form* $\mathbf{y} = N\mathbf{z} + \epsilon$. *Let* $\xi$ *be a sufficient small constant. There exists a sufficiently small constant* $\rho_0$ *(that depends on* $\xi$*) and a constant c such that for any* $\rho \leq \rho_0$, $\mathbf{r}(\mathbf{x}, M, n, \sigma_\epsilon) \geq (1 - c\rho^{\frac{1}{2}-\xi})\sum_{i \geq \underline{t}}(\sigma_i^N)^2 - O\left(\frac{\rho^{\frac{1}{2}-\xi}}{d_2^{\omega-1}}\right)$, *where* $\underline{t}$ *is the smallest index such that* $\sigma_{\underline{t}}^N \leq \rho\sigma_\epsilon\sqrt{\frac{d_2}{n}}$.

Proposition 2 gives the lower bound on the MSE in expectation; it can be turned into a high probability result with suitable modifications. The proof of the lower bound uses a similar 'trick' to the one used in the analysis of the upper bound analysis to cut the tail. This results in an additional term $O\left(\frac{\rho^{\frac{1}{2}-\xi}}{d_2^{\omega-1}}\right)$ which is generally smaller than the $n^{-c_0}$ tail term in Theorem 1 and does not dominate the gap.

**Gap requirement and bi-criteria approximation algorithms.** Let $\tau = \sigma_\epsilon\sqrt{\frac{d_2}{n}}$. Theorem 1 asserts that any signal above the threshold $\theta\tau$ can be detected, i.e., the MSE is at most $\sum_{\sigma_i^N > \theta\tau}\sigma_i^2(N)$ (plus inevitable noise), whereas Proposition 2 asserts that any signal below the threshold $\rho\tau$ cannot be detected, i.e., the MSE is approximately at least $\sum_{\sigma_i^N \geq \rho\tau}(1 - \text{poly}(\rho))\sigma_i^2(N)$. There is a 'gap' between $\theta\tau$ and $\rho\tau$, as $\theta > 1$ and $\rho < 1$. See Fig. 3(a). This kind of gap is inevitable because both bounds are 'high probability' statements. This gap phenomenon appears naturally when the sample size is small as can be illustrated by this simple example. Consider the problem of estimating $\mu$ when we see one sample from $N(\mu, \sigma^2)$. Roughly speaking, when $\mu \gg \sigma$, the estimation is feasible, and whereas $\mu \ll \sigma$, the estimation is impossible. For the region $\mu \approx \sigma$, algorithms fail with constant probability and we cannot prove a high probability lower bound either.

While many of the signals can 'hide' in the gap, the inability to detect signals in the gap is a transient phenomenon. When the number of samples $n$ is modestly increased, our detection threshold $\tau = \theta\sigma_\epsilon\sqrt{\frac{d_2}{n}}$ shrinks, and this hidden signal can be fully recovered. This observation naturally leads to a notion of bi-criteria optimization that frequently arises in approximation algorithms.

**Definition 2.** *An algorithm for solving the* $\mathbf{y} = M\mathbf{x} + \epsilon$ *problem is* $(\alpha, \beta)$*-optimal if, when given an i.i.d. sample of size* $\alpha n$ *as input, it outputs an estimator whose MSE is at most* $\beta$ *worse than the local minimax bound, i.e.,* $\mathbb{E}[\|\hat{\mathbf{y}} - \mathbf{y}\|_2^2] \leq \mathbf{r}(\mathbf{x}, M, n, \sigma_\epsilon) + \beta$.

**Corollary 1.** *Let* $\xi$ *and* $c_0$ *be small constants and* $\rho$ *be a tunable parameter. Our algorithm is* $(\alpha, \beta)$*-optimal for* $\alpha = \frac{\theta^2}{\rho^{\frac{5}{2}}}$ *and* $\beta = O(\rho^{\frac{1}{2}-\xi})\|M\mathbf{x}\|_2^2 + O(n^{-c_0})$

The error term $\beta$ consists of $\rho^{\frac{1}{2}-\xi}\|M\mathbf{x}\|_2^2$ that is directly characterized by the signal strength and an additive term $O(n^{-c_0}) = o(1)$. Assuming that $\|M\mathbf{x}\| = \Omega(1)$, i.e., the signal is not too weak, the term $\beta$ becomes a single multiplicative bound $O(\rho^{\frac{1}{2}-\xi} + n^{-c_0})\|M\mathbf{x}\|_2^2$. This gives an easily interpretable result. For example, when our data size is $n \log n$, the performance gap between our algorithm and *any* algorithm that uses $n$ samples is at most $o(\|M\mathbf{x}\|_2^2)$. The improvement is significant when other baselines deliver MSE in the additive form that could be larger than $\|M\mathbf{x}\|_2^2$ in the regime $n \leq d_1$.

**Preview of techniques.** Let $N = U^N\Sigma^N(V^N)^{\mathrm{T}}$ be the instance (in orthogonalized form). Our goal is to construct a collection $\mathcal{N} = \{N_1, \ldots, N_K\}$ of $K$ matrices so that *(i)* For any $N_i \in \mathcal{N}$, $\Sigma^{N_i} = \Sigma^N$ and $V^{N_i} = V^N$. *(ii)* For any two $N_i, N_j \in \mathcal{N}$, $\|N - N'\|_F$ is large, and *(iii)* $K = \exp(\Omega(\text{poly}(\rho)d^2))$ (cf. [43, Chap. 2])

Condition (i) ensures that it suffices to construct unitary matrices $U^{N_i}$'s for $\mathcal{N}$, and that the resulting instances will be in the same equivalence class. Conditions (ii) and (iii) resemble standard construction of codes in information theory: we need a large 'code rate', corresponding to requiring a large $K$ as well as large distances between codewords, corresponding to requiring that $\|U_i - U_j\|_F$ be large. Standard approaches for constructing such collections run into difficulties. Getting a sufficiently tight concentration bound on the distance between two random unitary matrices is difficult as the matrix

**Table 1:** Summary of results for equity return forecasts (left) and average results for Twitter (right) from 10 random samples. $R^2$ are measured by basis points (bps). $1\text{bps} = 10^{-4}$. Bold font denotes the best *out-of-sample* results and smallest gap. $out - in$ denotes $\text{MSE}_{out-in}$

| Model | Equity return | | | | | Twitter dataset | | |
|---|---|---|---|---|---|---|---|---|
| | $R^2_{out}$ | Sharp | t-stat | $\text{MSE}_{out}$ | $out - in$ | $\text{corr}_{out}$ | $\text{MSE}_{out}$ | $out - in$ |
| ADAPTIVE-RRR | **18.576** | **1.623** | **15.413** | **1.005** | **0.1006** | **0.67 ± 0.13** | **9.42 ± 2.31** | **4.417** |
| Lasso | 1.124 | 0.595 | 0.018 | 1.063 | 0.534 | 0.47 ± 0.15 | 14.82 ± 4.81 | 12.452 |
| Ridge | 0.212 | 0.574 | 0.067 | 1.029 | 0.355 | 0.47 ± 0.17 | 13.62 ± 4.39 | 12.2 27 |
| Reduced ridge | 1.082 | 1.548 | 0.062 | 1.972 | 1.235 | 0.49 ± 0.18 | 12.23 ± 2.70 | 7.708 |
| RRR | 4.580 | -0.477 | 0.640 | 1.087 | 0.474 | 0.38 ± 0.22 | 13.07 ± 2.63 | 8.731 |
| Nuclear norm | 2.210 | -0.370 | -0.899 | 1.109 | 0.955 | 0.48 ± 0.16 | 13.05 ± 4.38 | 8.668 |
| PCR | 5.233 | 1.280 | 0.699 | 1.026 | 0.493 | 0.48 ± 0.15 | 13.08 ± 4.19 | 8.889 |

entries, by necessity, are correlated. On the other hand, starting with a large collection of random unit vectors and using its Cartesian product to build matrices does not necessarily yield unitary matrices.

We design a two-stage approach to decouple condition (iii) from (i) and (ii) by only generating sparse matrices $U^{N_i}$. See Fig. 3(b)-(d). In the first stage (Steps 1 & 2 in Fig. 3(b)-(c)), we only specify the non-zero positions (sparsity pattern) in each $U^{N_i}$. It suffices to guarantee that the sparsity patterns of the matrices $U^{N_i}$ and $U^{N_j}$ have little overlap. The existence of such objects can easily be proved using the probabilistic method. Thus, in the first stage, we can build up a large number of sparsity patterns. In the second stage (Step 3 in Fig. 3(d)), we carefully fill in values in the non-zero positions for each $U^{N_i}$. When the number of non-zero entries is not too small, satisfying the unitary constraint is feasible. As the overlap of sparsity patterns of any two matrices is small, we can argue the distance between them is large. By carefully trading off the number of non-zero positions and the portion of overlap, we can simultaneously satisfy all three conditions.

## 5   Related work and comparison

In this section, we compare our results to other regression algorithms that make low rank constraints on $M$. Most existing MSE results are parametrized by the rank or spectral properties of $M$, e.g. [30] defined a generalized notion of rank $\mathbb{B}_q(R_q^A) \in \{A \in \mathbf{R}^{d_2 \times d_1} : \sum_{i=1}^{d_2} |\sigma_i^A|^q \leq R_q\}$, where $q \in [0, 1], A \in \{N, M\}$, i.e. $R_q^N$ characterizes the generalized rank of $N$ whereas $R_q^M$ characterizes that of $M$. When $q = 0$, $R_q^N = R_q^M$ is the rank of the $N$ because $\text{rank}(N) = \text{rank}(M)$ in our setting. In their setting, the MSE is parametrized by $R^M$ and is shown to be $O\left(R_q^M\left(\frac{\sigma_\epsilon^2 \lambda_1^*(d_1 + d_2)}{(\lambda_{\min}^*)^2 n}\right)^{1-q/2}\right)$. In the special case when $q = 0$, this reduces to $O\left(\frac{\sigma_\epsilon^2 \lambda_1^* \text{rank}(M)(d_1 + d_2)}{(\lambda_{\min}^*)^2 \cdot n}\right)$. On the other hand, the MSE in our case is bounded by (cf. Thm. 1). We have $\mathbb{E}[\|\hat{\mathbf{y}} - \mathbf{y}\|_2^2] = O\left(R_q^N(\frac{\sigma_\epsilon^2 d_2}{n})^{1-q/2} + n^{-c_0}\right)$. When $q = 0$, this becomes $O\left(\frac{\sigma_\epsilon^2 \text{rank}(M) d_2}{n} + n^{-c_0}\right)$.

The improvement here is twofold. First, our bound is directly characterized by $N$ in orthogonalized form, whereas result of [30] needs to examine the interaction between $M$ and $C^*$, so their MSE depends on both $R_q^M$ and $\lambda_{\min}^*$. Second, our bound no longer depends on $d_1$ and pays only an additive factor $n^{-c_0}$, thus, when $n < d_1$, our result is significantly better. Other works have different parameters in the upper bounds, but all of these existing results require that $n > d_1$ to obtain non-trivial upper bounds [26, 9, 12, 26]. Unlike these prior work, we require a stochastic assumption on $\mathbf{X}$ (the rows are i.i.d.) to ensure that the model is identifiable when $n < d_1$, e.g. there could be two sets of disjoint features that fit the training data equally well. Our algorithm produces an adaptive model whose complexity is controlled by $k_1$ and $k_2$, which are adjusted dynamically depending on the sample size and noise level. [9] and [12] also point out the need for adaptivity; however they still require $n > d_1$ and make some strong assumptions. For instance, [9] assumes that there is a gap between $\sigma_i(\mathbf{X}M^{\mathrm{T}})$ and $\sigma_{i+1}(\mathbf{X}M^{\mathrm{T}})$ for some $i$. In comparison, our sufficient condition, the decay of $\lambda_i^*$, is more natural. Our work is not directly comparable to standard variable selection techniques such as LASSO [42] because they handle univariate $\mathbf{y}$. Column selection algorithms [15] generalize variable selection methods for vector responses, but they cannot address the identifiability concern.

# 6 Experiments

We apply our algorithm on an equity market and a social network dataset to predict equity returns and user popularity respectively. Our baselines include ridge regression ("Ridge"), reduced rank ridge regression [29] ("Reduced ridge"), LASSO ("Lasso"), nuclear norm regularized regression ("Nuclear norm"), reduced rank regression [45] ("RRR"), and principal component regression [1] ("PCR").

**Predicting equity returns.** We use a stock market dataset from an emerging market that consists of approximately 3600 stocks between 2011 and 2018. We focus on predicting the *next 5-day returns*. For each asset in the universe, we compute its past 1-day, past 5-day, and past 10-day returns as features. We use a standard approach to translate forecasts into positions [4, 47]. We examine two universes in this market: (i) *Universe 1* is equivalent to S&P 500 and consists of 983 stocks, and (ii) *Full universe* consists of all stocks except for illiquid ones.

*Results.* Table 1 (left) reports the forecasting power and portfolio return for *out-of-sample* periods in *Full universe* (see our full version for *Universe 1*). We observe that *(i)* The data has a low signal-to-noise ratio. The out-of-sample $R^2$ values of all the methods are close to 0. *(ii)* ADAPTIVE-RRR has the highest forecasting power. *(iii)* ADAPTIVE-RRR has the smallest in-sample and out-of-sample gap (see column $out - in$), suggesting that our model is better at avoiding spurious signals.

**Predicting user popularity in social networks.** We collected tweet data on political topics from Oct. 2016 to Dec. 2017. Our goal is to predict a user's *next 1-day* popularity, which is defined as the sum of retweets, quotes, and replies received by the user. There are a total of 19 million distinct users, and due to the huge size, we extract the subset of 2000 users with the most interactions for evaluation. For each user in the 2000-user set, we use its past 5 days' popularity as features. We further randomly sample 200 users and make predictions for them, i.e., setting $d_2 = 200$ to make $d_2$ of the same magnitude as $n$.

*Results.* We randomly sample users for 10 times and report the average MSE and correlation (with standard deviations) for both *in-sample* and *out-of-sample* data (see full version for more results). In Table 1 (right) we can see results consistent with the equity returns experiment: *(i)* ADAPTIVE-RRR yields the best performance in out-of-sample MSE and correlation. *(ii)* ADAPTIVE-RRR achieves the best generalization error by having a much smaller gap between training and test metrics.

# 7 Conclusion

This paper examines the low-rank regression problem under the high-dimensional setting. We design the first learning algorithm with provable statistical guarantees under a mild condition on the features' covariance matrix. Our algorithm is simple and computationally more efficient than low rank methods based on optimizing nuclear norms. Our theoretical analysis of the upper bound and lower bound can be of independent interest. Our preliminary experimental results demonstrate the efficacy of our algorithm. The full version explains why our (algorithm) result is unlikely to be known or trivial.

## Broader Impact

The main contribution of this work is theoretical. Productionizing downstream applications stated in the paper may need to take six months or more so there is no immediate societal impact from this project.

## Acknowledgement

We thank anonymous reviewers for helpful comments and suggestions. Varun Kanade is supported in part by the Alan Turing Institute under the EPSRC grant EP/N510129/1. Yanhua Li was supported in part by NSF grants IIS-1942680 (CAREER), CNS-1952085, CMMI-1831140, and DGE-2021871. Qiong Wu and Zhenming Liu are supported by NSF grants NSF-2008557, NSF-1835821, and NSF-1755769. The authors acknowledge William & Mary Research Computing for providing computational resources and technical support that have contributed to the results reported within this paper.

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
