[Supplementary Material]

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

## 3.2 Analysis

We now analyze our algorithm. Our analysis consists of three steps. In step 1, we prove Proposition 1. In step 2, we relate $\mathbf{Z}^\mathrm{T}\mathbf{Y}$ with the product produced by our estimate $\hat{\mathbf{Z}}_+^\mathrm{T}\mathbf{Y}$. In step 3, we prove Theorem 1.

### 3.2.1 Step 1. PCA for the features (proof of Proposition 1)

This section proves Proposition 1. We shall first show that the algorithm always terminates. We have the following lemma.

**Lemma 1.** *Let $\{\lambda_i\}_{i \leq d}$ be a sequence such that $\sum_{i \leq n} \lambda_i = 1$, $\lambda_i \leq ci^{-\omega}$ for some constant $c$, $\omega \geq 2$, and $\lambda_1 < 1$. Define $\delta_i = \lambda_i - \lambda_{i+1}$ for $i \geq 1$. Let $\ell_0$ be a sufficiently large number, and $c_1$ and $c_2$ are two suitable constants. Let $\ell$ be any number such that $\ell \geq \ell_0$. Let $\tau$ be any parameter such that $\tau < \rho - 1$. There exists an $i^*$ such that (i) Gap is sufficiently large: $\delta_{i^*} \geq c_1 \cdot \ell^{-(\tau\omega/(\omega-1)+1)}$, and (ii) tail sum is small: $\sum_{i \geq i^*} \lambda_i \leq c_2/\ell^{-\tau}$.*

We remark that Lemma 1 will also be able to show part (ii) of Proposition 1 (this will be explained below). This lemma also corresponds to the "connection between gap and tail" building block referred in Section 3.1.

*Proof of Lemma 1.* Define the function $h(t) = \sum_{i \geq t} c/i^\omega = \frac{c_3 + o(1)}{t^{\omega-1}}$ (by Euler–Maclaurin formula), where $o(1)$ is a function of $t$

Next, let us define

$$i_1 = \min\left\{ i^* : \sum_{i \leq i^*} \lambda_i \geq 1 - h(\ell^{\frac{\tau}{\omega-1}}) \right\} - 1$$

$$i_2 = \min\left\{ i^* : \sum_{i \leq i^*} \lambda_i \geq 1 - \frac{1}{2} \times h(\ell^{\frac{\tau}{\omega-1}}) \right\} - 1.$$

Roughly speaking, we want to identify an $i_1$ such that $\sum_{i \leq i_1} \lambda_i$ is smaller than $1 - h\left(\ell^{\frac{\tau}{\omega-1}}\right)$ but is as close to it as possible. We can interpret $i_2$ in a similar manner. $i_1, i_2 \geq 1$ because of the assumption $\lambda_1 < 1$.

We can verify that $i_1 < \ell^{\frac{\tau}{\omega-1}}$ because $\sum_{i \leq \ell^{\frac{\tau}{\omega-1}}} \lambda_i \geq 1 - h\left(\ell^{\frac{\tau}{\omega-1}}\right)$. We can similarly verify that $i_2 < c_4 \ell^{\frac{\tau}{\omega-1}}$ for some constant $c_4$. Now using

$$\sum_{i \leq i_2+1} \lambda_i \geq 1 - \frac{(c_3 + o(1))\ell^{-\tau}}{2}$$

$$\sum_{i \leq i_1} \lambda_i \leq 1 - (c_3 + o(1))\ell^{-\tau}.$$

We may use an averaging argement and show that there exists an $i_3 \in [i_1 + 1, i_2 + 1]$ such that

$$\lambda_{i_3} \geq \frac{(c_3 + o(1))\ell^{-\tau}}{i_2 - i_1} \geq \frac{(c_3 + o(1))\ell^{-\tau}}{c_4 \ell^{\frac{\tau}{\omega-1}}}$$

$$\geq c_5 \ell^{-\tau - \frac{\tau}{\omega-1}} = c_5 \ell^{-\frac{\tau\omega}{\omega-1}}.$$

Note that $c_5 \ell^{-\frac{\tau\omega}{\omega-1}} \geq 2c\ell^{-\omega}$ because $\tau < \omega - 1$. Next, using that $\lambda_\ell \leq c/\ell^\omega$, we have

$$\lambda_\ell = \overbrace{\lambda_{i_3}}^{\geq c_5 \ell^{-\frac{\tau\omega}{\omega-1}}} + (\lambda_{i_3+1} - \lambda_{i_3}) + \cdots + (\lambda_\ell - \lambda_{\ell-1}) \leq \overbrace{c/\ell^\omega}^{\leq \frac{c_5}{2} \cdot \ell^{-\frac{\tau\omega}{\omega-1}}}. \tag{3}$$

This implies one of $(\lambda_{i_3} - \lambda_{i_3+1}), \ldots, (\lambda_{\ell-1} - \lambda_\ell)$ is at least $\frac{c_5}{2} \cdot \ell^{-\frac{\tau\omega}{\omega-1}}/\ell$. In other words, there exists an $i^* \in [i_3 + 1, \ell]$ such that $\lambda_{i^*} - \lambda_{i^*+1} \geq \frac{c_5}{2} \ell^{-\left(\frac{\tau\omega}{\omega-1}+1\right)}$. Finally, we can check that

$$\sum_{i \geq i^*} \lambda_{i^*} \leq \sum_{i \geq i_1} \lambda_{i^*} \leq h\left(\ell^{\frac{\tau}{\omega-1}}\right) \leq \frac{c_2}{\ell^\tau}. \tag{4}$$

$\square$

We apply Lemma 1 by setting $\tau \to \omega - 1$. There is a parameter $\ell$ that we can tune, such that it is always possible to find an $i^*$ where $\delta_{i^*} \geq c_1 \ell^{-(\omega+1)}$ and $\sum_{i \geq i^*} \lambda_i \leq 1 - c_2 \ell^{-(\omega-1)}$. For any $\delta = o(1)$ (a function of $n$), we can set $\ell = \Theta\left(\left(\frac{1}{\delta}\right)^{\frac{1}{\omega+1}}\right)$. In addition, $\sum_{i \geq k_1} \lambda_i = O\left(\delta^{\frac{\omega-1}{\omega+1}}\right)$. This also proves the second part of the Proposition.

It remains to prove part (i) of Proposition 1. It consists of three steps.

*Step 1. Dimension-free Chernoff bound for matrices.* We first give a bound on $\|C^* - C\|_2$, which characterizes the tail probability by using the first and second moments of random vectors. This is the key device enabling us to meaningfully recover signals even when $n \ll d_1$.

**Lemma 2.** *Recall that $C^* = \mathbb{E}[\mathbf{x}\mathbf{x}^{\mathrm{T}}]$ and $C = \frac{1}{n}\mathbf{X}^{\mathrm{T}}\mathbf{X}$. For any $\xi > 0$,*

$$\Pr[\|C^* - C\|_2 \geq \xi] \leq (2n^2)\exp(-n\xi^2/(\log^4 n)) + n^{-10}. \tag{5}$$

The exponent 10 is chosen arbitrarily and is not optimized.

*Proof of Lemma 2.* We use the following specific form of Chernoff bound ([35])

**Lemma 3.** *Let $\mathbf{z}_1, \mathbf{z}_2, \ldots, \mathbf{z}_n$ be i.i.d. random vectors such that $\|\mathbf{z}_i\| \leq \alpha$ a.s. and $\|\mathbb{E}[\mathbf{z}_i \mathbf{z}_i^{\mathrm{T}}]\| \leq \beta$. Then for any $\epsilon > 0$,*

$$\Pr\left[\left\|\frac{1}{n}\sum_{i \leq n}\mathbf{z}_i\mathbf{z}_i^{\mathrm{T}} - \mathbb{E}[\mathbf{z}_i\mathbf{z}_i^{\mathrm{T}}]\right\|_2 \geq \xi\right] \leq (2n^2)\exp\left(-\frac{n\xi^2}{16\beta\alpha^2 + 8\alpha^2\xi}\right) \tag{6}$$

We aim to use Lemma 3 to show Lemma 2 and we set $\mathbf{z}_i = \mathbf{x}_i$. But the $\ell_2$-norm of $\mathbf{z}_i$'s are unbounded so we need to use a simple coupling technique to circumvent the problem. Specifically, let $c_0$ be a suitable constant and define

$$\tilde{\mathbf{z}}_i = \begin{cases} \mathbf{z}_i & \text{if } |\mathbf{z}_i| \leq c_0 \log^2 n \\ 0 & \text{otherwise.} \end{cases} \tag{7}$$

By using a standard Chernoff bound, we have

$$\Pr[\exists i : \tilde{\mathbf{z}}_i \neq \mathbf{z}_i] \leq \frac{1}{n^{10}}. \tag{8}$$

Let us write $\tilde{C} = \frac{1}{n}\sum_{i \leq n}\tilde{\mathbf{z}}_i\tilde{\mathbf{z}}_i^{\mathrm{T}}$. We set $\alpha = c_0\log^2 n$ and $\beta = \Theta(1)$ in Lemma 3. One can see that

$$\Pr[\|C^* - C\|_2 \geq \xi] \leq \Pr\left[(\|\tilde{C} - C\|_2 \geq \xi) \vee (\tilde{C} \neq C)\right] \leq 2n^2\exp\left(-\frac{n\xi^2}{\log^4 n}\right) + \frac{1}{n^{10}}. \tag{9}$$

$\square$

*Step 2. Davis-Kahan bound.* The above analysis gives us that $\|C^* - C\|_2 \leq \xi$. We next show that the first a few eigenvectors of $C$ are close to those of $C^*$.

**Lemma 4.** *Let $\xi = \omega\left(\frac{\log^3 n}{\sqrt{n}}\right)$ and $\delta^3 = \omega(\xi)$. Considering running STEP-1-PCA-X in Fig. 1. Let $\mathcal{P}^* = V_{k_1}^*(V_{k_1}^*)^{\mathrm{T}}$ and $\mathcal{P} = V_{k_1}V_{k_1}^{\mathrm{T}}$. When $\|C^* - C\|_2 \leq \xi$, $\|\mathcal{P}^* - \mathcal{P}\|_2 \leq \frac{2\xi}{\delta}$.*

*Proof.* Recall that $\lambda_1^*, \lambda_2^*, \ldots, \lambda_{d_1}^*$ are the eigenvalues of $C^*$. Let also $\lambda_1, \lambda_2, \ldots, \lambda_{d_1}$ be the eigenvalues of $C$. Define

$$S_1 = [\lambda_{k_1} - \delta/10, \infty] \quad \text{and} \quad S_2 = [0, \lambda_{k_1+1} + \delta/10]. \tag{10}$$

The constant 10 is chosen in an arbitrary manner. Because $\|C^* - C\|_2 \leq \xi$, we know that $S_1$ contains $\lambda_1^*, \ldots, \lambda_k^*$ and that $S_2$ contains $\lambda_{k_1+1}^*, \ldots, \lambda_{d_1}^*$ [25]. Using the Davis-Kahan Theorem [15], we get

$$\|\mathcal{P}^* - \mathcal{P}\|_2 \leq \frac{\|C^* - C\|_2}{0.8\delta} \leq \frac{2\epsilon}{\delta} \tag{11}$$

$\square$

We also need the following building block.

**Lemma 5.** *[47] Let $A$ and $B$ be $n \times n$ positive semidefinite matrices with the same rank of $d$. Let $X$ and $Y$ be of full column rank such that $XX^{\mathrm{T}} = A$ and $YY^{\mathrm{T}} = B$. Let $\delta$ be the smallest non-zero eigenvalue of $B$. Then there exists a unitary matrix $W \in \mathbf{R}^{d \times d}$ such that*

$$\|XW - Y\|_2 \leq \frac{\|A - B\|_2(\sqrt{\|A\|_2} + \sqrt{\|B\|_2})}{\delta}.$$

*Step 3. Commuting the unitary matrix.* Roughly speaking, Lemma 4 and Lemma 5 show that there exists a unitary matrix $W$ such that $\|V_{k_1}W - V_{k_1}^*\|_2$ is close to 0. Standard matrix perturbation result also shows that $\Lambda_{k_1}$ and $\Lambda_{k_1}^*$ are close. This gives us that $V_{k_1}W\Lambda_{k_1}^{-\frac{1}{2}}$ and $V_{k_1}^*(\Lambda_{k_1}^*)^{-\frac{1}{2}}$ are close, whereas we need that $V_{k_1}\Lambda_{k_1}^{-\frac{1}{2}}W$ and $V_{k_1}^*(\Lambda_{k_1}^*)^{-\frac{1}{2}}$ are close. The unitary matrix $W$ is not in the right place. This is a standard technical obstacle for analyzing PCA based techniques [47, 18, 28]. We develop the following lemma to address the issue.

**Lemma 6.** *Let $U_1, U_2$ be $n \times d$ matrices such that $U_1^\top U_1 = U_2^\top U_2 = I$. Let $S_1$, $S_2$ be diagonal matrices with strictly positive entries, and let $W \in \mathbf{R}^{d \times d}$ be a unitary matrix. Then,*

$$\|U_1 S_1^{-1} W - U_2 S_2^{-1}\| \leq \frac{\|U_1 S_1 W - U_2 S_2\|}{\min\{(S_1)_{ii}\} \cdot \min\{(S_2)_{ii}\}} + \frac{\|U_1 U_1^\top - U_2 U_2^\top\|}{\min\{(S_2)_{ii}\}}$$

*Proof.* Observe that,

$$U_1 S_1^{-1} W - U_2 S_2^{-1} = U_1 S_1^{-1} W (S_2 U_2^\top - W^\top S_1 U_1^\top) U_2 S_2^{-1} + U_1 U_1^\top U_2 S_2^{-1} - U_2 U_2^\top U_2 S_2^{-1}.$$

The result then follows by taking spectral norms of both sides, the triangle inequality and the sub-multiplicativity of the spectral norm. □

Results from Step 1 to Step 3 suffice to prove the first part of Proposition 1. First, we use Lemma 4 and Lemma 5 (adopted from [47]) to get that

$$\|V_{k_1}^*(\Lambda_{k_1}^*)^{\frac{1}{2}}W - V_{k_1}(\Lambda_{k_1})^{\frac{1}{2}}\|_2 \leq \frac{c_0\xi}{\delta^2}. \tag{12}$$

Next, observe that $\lambda_{k_1}, \lambda_{k_1}^* = \Omega(\delta)$. By applying Lemma 6, with $U_1 = V_{k_1}^*$ and $S_1 = (\Lambda_{k_1}^*)^{\frac{1}{2}}$, $U_2 = V_{k_1}$ and $S_2 = (\Lambda_{k_1})^{\frac{1}{2}}$, we obtain

$$\|V_{k_1}\Lambda_{k_1}^{-\frac{1}{2}}W - V_{k_1}^*\Lambda_{k_1}^{*-\frac{1}{2}}\|_2 \leq \frac{\|V_{k_1}^*\Lambda_{k_1}^{*\frac{1}{2}}W - V_{k_1}\Lambda_{k_1}^{\frac{1}{2}}\|_2}{\delta} + \frac{\|\mathcal{P}^* - \mathcal{P}\|_2}{\delta} \leq \frac{c_1\xi}{\delta^3}$$

This completes the proof of Proposition 1.

### 3.2.2 Step 2. Analysis of $\mathbf{Z}^\mathrm{T}\mathbf{Y}$

**Proposition 2.** *Consider running* ADAPTIVE-RRR *in Fig. 1 to solve the regression problem* $\mathbf{y} = M\mathbf{x} + \epsilon$. *Let $\hat{\mathbf{Z}}_+$ be the output of the first stage* STEP-1-PCA-X. *Let $W$ be the unitary matrix specified in Proposition 1. Let $\hat{N}_+ = \hat{\mathbf{Z}}_+^\mathrm{T}\mathbf{Y}$. We have with high probability (over the training data),*

$$\hat{N}_+^\mathrm{T} = W^\mathrm{T}N^\mathrm{T} + \mathcal{E}_L + \mathcal{E}_T,$$

*where*

$$\|\mathcal{E}_L\|_2 \leq 2.2\sigma_\epsilon\sqrt{\frac{d_2}{n}} \quad and \quad \|\mathcal{E}_T\|_F = O(\epsilon/\delta^3).$$

The rest of this Section proves Proposition 2. Recall that $\mathbf{Y} = \mathbf{Z}N^\mathrm{T} + E$ is the orthogonalized form of our problem. Let us split $N = [N_+, N_-]$, where $N_+ \in \mathbf{R}^{d_2 \times k_1}$ consists of the $k_1$ leading columns of $N$ and $N_- \in \mathbf{R}^{d_2 \times (d_1 - k_1)}$ consists of the remaining columns. Similarly, let $\mathbf{z} = [\mathbf{z}_+, \mathbf{z}_-]$, where $\mathbf{z}_+ \in \mathbf{R}^{n \times k_1}$ and $\mathbf{z}_- \in \mathbf{R}^{n \times (d_1 - k_1)}$. Let $\mathbf{Z} = [\mathbf{Z}_+, \mathbf{Z}_-]$, where $\mathbf{Z}_+ \in \mathbf{R}^{n \times k_1}$ and $\mathbf{Z}_- \in \mathbf{R}^{n \times (d_1 - k_1)}$. Finally, when we refer to estimated features of an individual instance produced from Step 1, we use $\hat{\mathbf{z}}_+$.

We have $\mathbf{Y} = \mathbf{Z}_+ N_+^\mathrm{T} + \mathbf{Z}_- N_-^\mathrm{T} + E$. We let

$$\delta_+ = \hat{\mathbf{z}}_+ - W^\mathrm{T}\mathbf{z}_+$$
$$\Delta_+ = \hat{\mathbf{Z}}_+ - \mathbf{Z}_+ W,$$

where $\|\delta_+\|_2 = O(\epsilon/\delta^3)$ and $\|\Delta_+\|_2 = O(\sqrt{n}\epsilon/\delta^3)$. We have

$$\frac{1}{n}\hat{\mathbf{Z}}_+^{\mathrm{T}}\mathbf{Y} = \frac{1}{n}(\Delta_+ + \mathbf{Z}_+ W)^{\mathrm{T}}(\mathbf{Z}_+ N_+^{\mathrm{T}} + \mathbf{Z}_- N_-^{\mathrm{T}} + E)$$

$$= W^{\mathrm{T}} N_+^{\mathrm{T}} + W^{\mathrm{T}}\left(\frac{1}{n}\mathbf{Z}_+^{\mathrm{T}}\mathbf{Z}_+ - I_{k_1 \times k_1}\right) N_+^{\mathrm{T}} + \frac{1}{n}W^{\mathrm{T}}\mathbf{Z}_+^{\mathrm{T}}\mathbf{Z}_- N_-^{\mathrm{T}} + \frac{1}{n}W^{\mathrm{T}}\mathbf{Z}_+^{\mathrm{T}}E$$

$$+ \frac{1}{n}\Delta_+^{\mathrm{T}}(\mathbf{Z}_+ N_+^{\mathrm{T}} + \mathbf{Z}_- N_-^{\mathrm{T}} + E).$$

We shall let

$$\hat{N}_+^{\mathrm{T}} = W^{\mathrm{T}} N^{\mathrm{T}} + \mathcal{E}, \tag{13}$$

where

$$\mathcal{E} = \mathcal{E}_1 + \mathcal{E}_2 + \mathcal{E}_3 + \mathcal{E}_4 + \mathcal{E}_5$$

$$\mathcal{E}_1 = W^{\mathrm{T}}\left(\frac{1}{n}\mathbf{Z}_+^{\mathrm{T}}\mathbf{Z}_+ - I_{k_1 \times k_1}\right) N_+^{\mathrm{T}}$$

$$\mathcal{E}_2 = \frac{1}{n}W^{\mathrm{T}}\mathbf{Z}_+^{\mathrm{T}}\mathbf{Z}_- N^{\mathrm{T}}$$

$$\mathcal{E}_3 = \frac{1}{n}W^{\mathrm{T}}\mathbf{Z}_+^{\mathrm{T}}E$$

$$\mathcal{E}_4 = \frac{1}{n}\Delta_+^{\mathrm{T}}E$$

$$\mathcal{E}_5 = \frac{1}{n}\Delta_+^{\mathrm{T}}(\mathbf{Z}_+ N_+^{\mathrm{T}} + \mathbf{Z}_- N_-^{\mathrm{T}}).$$

We next analyze each term. We aim to find bounds in either spectral norm or Frobenius norm. In some cases, it suffices to use $\|\mathcal{E}_i\|_2 \cdot \mathrm{rank}(\mathcal{E}_i)$ to upper bound $\|\mathcal{E}_i\|_F$. So we bound only $\mathcal{E}_i$'s spectral norm. On the other hand, in the case of analyzing $\mathcal{E}_5$, we can get a tight Frobenius norm bound but we cannot get a non-trivial spectral bound.

From time to time, we will label the dimension of matrices in complex multiplication operations to enable readers to do sanity checks.

**Bounding $\mathcal{E}_1$.** We use the following Lemmas.

**Lemma 7.** *Let $\mathbf{Z} \in \mathbf{R}^{n \times k_1}$, where $k_1 < n$. Let each entry of $\mathbf{Z}$ be an independent standard Gaussian. We have*

$$\left\|\frac{1}{n}\mathbf{Z}^{\mathrm{T}}\mathbf{Z} - I\right\| \leq \max\left\{\frac{10\log^2 n}{\sqrt{n}}, 4\sqrt{\frac{k_1}{n}}\right\} \tag{14}$$

*Proof of Lemma 7.* We rely on the Lemma [41]:

**Lemma 8.** *Let $S \in \mathbf{R}^{n \times k}$ ($n > k$) be a random matrix so that each $S_{i,j}$ is an independent standard Gaussian random variable. Let $\sigma_{\max}(S)$ be the maximum singular value of $S$ and $\sigma_{\min}(S)$ be the minimum singular value of it. We have*

$$\Pr[\sqrt{n} - \sqrt{k} - t \leq \sigma_{\min}(S) \leq \sigma_{\max}(S) \leq \sqrt{n} + \sqrt{k} + t] \geq 1 - 2 \times \exp(-t^2/2). \tag{15}$$

We set $t = \max\left\{\frac{\sqrt{k_1}}{10}, \log^2 n\right\}$. Let us start with considering the case $\frac{\sqrt{k_1}}{10} > \log^2 n$. We have

$$\sigma_{\min}(\mathbf{Z}^{\mathrm{T}}\mathbf{Z}) \geq n - 2.2\sqrt{nk_1} + 1.21k_1 \geq n - 2.2\sqrt{nk_1}. \tag{16}$$

and

$$\sigma_{\max}(\mathbf{Z}^{\mathrm{T}}\mathbf{Z}) \leq n + 2.2\sqrt{nk_1} + 1.21k_1 \leq n + 4\sqrt{nk_1}. \tag{17}$$

The case $\frac{\sqrt{k_1}}{10} \leq \log^2 n$ can be analyzed in a similar fashion so that we can get

$$\left\|\frac{1}{n}\mathbf{Z}^{\mathrm{T}}\mathbf{Z} - I\right\| \leq \max\left\{\frac{10\log^2 n}{\sqrt{n}}, 4\sqrt{\frac{k_1}{n}}\right\}. \tag{18}$$

$\square$

Therefore, we have

$$\|\mathcal{E}_1\|_2 \le \max\left\{\frac{10\log^2 n}{\sqrt{n}}, 4\sqrt{\frac{k_1}{n}}\right\}\|N_+^{\mathrm{T}}\|_2 = \Upsilon \max\left\{\frac{10\log^2 n}{\sqrt{n}}, 4\sqrt{\frac{k_1}{n}}\right\}.$$

**Bounding $\mathcal{E}_2$.** Observe that $\mathbb{E}[\|\mathbf{Z}_- N_-^{\mathrm{T}}\|_F^2] = n\|N_-^{\mathrm{T}}\|_F^2$. Also,

$$\mathbb{E}[\|\ \underbrace{\mathbf{Z}_+^{\mathrm{T}}}_{k_1\times n}\ \underbrace{\mathbf{Z}_-}_{n\times(d_1-k_1)}\ \underbrace{N_-^{\mathrm{T}}}_{(d_1-k_1)\times d_2}\ \|_F^2 \mid \mathbf{Z}_- N_-^{\mathrm{T}}] = k_1\|\mathbf{Z}_- N_-^{\mathrm{T}}\|_F^2.$$

Therefore,

$$\mathbb{E}[\mathbf{Z}_+^{\mathrm{T}}\mathbf{Z}_- N_-^{\mathrm{T}}] = k_1 n\|N_-^{\mathrm{T}}\|_F. \tag{19}$$

We next bound $\|N_-\|_F$.

**Lemma 9.** *Let $N$ be the learnable parameter in normalized form $N = [N_+, N_-]$, where $N_+ \in \mathbf{R}^{d_2\times k_1}$ and $N_- \in \mathbf{R}^{d_2\times(d_1-k_1)}$, and $k_1$ is determined by* STEP-1-PCA-X. *We have $\|N_-\|_F = O\left(\delta^{\frac{\omega-1}{\omega+1}}\right) = o(1)$.*

*Proof of Lemma 9.* Recall that

$$N = \underbrace{M}_{d_2\times d_1}\ \underbrace{V^*}_{d_1\times d_1}\ \underbrace{(\Lambda^*)^{\frac{1}{2}}}_{d_1\times d_1}.$$

We let $\Lambda^* = [\Lambda_+^*, \Lambda_-^*]$, where $\Lambda_+^* \in \mathbf{R}^{d_1\times k_1}$ and $\Lambda_-^* \in \mathbf{R}^{d_1\times(d_1-k_1)}$. We have $N_- = MV^*(\Lambda_-^*)^{\frac{1}{2}}$. Therefore,

$$\|N_-\|_F^2 \le \|M\|_2^2\|V^*\|_2^2 \left\|(\Lambda^*)^{\frac{1}{2}}\right\|_F^2 = O\left(\Upsilon\delta^{\frac{\omega-1}{\omega+1}}\right) = o(1). \tag{20}$$

Here, we used the assumption $\|M\|_2 = O(1)$ and the last equation holds because of Proposition 1. $\qquad\square$

By (19), (20), and a standard Chernoff bound, we have whp

$$\|\mathcal{E}_2\|_2 \le \|\mathcal{E}_2\|_F \le 2\sqrt{\frac{k_1}{n}}\|N_-\|_F = o\left(\sqrt{\frac{k_1}{n}}\right).$$

*Bounding $\mathcal{E}_3$.* We have the following Lemma.

**Lemma 10.** *Let $\mathbf{Z} \in \mathbf{R}^{n\times k_1}$ so that each entry in $\mathbf{Z}$ is an independent standard Gaussian and $E \in \mathbf{R}^{n\times d_2}$ so that each entry in $E$ is an independent Gaussian $N(0, \sigma_\epsilon^2)$. For sufficiently large $n$, $k_1$, and $d_2$, where $k_1 \le d_2$, we have*

$$\left\|\frac{1}{n}\mathbf{Z}^{\mathrm{T}}E\right\| \le \frac{1.1\sigma_\epsilon}{\sqrt{n}}(\sqrt{k_1} + \sqrt{d_2}).$$

*Proof of Lemma 10.* Let $t = \max\left\{\frac{10\log^2 n}{\sqrt{n}}, 4\sqrt{\frac{k_1}{n}}\right\}$. By Lemma 7, with high probability $\|\frac{1}{n}\mathbf{Z}^{\mathrm{T}}\mathbf{Z} - I\| \le t$. This implies that the eigenvalues of $\mathbf{Z}^{\mathrm{T}}\mathbf{Z}$ are all within the range $n(1 \pm t)$. Note that for $0 < \eta < 1/3$, if $\xi \in [1-\eta, 1+\eta]$, then $\sqrt{\xi} \in [1-2\eta, 1+2\eta]$. This implies that the singular values of $\mathbf{Z}$ are within the range $\sqrt{n}(1 \pm 2t)$.

Let $\Sigma^Z/\sqrt{n} = I + \Delta^Z$, where $\|\Delta^Z\| \le 2t$. We have

$$\frac{1}{n}\mathbf{Z}^{\mathrm{T}}E = V^Z\left(\frac{\Sigma^Z}{\sqrt{n}}\right)(U^Z)^{\mathrm{T}}\frac{E}{\sqrt{n}} = V^Z(I + \Delta^Z)(U^Z)^{\mathrm{T}}\frac{E}{\sqrt{n}}$$

$$= \underbrace{V^Z}_{k_1\times k_1}\underbrace{(U^Z)^{\mathrm{T}}}_{k_1\times n}\underbrace{\frac{E}{\sqrt{n}}}_{n\times d_2} + V^Z\Delta^Z(U^Z)^{\mathrm{T}}\frac{E}{\sqrt{n}}. \tag{21}$$

Using the fact that the columns of $U^Z$ are orthonormal vectors, $V^Z$ is a unitary matrix, and $k_1 \leq d_2$, we see that $V^Z(U^Z)^{\mathrm{T}}E/\sqrt{n}$ is a matrix with i.i.d. Gaussian entries with standard deviation $\sigma_\epsilon/\sqrt{n}$.

Let $B = V^Z(U^Z)^{\mathrm{T}}E/\sigma_\epsilon$ and $\tilde{B} = (U^Z)^{\mathrm{T}}E/\sigma_\epsilon$. Then, from (21), we have

$$\frac{1}{n}\mathbf{Z}^{\mathrm{T}}E = \frac{\sigma_\epsilon}{\sqrt{n}}\left(B + V^Z\Delta^Z\tilde{B}\right). \tag{22}$$

The entries in $B$ ($\tilde{B}$) are all i.i.d Gaussian. By Marchenko-Pastar's law (and the finite sample bound of it [41]), we have with high probability $\|\tilde{B}\|, \|B\| = \sqrt{k_1} + \sqrt{d_2} + o(\sqrt{k_1} + \sqrt{d_2})$. Therefore, with high probability:

$$\left\|\frac{1}{n}\mathbf{Z}^{\mathrm{T}}E\right\|_2 \leq \frac{1.1\sigma_\epsilon}{\sqrt{n}}(\sqrt{k_1} + \sqrt{d_2}).$$

$\square$

Lemma 10 implies that

$$\|\mathcal{E}_3\|_2 \leq \frac{1.1\sigma_\epsilon}{\sqrt{n}}(\sqrt{k_1} + \sqrt{d_2}).$$

**Bounding $\mathcal{E}_4$.** We have

$$\mathbb{E}[\|\mathcal{E}_4\|_F^2] = \frac{1}{n^2}\mathbb{E}[\|\Delta_+^{\mathrm{T}}E\|_F^2] = \frac{d_2}{n}\|\Delta_+\|_F^2 = O\left(\frac{d_2\epsilon}{n\delta^3}\right) = o\left(\frac{d_2}{n}\right).$$

Using a Chernoff bound, we have whp $\|\mathcal{E}_4\|_F = o\left(\frac{d_2}{n}\right)$.

**Bounding $\mathcal{E}_5$.** Because $\mathcal{E}_5 = \frac{1}{n}\Delta_+^{\mathrm{T}}(\mathbf{Z}N^{\mathrm{T}})$, we have

$$\|\mathcal{E}_5\|_F^2 \leq \frac{1}{n^2}\|\Delta_+^{\mathrm{T}}\|_2^2\|\mathbf{Z}N^{\mathrm{T}}\|_F^2.$$

Using a simple Chernoff bound, we have whp,

$$\|\mathbf{Z}N^{\mathrm{T}}\|_F^2 \leq 2n\|M\mathbf{x}\|_2^2 \leq 2n\|M\|_2^2\|\mathbf{x}\|_2^2 \leq 2\Upsilon n.$$

This implies $\|\mathcal{E}_5\|_F^2 \leq O\left(\frac{\epsilon^2}{n^2\delta^6}n^2\Upsilon^2\right) = O\left(\frac{\epsilon^2}{\delta^6}\right)$.

We may let

$$\mathcal{E}_L = \mathcal{E}_1 + \mathcal{E}_2 + \mathcal{E}_3 + \mathcal{E}_4$$
$$\mathcal{E}_T = \mathcal{E}_5.$$

We can check that

$$\|\mathcal{E}_L\|_2 \leq \|\mathcal{E}_1\|_2 + \|\mathcal{E}_2\|_2 + \|\mathcal{E}_3\|_2 + \|\mathcal{E}_4\|_2$$
$$\leq \Upsilon\max\left\{\frac{10\log^2 n}{\sqrt{n}}, 4\frac{k_1}{n}\right\} + o\left(\sqrt{\frac{k_1}{n}}\right) + \frac{1.1\sigma_\epsilon}{\sqrt{n}}(\sqrt{k_1} + \sqrt{d_2}) + o\left(\frac{d_2}{n}\right)$$
$$\leq 2.2\sigma_{\sigma_\epsilon}\sqrt{\frac{d_2}{n}}$$

Also, we can see that $\|\mathcal{E}_T\|_F = \|\mathcal{E}_5\|_F = O(\epsilon/\delta^3)$. This completes the proof for Proposition 2.

### 3.2.3 Step 3. Analysis of our algorithm's MSE

Let us recall our notation:

1. $\mathbf{z} = (\Lambda^*)^{-\frac{1}{2}}(V^*)^{\mathrm{T}}\mathbf{x}$ and $\delta_+ = \hat{\mathbf{z}}_+ - W^{\mathrm{T}}\mathbf{z}_+$.

2. We let $\hat{N}_+^{\mathrm{T}} = \hat{\mathbf{Z}}_+^{\mathrm{T}}\mathbf{Y}$ be the output of STEP-1-PCA-X in Fig. 1.

3. All singular vectors in $\hat{N}_+$ whose associated singular values $\geq \theta\sigma_\epsilon\sqrt{\frac{d_2}{n}}$ are kept.

Let $\ell$ be the largest index such that $\sigma_\ell^{N_+} \geq \theta\sigma_\epsilon\sqrt{\frac{d_2}{n}}$. One can see that our testing forecast is $\mathbf{P}_{k_2}(\hat{N}_+)\hat{\mathbf{z}}_+$. Therefore, we need to bound $\mathbb{E}_{\mathbf{z}}[\|\mathbf{P}_{k_2}(\hat{N}_+)\hat{\mathbf{z}}_+ - N\mathbf{z}\|^2]$.

By Proposition 2, we have $\hat{N}_+ = (W^{\mathrm{T}}N_+^{\mathrm{T}} + \mathcal{E}_L + \mathcal{E}_T)^{\mathrm{T}}$, where $\|\mathcal{E}_L\|_2 \leq 2.2\sigma_\epsilon\sqrt{\frac{d_2}{n}}$ and $\|\mathcal{E}_T\|_F = O(\xi/\delta^3)$ whp. Let $\mathcal{E} \triangleq \mathcal{E}_L + \mathcal{E}_T$. We have

$$\mathbf{P}_{k_2}(N_+W + \mathcal{E}^{\mathrm{T}})\hat{\mathbf{z}}_+ = \mathbf{P}_{k_2}(N_+W + \mathcal{E}^{\mathrm{T}})(W^{\mathrm{T}}\mathbf{z}_+ + \delta_+)$$
$$= \mathbf{P}_{k_2}(N_+W + \mathcal{E}^{\mathrm{T}})W^{\mathrm{T}}W(W^{\mathrm{T}}\mathbf{z}_+ + \delta_+)$$
$$= \mathbf{P}_{k_2}((\underbrace{N_+}_{d_2\times k_1}\underbrace{W}_{k_1\times k_1} + \underbrace{\mathcal{E}^{\mathrm{T}}}_{d_2\times k_1})\underbrace{W^{\mathrm{T}}}_{k_1\times k_1})(\underbrace{WW^{\mathrm{T}}}_{k_1\times k_1}\underbrace{\mathbf{z}_+}_{k_1\times 1} + \underbrace{W}_{k_1\times k_1}\underbrace{\delta_+}_{k_1\times 1})$$
$$= \mathbf{P}_{k_2}\left(N_+ + (W\mathcal{E})^{\mathrm{T}}\right)(\mathbf{z}_+ + W\delta_+).$$

Let $\mathcal{E}' = (W\mathcal{E})^{\mathrm{T}}$, $\mathcal{E}'_L = (W\mathcal{E}_L)^{\mathrm{T}}$, $\mathcal{E}'_T = (W\mathcal{E}_T)^{\mathrm{T}}$, and $\delta'_+ = W\delta_+$. We still have $\|\mathcal{E}'_L\|_2 \leq 2.2\sigma_\epsilon\sqrt{\frac{d_2}{n}}$, and $\|\mathcal{E}'_T\|_F = O(\epsilon/\delta^3)$.

We next have

$$\mathbb{E}_{\mathbf{z}}\left[\|\mathbf{P}_{k_2}(\hat{N}_+)\hat{\mathbf{z}}_+ - N\mathbf{z}\|_2^2\right]$$
$$= \mathbb{E}_{\mathbf{z}}\left[\|(\mathbf{P}_{k_2}(N_+ + \mathcal{E}')\mathbf{z}_+ - N_+\mathbf{z}_+) + \mathbf{P}_{k_2}(N_+ + \mathcal{E}')\delta'_+ - N_-\mathbf{z}_-\|_2^2\right]$$
$$\leq \underbrace{\mathbb{E}_{\mathbf{z}}\left[\|(\mathbf{P}_{k_2}(N_+ + \mathcal{E}')\mathbf{z}_+ - N_+\mathbf{z}_+)\|_2^2\right]}_{\triangleq \Phi_1} + \underbrace{\mathbb{E}_{\mathbf{z}}\left[\|\mathbf{P}_{k_2}(N_+ + \mathcal{E}')\delta'_+ - N_-\mathbf{z}_-\|_2^2\right]}_{\triangleq \Phi_2}$$
$$+ 2\sqrt{\mathbb{E}_{\mathbf{z}}\left[\|(\mathbf{P}_{k_2}(N_+ + \mathcal{E}')\mathbf{z}_+ - N_+\mathbf{z}_+)\|_2^2\right] \cdot \mathbb{E}_{\mathbf{z}}\left[\|\mathbf{P}_{k_2}(N_+ + \mathcal{E}')\delta'_+ - N_-\mathbf{z}_-\|_2^2\right]}$$

(Cauchy Schwarz for random variables)

$$= \Phi_1 + \Phi_2 + 2\sqrt{\Phi_1\Phi_2}.$$

We first bound $\Phi_2$ (the easier term). We have

$$\Phi_2 = \mathbb{E}_{\mathbf{z}}\left[\|\mathbf{P}_{k_2}(N_+ + \mathcal{E}')\delta'_+ - N_-\mathbf{z}_-\|_2^2\right]$$
$$\leq 2\mathbb{E}_{\mathbf{z}}\left[\|\mathbf{P}_{k_2}(N_+ + \mathcal{E}')\delta'_+\|_2^2\right] + 2\mathbb{E}\left[\|N_-\mathbf{z}_-\|_2^2\right]$$

We first bound $\mathbb{E}_{\mathbf{z}}\left[\|\mathbf{P}_{k_2}(N_+ + \mathcal{E}')\delta'_+\|_2^2\right]$. We consider two cases.

*Case 1.* $\sigma_{\max}(N_+) > \frac{\theta}{2}\sigma_\epsilon\sqrt{\frac{d_2}{n}}$. In this case, we observe that $\|\mathcal{E}\|_2 \leq 2.2\sigma_\epsilon\sqrt{\frac{d_2}{n}} + o(1)$. This implies that $\|N_+ + \mathcal{E}'\|_2 = O(\|N_+\|_2) = O(1)$. Therefore, $\mathbb{E}_{\mathbf{z}}\left[\|\mathbf{P}_{k_2}(N_+ + \mathcal{E}')\delta'_+\|_2^2\right] \leq \|(N_+ + \mathcal{E}')\delta'_+\|_2^2 = O(\|\delta'_+\|_2^2)$.

*Case 2.* $\sigma_{\max}(N_+) \leq \frac{\theta}{2}\sigma_\epsilon\sqrt{\frac{d_2}{n}}$. In this case, $\|N_+ + \mathcal{E}'\|_2 \leq \theta\sigma\sqrt{\frac{d_2}{n}}$. This implies $\mathbf{P}_{k_2}(N_+ + \mathcal{E}')\delta'_+ = 0$ (i.e., the projection $\mathbf{P}_{k_2}(\cdot)$ will not keep any subspace).

This case also implies $\mathbb{E}_{\mathbf{z}}\left[\left\|\mathbf{P}_{k_2}(N_+ + \mathcal{E}')\delta'_+\right\|_2^2\right] = 0 = O(\|\delta'_+\|_2^2)$

Next, we have $\mathbb{E}[\|N_-\mathbf{z}_-\|_2^2] = \|N_-\|_F^2 = O\left(\delta^{\frac{\omega-1}{\omega+1}}\right)$.

Therefore,

$$\Phi_2 = O\left(\frac{\epsilon^2}{\delta^6} + \delta^{\frac{\omega-1}{\omega+1}}\right).$$

Next, we move to bound

$$\mathbb{E}_{\mathbf{z}}\left[\left\|(\mathbf{P}_{k_2}(N_+ + \mathcal{E}')\mathbf{z}_+ - N_+\mathbf{z}_+)\right\|_2^2\right].$$

We shall construct an orthonormal basis on $\mathbf{R}^{d_2}$ and use the basis to "measure the mass". Let us describe this simple idea at high-level first. Let $\mathbf{v}_1, \mathbf{v}_2, \cdots, \mathbf{v}_{d_2}$ be a basis for $\mathbf{R}^{d_2}$ and let $A \in \mathbf{R}^{d_2 \times k_1}$ be an arbitrary matrix. We have $\|A\|_F^2 = \sum_{i \leq d_2} \|\mathbf{v}_i^{\mathrm{T}} A\|_2^2$. The meaning of this equality is that we may apply a change of basis on the columns of $\overline{A}$ and the "total mass" of $A$ should remain unchanged after the basis change. Our orthonormal basis consists of three groups of vectors.

*Group 1.* $\{U_{:,i}^{N_+}\}$ for $i \leq \ell$, where $\ell$ is the number of $\sigma_i(N^+)$ such that $\sigma_i(N^+) \geq \theta\sigma_\epsilon\sqrt{\frac{d_2}{n}}$.

*Group 2.* The subspace in $\mathrm{Span}(\{U_{:,i}^{\hat{N}}\}_{i \leq k_2})$ that is orthogonal to $\{U_{:,i}^{N_+}\}_{i \leq \ell}$. Let us refer to these vectors as $\hat{\mathbf{u}}_1, \ldots, \hat{\mathbf{u}}_s$ and $\hat{U}_{[s]} = [\hat{\mathbf{u}}_1, \ldots \hat{\mathbf{u}}_s]$.

*Group 3.* An arbitrary basis that is orthogonal to vectors in group 1 and group 2. Let us refer to them as $\mathbf{r}_1, \ldots, \mathbf{r}_t$.

We have

$$\|\mathbf{P}_{k_2}(N_+ + \mathcal{E}') - N_+\|_F^2$$

$$= \sum_{i \leq \ell}\left\|\left(U_{:,i}^{N_+}\right)^{\mathrm{T}}(\mathbf{P}_{k_2}(N_+ + \mathcal{E}') - N_+)\right\|_2^2 \qquad \text{Term 1}$$

$$+ \sum_{i \leq s}\|\hat{\mathbf{u}}_i^{\mathrm{T}}(\mathbf{P}_{k_2}(N_+ + \mathcal{E}') - N_+)\|_2^2 \qquad \text{Term 2}$$

$$+ \sum_{i \leq r}\|\mathbf{r}_i^{\mathrm{T}}(\mathbf{P}_{k_2}(N_+ + \mathcal{E}') - N_+)\|_2^2 \qquad \text{Term 3}$$

To understand the reason we perform such grouping, we can imagine making a decision for an (overly) simplified problem for each direction in the basis: consider a univariate estimation problem $y = \mu + \epsilon$ with $\mu$ being the signal, $\epsilon \sim N(0, \sigma^2)$ being the noise, and $y$ being the observation. Let us consider the case we observe only one sample. Now when $y \gg \sigma$, we can use $y$ as the estimator and $\mathbb{E}[(y - \mu)^2] = \sigma^2$. This high signal-to-noise setting corresponds to the vectors in Group 1.

When $y \approx 3\sigma$, we have $\mu^2 = \mathbb{E}[(y-\epsilon)^2] \approx (3-1)^2\sigma^2 = 4\sigma^2$. On the other hand, $\mathbb{E}[(y-\mu)^2] = \sigma^2$. This means if we use $y$ as the estimator, the forecast is at least better than trivial. The median signal-to-noise setting corresponds to the vectors in group 2.

When $y \ll \sigma$, we can simply use $\hat{y} = 0$ as the estimator. This low signal-to-noise setting corresponds to vectors in group 3.

In other words, we expect: *(i)* In term 1, signals along each direction of vectors in group 1 can be extracted. Each direction also pays a $\sigma^2$ term, which in our setting corresponds to $\theta\sigma_\epsilon\sqrt{\frac{d_2}{n}}$. Therefore, the MSE can be bounded by $O(\ell\theta^2\sigma_\epsilon^2 d_2/n)$. *(ii)* In terms 2 and 3, we do at least (almost) as well as the "trivial forecast" ($\hat{\mathbf{y}} = 0$). There is also an error produced by the estimator error from $\hat{\mathbf{z}}_+$, and the tail error produced from cutting out features in STEP-1-PCA-X in Fig. 1.

Now we proceed to execute this idea.

**Term 1.** $\sum_{i \le \ell} \left\| \left(U_{:,i}^{N_+}\right)^{\mathrm{T}} \left(\mathbf{P}_{k_2}(N_+ + \mathcal{E}') - N_+\right) \right\|_2^2$. Let $\hat{U} \in \mathbf{R}^{d_2 \times d_2}$ be the left singular vector of $N_+ + \mathcal{E}'$. We let $\hat{U}$ have $d_2$ columns to include those vectors whose corresponding singular values are 0 for the ease of calculation. We have

$$
\sum_{i \le \ell} \left\| \left(U_{:,i}^{N_+}\right)^{\mathrm{T}} \left(\mathbf{P}_{k_2}(N_+ + \mathcal{E}') - N_+\right) \right\|_2^2
$$

$$
= \sum_{i \le \ell} \left\| \left(U_{:,i}^{N_+}\right)^{\mathrm{T}} \left(\hat{U}_{:,1:k_2} \hat{U}_{:,1:k_2}^{\mathrm{T}} (N_+ + \mathcal{E}') - N_+\right) \right\|_2^2
$$

$$
= \sum_{i \le \ell} \left\| \left(U_{:,i}^{N_+}\right)^{\mathrm{T}} \left(\left(\hat{U}\hat{U}^{\mathrm{T}} - \hat{U}_{:,k_2+1:d_2} \hat{U}_{:,k_2+1:d_2}^{\mathrm{T}}\right)(N_+ + \mathcal{E}') - N_+\right) \right\|_2^2
$$

$$
= \sum_{i \le \ell} \left\| \left(U_{:,i}^{N_+}\right)^{\mathrm{T}} \left(\mathcal{E}' - \hat{U}_{:,k_2+1:d_2} \hat{U}_{:,k_2+1:d_2}^{\mathrm{T}}(N_+ + \mathcal{E}')\right) \right\|_2^2
$$

$$
\le 2 \left\{ \sum_{i \le t} \left\| \left(U_{:,i}^{N_+}\right)^{\mathrm{T}} \mathcal{E}' \right\|_2^2 + \sum_{i \le \ell} \left\| \left(U_{:,i}^{N_+}\right)^{\mathrm{T}} \hat{U}_{:,k_2+1:d_2} \hat{U}_{:,k_2+1:d_2}^{\mathrm{T}}(N_+ + \mathcal{E}') \right\|_2^2 \right\}
$$

$$
\le O \left( \ell \|\mathcal{E}_L'\|_2^2 + \|\mathcal{E}_T'\|_F^2 + \sum_{i \le \ell} \left\| \left(U_{:,i}^{N_+}\right)^{\mathrm{T}} \right\|_2^2 \left\| \hat{U}_{:,k_2+1:d_2} \right\|_2^2 \underbrace{\left\| \hat{U}_{:,k_2+1:d_2}^{\mathrm{T}}(N_+ + \mathcal{E}') \right\|_2^2}_{\le \frac{\theta^2 \sigma_\epsilon^2 d_2}{n} \text{ by the definition of } k_2.} \right)
$$

$$
= O \left( \ell \|\mathcal{E}_L'\|_2^2 + \|\mathcal{E}_T'\|_F^2 + \frac{\ell d_2 \theta^2 \sigma_\epsilon^2}{n} \right)
$$

$$
= O \left( \frac{\ell d_2 \theta^2 \sigma_\epsilon^2}{n} + + \|\mathcal{E}_T'\|_F^2 \right)
$$

**Term 2.** $\sum_{i \le s} \|\hat{\mathbf{u}}_i^{\mathrm{T}} \left(\mathbf{P}_{k_2}(N_+ + \mathcal{E}') - N_+\right)\|_2^2$. We have

$$
\sum_{i \le s} \|\hat{\mathbf{u}}_i^{\mathrm{T}} \left(\mathbf{P}_{k_2}(N_+ + \mathcal{E}') - N_+\right)\|_2^2 = \sum_{i \le s} \|\hat{\mathbf{u}}_i^{\mathrm{T}} \left(\mathbf{P}_{k_2}(N_+ + \mathcal{E}') - (N_+ + \mathcal{E}') + \mathcal{E}'\right)\|_2^2
$$

$$
= \sum_{i \le s} \|\hat{\mathbf{u}}_i^{\mathrm{T}} \mathcal{E}'\|_2^2
$$

On the other hand, note that

$$
\sum_{i \le s} \|\hat{\mathbf{u}}_i^{\mathrm{T}} N_+\|_2^2
$$

$$
= \sum_{i \le s} \left\| \hat{\mathbf{u}}_i^{\mathrm{T}}(N_+ + \mathcal{E}') - \hat{\mathbf{u}}_i^{\mathrm{T}} \mathcal{E}' \right\|_2^2
$$

$$
= \sum_{i \le s} \left( \left\| \hat{\mathbf{u}}_i^{\mathrm{T}}(N_+ + \mathcal{E}') \right\|_2^2 + \|\hat{\mathbf{u}}_i^{\mathrm{T}} \mathcal{E}'\|_2^2 - 2 \left\langle \hat{\mathbf{u}}_i^{\mathrm{T}}(N_+ + \mathcal{E}'), \hat{\mathbf{u}}_i^{\mathrm{T}}(\mathcal{E}_L' + \mathcal{E}_T') \right\rangle \right)
$$

$$
= \sum_{i \le s} \left( \left\| \hat{\mathbf{u}}_i^{\mathrm{T}}(N_+ + \mathcal{E}') \right\|_2^2 - 2 \langle \hat{\mathbf{u}}_i^{\mathrm{T}}(N_+ + \mathcal{E}'), \hat{\mathbf{u}}_i^{\mathrm{T}} \mathcal{E}_L' \rangle \right) + \underbrace{\sum_{i \le s} \|\hat{\mathbf{u}}_i^{\mathrm{T}} \mathcal{E}'\|_2^2}_{= \text{ Term 2.}} - 2 \sum_{i \le s} \langle \hat{\mathbf{u}}_i^{\mathrm{T}}(N_+ + \mathcal{E}'), \hat{\mathbf{u}}_i^{\mathrm{T}} \mathcal{E}_T' \rangle
$$

$$
\tag{23}
$$

Note that

$$\sum_{i \le s} \left( \left\| \hat{\mathbf{u}}_i^{\mathrm{T}}(N_+ + \mathcal{E}') \right\|_2^2 - 2\langle \hat{\mathbf{u}}_i^{\mathrm{T}}(N_+ + \mathcal{E}'), \hat{\mathbf{u}}_i^{\mathrm{T}} \mathcal{E}_L' \rangle \right)$$

$$\ge \sum_{i \le s} \| \hat{\mathbf{u}}_i^{\mathrm{T}}(N_+ + \mathcal{E}') \|_2 \big( \underbrace{\| \hat{\mathbf{u}}_i^{\mathrm{T}}(N_+ + \mathcal{E}') \|_2}_{\ge \theta \sigma_\epsilon \sqrt{\frac{d_2}{n}}} - 2 \underbrace{\| \hat{\mathbf{u}}_i^{\mathrm{T}} \mathcal{E}_L' \|_2}_{\le 2.2\sigma_\epsilon \sqrt{\frac{d_2}{n}}} \big)$$

$$\ge 0 \text{ (using the fact that } \theta \text{ is sufficiently large).} \tag{24}$$

Next, we examine the term $-2\sum_{i \le s}\langle \hat{\mathbf{u}}_i^{\mathrm{T}}(N_+ + \mathcal{E}'), \hat{\mathbf{u}}_i^{\mathrm{T}} \mathcal{E}_T' \rangle$.

$$-2\sum_{i \le s}\langle \hat{\mathbf{u}}_i^{\mathrm{T}}(N_+ + \mathcal{E}'), \hat{\mathbf{u}}_i^{\mathrm{T}} \mathcal{E}_T' \rangle$$

$$= -2\langle \hat{U}_{[s]}^{\mathrm{T}}(N_+ + \mathcal{E}'), \hat{U}_{[s]}^{\mathrm{T}} \mathcal{E}_T' \rangle$$

$$= -2\mathrm{Tr}\left( \hat{U}_{[s]}^{\mathrm{T}} \mathcal{E}_T' (N_+ + \mathcal{E}')^{\mathrm{T}} \hat{U}_{[s]} \right)$$

$$\ge -2 \left\| \mathrm{Tr}(\mathcal{E}_T'(N_+ + \mathcal{E}')^{\mathrm{T}} \right\|_2 \left\| \hat{U}_{[s]}^{\mathrm{T}} \hat{U}_{[s]} \right\|_2$$

$$= -2 \left| \langle (\mathcal{E}_T')^{\mathrm{T}}, N_+ + \mathcal{E}' \rangle \right|$$

$$\ge -2\|\mathcal{E}_T'\|_F \|N_+ + \mathcal{E}'\|_F \qquad \text{(Cauchy Schwarz)}$$

$$\ge -2\|\mathcal{E}_T'\|_F (\|N_+\|_F + \|\mathcal{E}'\|_F))$$

$$\ge -O(\|\mathcal{E}_T'\|_F) \qquad (\|N_+\|_F^2 \le \|N\|_F^2 = \mathbb{E}[\|Nz\|^2] = \mathbb{E}[\|Mx\|^2] = O(1))$$
$$\tag{25}$$

(23), (24), and (25) imply that

$$\sum_{i \le s} \|\mathbf{P}_{k_2}(N_+ + \mathcal{E}') - N_+\|_2^2 \le \sum_{i \le s} \left\| \hat{U}^{\mathrm{T}} N_+ \right\|_2^2 + O(\|\mathcal{E}_T'\|_F).$$

**Term 3.** We have

$$\sum_{i \le r} \|\mathbf{r}_i^{\mathrm{T}} \left( \mathbf{P}_{k_2}(N_+ + \mathcal{E}') - N_+ \right) \|_2^2 = \sum_{i \le r} \|\mathbf{r}_i^{\mathrm{T}} N_+\|_2^2.$$

This is because $\mathbf{r}_i$'s are orthogonal to the first $k_2$ left singular vectors of $N_+ + \mathcal{E}'$.

We sum together all the terms:

$$\|\mathbf{P}_{k_2}(N_+ + \mathcal{E}') - N_+\|_F^2 \le O\left( \frac{\ell d_2 \theta^2 \sigma_\epsilon^2}{n} \right) + \underbrace{\sum_{i \le s} \left\| \hat{U}_i^{\mathrm{T}} N_+ \right\|_2^2}_{(*)} + \underbrace{\sum_{i \le t} \|\mathbf{r}_i^{\mathrm{T}} N_+\|_2^2}_{(**)} + O\left( \|\mathcal{E}_T'\|_F \right).$$

$$= O\left( \frac{\ell d_2 \theta^2 \sigma_\epsilon^2}{n} \right) + \|N_+\|_F^2 - \sum_{i \le \ell} \left( \sigma_i^{N_+} \right)^2 + O\left( \|\mathcal{E}_T'\|_F \right) \tag{26}$$

In the above analysis, we used $\mathrm{Span}(\{\hat{\mathbf{u}}_i\}_{i \le s}, \{\mathbf{r}_i\}_{i \le t})$ is orthogonal to $\mathrm{Span}(\{U_{:,i}^{N_+}\}_{i \le \ell})$. Therefore, we can collect (*) and (**) and obtain

$$\sum_{i \le s} \left\| \hat{U}_i^{\mathrm{T}} N_+ \right\|_2^2 + \sum_{i \le t} \|\mathbf{r}_i^{\mathrm{T}} N_+\|_2^2 = \|N_+\|_F^2 - \sum_{i \le \ell} (\sigma_i^{N_+})^2.$$

Now the MSE is in terms of $\sigma_i^{N_+}$. We aim to bound the MSE in $\sigma_i^N$. So we next relate $\sum_{i \le \ell}(\sigma_i^{N_+})^2$ with $\sum_{i \le \ell}(\sigma_i^N)^2$. Recall that $\tilde{N}_+ = [N_+, \mathbf{0}]$, where $\mathbf{0} \in \mathbf{R}^{d_2 \times (d_1 - k_1)}$. The singular values of $\tilde{N}_+$ are the same as those of $N_+$. By using a standard matrix perturbation result, we have

$$\sum_{i \le \ell} \left( \sigma_i^{N_+} - \sigma_i^N \right)^2 = \sum_{i \le \ell} \left( \sigma_i^{\tilde{N}_+} - \sigma_i^N \right)^2 \le \|N_-\|_F^2 = c\delta^{\frac{\omega-1}{\omega+1}} \tag{27}$$

for some constant $c$. We may think (27) as a constraint and maximize the difference $\sum_{i\leq\ell}(\sigma_i^N)^2 - \sum_{i\leq\ell}(\sigma_i^{N+})^2$. This is maximized when $\sigma_1^N = \sigma_1^{N+} + \sqrt{c\delta^{\frac{\omega-1}{\omega+1}}}$ and $\sigma_i^N = \sigma_i^{N+}$ for $i > 1$.

Therefore,

$$\sum_{i\leq\ell}\left(\sigma_i^N\right)^2 \leq \sum_{1\leq i\leq\ell}\left(\sigma_i^{N+}\right)^2 + \left(\sigma_1^{N+} + \sqrt{c\delta^{\frac{\omega-1}{\omega+1}}}\right)^2$$

$$= \sum_{1\leq i\leq\ell}\left(\sigma_i^{N+}\right)^2 + O\left(\sqrt{\delta^{\frac{\omega-1}{\omega+1}}}\right). \tag{28}$$

Now (26) becomes

$$(26) \leq \|N_+\|_F^2 - \sum_{i\leq\ell}(\sigma_i^N)^2 + O\left(\frac{\ell d_2\theta^2\sigma_\epsilon^2}{n}\right) + O(\xi/\delta^3) + O\left(\sqrt{\delta^{\frac{\omega-1}{\omega+1}}}\right). \tag{29}$$

Next, we assert that $\ell \geq \ell^*$. Recall that $\|\sigma_i^{N+} - \sigma_i^N\|_2^2 = \|N_-\|_F^2 = o(1)$. This implies $\sigma_i^{N+} > \theta\sigma_\epsilon\sqrt{\frac{d_2}{n}}$ for $i \leq \ell^*$, i.e., $\ell \geq \ell^*$. So we have

$$\Phi_1 = \|\mathbf{P}_{k_2}(N_+ + \mathcal{E}') - N_+\|_F^2 \leq \|N\|_F^2 - \sum_{i\leq\ell^*}(\sigma_i^N)^2 + O\left(\frac{\ell^* d_2\theta^2\sigma_\epsilon^2}{n}\right) + O(\xi/\delta^3) + O\left(\sqrt{\delta^{\frac{\omega-1}{\omega+1}}}\right). \tag{30}$$

Finally, we obtain the bound for $\Phi_1 + \Phi_2 + 2\sqrt{\Phi_1\Phi_2}$. Note that

$$\frac{\Phi_2}{\Phi_1} \leq \frac{O\left(\frac{\xi^2}{\delta^6} + \delta^{\frac{\omega-1}{\omega+1}}\right)}{O\left(\frac{\xi}{\delta^3} + \sqrt{\delta^{\frac{\omega-1}{\omega+1}}}\right)} \leq \min\left\{\frac{\xi}{\delta^3}, \sqrt{\delta^{\frac{\omega-1}{\omega+1}}}\right\}(= o(1)).$$

We have

$$\Phi_1 + \Phi_2 + 2\sqrt{\Phi_1\Phi_2}$$

$$= \Phi_1\left(1 + 2\sqrt{\frac{\Phi_2}{\Phi_1}}\right) + \Phi_2$$

$$= \left[\|N\|_F^2 - \sum_{i\leq\ell^*}(\sigma_i^N)^2 + O\left(\frac{\ell^* d_2\theta^2\sigma_\epsilon^2}{n}\right) + O(\xi/\delta^3) + O\left(\sqrt{\delta^{\frac{\omega-1}{\omega+1}}}\right)\right]$$

$$\times \left\{1 + \min\left\{\sqrt{\frac{\xi}{\delta^3}} + \left(\delta^{\frac{\omega-1}{\omega+1}}\right)^{\frac{1}{4}}\right\}\right\} + O\left(\

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

## 4.1 Roadmap

This section describes the roadmap for executing the above idea.

**Normalized form.** Recall that $d_2 \leq d_1$, we have

$$M\mathbf{x} = \underbrace{U^M}_{d_2 \times d_2} \underbrace{\Sigma^M}_{d_2 \times d_2} \underbrace{(V^M)^{\mathrm{T}}}_{d_2 \times d_1} \underbrace{V^*}_{d_1 \times d_1} \underbrace{(\Lambda^*)^{\frac{1}{2}}}_{d_1 \times d_1} \underbrace{\mathbf{z}}_{d_1 \times 1} \tag{34}$$

We may perform an SVD on $\Sigma^M (V^M)^{\mathrm{T}} V^* (\Lambda^*)^{\frac{1}{2}} = \underbrace{A}_{d_2 \times d_2} \underbrace{L^\dagger}_{d_2 \times d_2} \underbrace{B^{\mathrm{T}}}_{d_2 \times d_1}$. We may also set $\mathbf{z}^\dagger = B^{\mathrm{T}}\mathbf{z}$,

which is a standard multi-variate Gaussian in $\mathbf{R}^{d_2}$. Then we have

$$M\mathbf{x} = U^M A L^\dagger B^{\mathrm{T}} \mathbf{z} = (U^M A) L^\dagger \mathbf{z}^\dagger. \tag{35}$$

Let $N^\dagger = (U^M A)L^\dagger$. The SVD of $N^\dagger$ is exactly $(U^M)AL^\dagger I_{d_2 \times d_2}$ because $U^M A$ is unitary. The *normalized form* of our problem is

$$\mathbf{y} = N^\dagger \mathbf{z}^\dagger + \epsilon. \tag{36}$$

Recall our local minimax has an oracle access interpretation. An algorithm with the oracle can reduce a problem into the normalized form on its own, and the algorithm knows $L^\dagger$. But the oracle still does not have any information on $N^\dagger$'s left singular vectors because being able to manipulate $U^M$ is the same as being able to manipulate $U^M A$. Therefore, we can analyze the lower bound in normalized form, with the assumption that the SVD of $N^\dagger = U^\dagger L^\dagger I_{d_2 \times d_2}$, in which $L^\dagger$ is known to the algorithm. We shall also let $\sigma_i^\dagger = L_{i,i}^\dagger = \sigma_i^{N^\dagger}$. Because $N^\dagger$ is square, we let $d = d_2$ in the analysis below.

We make two remarks in comparison to the orthogonalized form $\mathbf{y} = N\mathbf{z} + \epsilon$. (i) $\mathbf{z}^\dagger \in \mathbf{R}^{d_2}$, whereas $\mathbf{z} \in \mathbf{R}^{d_1}$. $\mathbf{z}^\dagger$'s dimension is smaller because the knowledge of $\Sigma^M$ and $V^M$ enable us to remove the directions from $\mathbf{x}$ that are orthogonal to $M$'s row space. *(ii)* $\sigma_i^N = \sigma_i^{N^\dagger}$ for $i \leq d_2$.

We rely on the following theorem (Chapter 2 in [49]) to construct the lower bound.

**Theorem 2.** *Let $\mathbb{N}^\dagger = \{N_1^\dagger, N_2^\dagger, \dots, N_K^\dagger\}$, where $K \geq 2$. Let $P_i$ be distribution of the training data produced from the model $\mathbf{y} = N_i^\dagger \mathbf{z}^\dagger + \epsilon$. Assume that*

- $\|N_i^\dagger - N_j^\dagger\|_F^2 \geq 2s > 0$ *for any $0 \leq j \leq k \leq K$.*

- *For any $j = 1, \dots K$ and*

$$\frac{1}{K} \sum_{j=1}^{K} \mathrm{KL}(P_j, P_1) \leq \alpha \log K \tag{37}$$

*with $0 \leq \alpha \leq \frac{1}{8}$.*

*Then*

$$\inf_{\hat{N}^\dagger} \sup_{N^\dagger \in \mathbb{N}^\dagger} \Pr_{N^\dagger}(\|\hat{N}^\dagger, N^\dagger\| \geq s) \geq \frac{\sqrt{K}}{1 + \sqrt{K}} \left(1 - 2\alpha - \sqrt{\frac{2\alpha}{\log K}}\right). \tag{38}$$

Because $L^\dagger$ is fixed, this problem boils down to finding a collection $\mathbb{U}^\dagger$ of unitary matrices in $\mathbf{R}^{d \times d}$ such that any two elements in $\mathbb{U}^\dagger$ are sufficiently far. Then we can construct $\mathbb{N}^\dagger = \{U^\dagger L^\dagger : U^\dagger \in \mathbb{U}\}$.

We next reiterate (with elaboration) the challenge we face when using existing techniques. Then we describe our approach. We shall first examine a simple case, in which we need only design vectors for one column. Then we explain the difficulties of using an existing technique to generalize the design. Finally, we explain our solution.

Recall that $(\mathbf{Z}^\dagger)^\mathrm{T}\mathbf{Y} = (N^\dagger)^\mathrm{T} + \mathcal{E}$, where we can roughly view $\mathcal{E}$ as a matrix that consists of independent Gaussian $(0, \sigma_\epsilon/\sqrt{n})$. For the sake of discussion, we assume $\sigma_\epsilon = 1$ in our discussion below.

**Warmup: one colume case.** The problem of packing one column roughly corresponds to establishing a lower bound on the estimation problem $\mathbf{y} = \mathbf{u} + \epsilon$, where $\mathbf{y}, \mathbf{u}, \epsilon \in \mathbf{R}^d$. $\epsilon$ corresponds to a column in $\mathcal{E}$ and consists of $d$ independent Gaussian $N(0, 1/\sqrt{n})$. $\mathbf{u}$ correponds to a column in $U^\dagger$ and we require $\|\mathbf{u}\|_2 \approx \rho\sqrt{d/n}$. To apply Theorem 2, we shall construct a $\mathbb{D} = \{\mathbf{u}_1, \ldots, \mathbf{u}_K\}$ such that $\|\mathbf{u}_i - \mathbf{u}_j\|$ is large for each $\{i, j\}$ pair and $K$ is also large. Specifically, we require $\|\mathbf{u}_i - \mathbf{u}_j\|_2^2 \approx 2\rho^2\frac{d}{n}$ (large distance requirement) and $K = \exp(\Theta(\sqrt{\rho}d))$ (large set requirement). $\sqrt{\rho}$ is carefully optimized and we will defer the reasoning to the full analysis below. A standard tool to construct $\mathbb{D}$ is to use a probabilistic method. We sample $\mathbf{u}_i$ independently from the same distribution and argue that with high probability $\|\mathbf{u}_i - \mathbf{u}_j\|_2^2$ is sufficiently large. Then a union bound can be used to derive $K$. For example, we may set $\mathbf{u}_i \sim N(0, \rho\sqrt{\frac{d}{n}}I_{d\times d})$ for all $i$, and the concentration quality suffices for us to sample $K$ vectors.

**Multiple column case.** We now move to the problem of packing multiple columns in $U$ together. $K$ is required to be much larger, e.g., $K = \exp(\Theta(\sqrt{\rho}d^2))$ for certain problem instances. A natural generalization of one-column case is to build our parameter set by taking the Cartesian product of multiple copies of $\mathbb{D}$. This gives us a large $K$ for free but the key issue is that vectors in $\mathbb{D}$ are generated independently. So there is no way to guarantee they are independent to each other. In fact, it is straightforward to show that many elements in the Cartesian product are far from unitary. One may also directly sample random unitary matrices and argue that they are far from each other. But there seems to exist no tool that enables us to build up a concentration of $\exp(-\Theta(\sqrt{\rho}d^2))$ between two random unitary matrices.

Therefore, a fundamental problem is that we need independence to build up a large set but the unitary constraint limits the independence. So we either cannot satisfy the unitary requirement (Cartesian product approach) or cannot properly use the independence to build concentrations (random unitary matrix approach).

**Our approach.** We develop a technique that decouples the three requirements (unitary matrices, large distance, and large $K$). Let us re-examine the Cartesian product approach. When the vectors for each column are completely determined, then it is remarkably difficult to build a Cartesian product that guarantees orthogonality between columns. To address this issue, our approach only "partially" specify vectors in each column. Then we take a Cartesian product of these partial specifications. So the size of the Cartesian product is sufficiently large; meanwhile an element in the product does not fully specify $U^\dagger$ so we still have room to make them unitary. Thus, our final step is to transform each partial specification in the Cartesian product into a unitary matrix. Specifically, it consists of three steps (See Fig. 3)

**Step 1. Partial specification of each column.** For each column $i$ of interest, we build up a collection $\mathbb{D}^{(i)} = \{R^{(i,1)}, \ldots, R^{(i,K)}\}$. Each $R^{(i,j)} \subset [d]$ specifies only the positions of non-zero entire for a vector prepared for filling in $U_{:,i}$.

**Step 2. Cartesian product.** Then we randomly sample elements from the Cartesian product $\mathbb{D} \triangleq \bigotimes_i \mathbb{D}^{(i)}$. Each element in the product specifies the non-zero entries of $U^\dagger$. We need to do another random sampling instead of using the full Cartesian product because we need to guarantee that any two matrices share a small number of non-zero entries. For example, $(R^{(1,1)}, R^{(2,1)}, R^{(3,1)})$ and $(R^{(1,1)}, R^{(2,1)}, R^{(3,2)})$ are two elements in $\bigotimes_i \mathbb{D}^{(i)}$ but they specify two matrices with the same locations of non-zero entries for the first two columns.

**Step 3. Building up unitary matrices.** Finally, for each element $\vec{R} \in \mathbb{D}$ (that specify positions of non-zero entries), we carefully fill in the values of the non-zero entries so that all our matrices are unitary and far from each other. We shall show that it is *always* possible to construct unitary matrices that "comply with" $\vec{R}$. In addition, our unitary matrices have few entries with large magnitude so when two matrices share few positions of non-zero entries, they are far.

## 4.2 Analysis

We now execute the plan outlined in the roadmap. Let $K$ and $\lambda$ be tunable parameters. For the purpose of getting intuition, $K$ is related to the size of $\mathbb{U}^\dagger$ so it can be thought as being exponential in $d$, whereas $\lambda$ is a constant and we use $\rho^\lambda$ to control the density of non-zero entries in each $U^\dagger \in \mathbb{U}^\dagger$.

Let $\mathbb{D}^{(i)} = \{R^{(i,1)}, R^{(i,2)}, \cdots, R^{(i,K)}\}$ be a collection of random subsets in $[d]$, in which each $R^{(i,j)}$ is of size $\rho^\lambda d$. We sample the subsets without replacement. Recall that $\underline{t}$ is the smallest index such that $\sigma_{\underline{t}}^\dagger \leq \rho\sigma_\epsilon\sqrt{\frac{d}{n}}$ and let us also define $\bar{t} = \lfloor \frac{\rho^\lambda d}{2} \rfloor$. We let $\gamma = \bar{t} - \underline{t} + 1$. We assume that $\bar{t} \geq \underline{t}$; otherwise Proposition 3 becomes trivial. Let $\mathbb{D}$ be the Cartesian product of $\mathbb{D}^{(i)}$ for integers $i \in [\underline{t}, \bar{t}]$. We use $\vec{R}$ to denote an element in $\mathbb{D}$. $\vec{R} = (\vec{R}_{\underline{t}}, \vec{R}_{\underline{t}+1}, \cdots, \vec{R}_{\bar{t}})$ is a $\gamma$-tuple so that each element $\vec{R}_i$ corresponds to an element in $\mathbb{D}^{(i)}$. There are two ways to represent $\vec{R}$. Both are found useful in our analysis.

1. *Set representation.* We treat $\vec{R}_i$ as a set in $\mathbb{D}^{(i)}$.

2. *Index representation.* We treat $\vec{R}_i$ as an index from $[K]$ that specifies the index of the set that $\vec{R}_i$ refers to.

Note that the subscript $i$ of $\vec{R}_i$ *starts at $\underline{t}$* (instead of 1 or 0) for convenience.

**Example 4.1.** The index of $\mathbb{D}_i$ starts at $\underline{t}$. Assume that $\bar{t} = \underline{t}+1$. Let $\mathbb{D}^{(\underline{t})} = \big(\{2,3\}, \{1,4\}, \{1,2\}\big)$ and $\mathbb{D}^{(\underline{t}+1)} = \big(\{1,3\}, \{2,4\}, \{3,4\}\big)$. The element $\big(\{1,2\}, \{2,4\}\big) \in \mathbb{D}^{(\underline{t})} \otimes \mathbb{D}^{(\underline{t}+1)}$. There are two ways to represent this element. *(i) Set representation.* $\vec{R} = \big(\{1,2\}, \{2,4\}\big)$, in which $\vec{R}_{\underline{t}} = \{1,2\}$ and $\vec{R}_{\underline{t}+1} = \{2,4\}$. *(ii) Index representation.* $\vec{R} = (3,2)$. $\vec{R}_{\underline{t}} = 3$ refers to that the third element $\{1,2\}$ in $\mathbb{D}^{(\underline{t})}$ is selected.

We now describe our proof in detail. We remark that throughout our analysis, constants are re-used in different proofs.

### 4.2.1 Step 1. Partial specification for each column

This step needs only characterize the behavior of an individual $\mathbb{D}^{(i)}$.

**Lemma 11.** *Let $\rho < 1$ be a sufficiently small variable, $\lambda$ be a tunable parameter and let $\mathbb{D}^{(i)} = \{R^{(i,1)}, R^{(i,2)}, \cdots, R^{(i,K)}\}$ be a collection of random subsets in $[d]$ (sampled without replacement) such that $|R^{(i,j)}| = \rho^\lambda d$ for all $j$. There exist constants $c_0$, $c_1$, and $c_2$ such that when $K = \exp(c_0\rho^{2\lambda}d)$, with probability $1 - \exp(-c_1\rho^{2\lambda}d)$, for any two distinct $R^{(i)}$ and $R^{(j)}$, $|R^{(i,j)} \cap R^{(i,k)}| \leq c_2\rho^{2\lambda}d$.*

*Proof of Lemma 11.* This can be proved by a standard probabilistic argument. Let $R^{(i,j)}$ be an arbitrary subset such that $|R^{(i,j)}| = \rho^\lambda d$. Let $R^{(i,k)}$ be a random subset of size $\rho^\lambda d$. We compute the probability that $|R^{(i,j)} \cap R^{(i,k)}| \geq c_2\rho^{2\lambda}d$ for a fixed $R^{(i,j)}$.

Let us sequentially sample elements from $R^{(i,k)}$ (without replacement). Let $I_t$ be an indicator random variable that sets to 1 if and only if the $t$-th random element in $R^{(i,k)}$ hits an element in $R^{(i,j)}$. We have

$$\Pr[I_t = 1] \leq \frac{\rho^\lambda d}{(1-\rho^\lambda)d} \leq \frac{\rho^\lambda}{2}. \tag{39}$$

By using a Chernoff bound, we have

$$\Pr\left[|\sum_{i=1}^{\rho^\lambda d} I_t| \geq \frac{c_2\rho^{2\lambda}d}{2}\right] \leq \exp(-\Omega(\rho^{2\lambda}d)). \tag{40}$$

By using a union bound, we have

$$\Pr[\exists i,j : |R^{(i,j)} \cap R^{(i,k)}| \geq \frac{c_2\rho^{2\lambda}d}{2}] \leq \binom{K}{2}\exp(-\Omega(\rho^{2\lambda}d)) \leq \exp(-\Omega(\rho^{2\lambda}d)+2\log K). \tag{41}$$

Therefore, when we set $K = \exp(c_0\rho^{2\lambda}d)$, the failure probability is $1 - \exp(-\Theta(\rho^{2\lambda}d))$. □

#### 4.2.2 Step 2. Random samples from the Cartesian product

We let $\mathbb{D} = \bigotimes_{i \in [\underline{t}, \overline{t}]} \mathbb{D}^{(i)}$. Note that each $\mathbb{D}^{(i)}$ is sampled independently. We define $\mathbb{S}$ be a random subset of $\mathbb{D}$. We next explain the reason we need to sample random subsets. Recall that for each $\vec{R} \in \mathbb{S}$, we aim to construct a unitary matrix $U^\dagger$ (to be discussed in Step 3) such that the positions of non-zero entries in $U_{:,i}^\dagger$ are specified by $\vec{R}_i$ (i.e., $U_{j,i}^\dagger \neq 0$ only when $j \in \vec{R}_i$).

Let $\vec{R}$ and $\vec{R}'$ be two distinct elements in $\mathbb{D}$. Let $U^\dagger$ and $\tilde{U}^\dagger$ be two unitary matrices generated by $\vec{R}$ and $\vec{R}'$. We ultimately aim to have that $\|U^\dagger L^\dagger - \tilde{U}^\dagger L^\dagger\|_F^2$ being large. We shall make sure *(i)* $U^\dagger$ and $\tilde{U}^\dagger$ share few non-zero positions (Step 2; this step), and *(ii)* few entries in $U^\dagger$ and $\tilde{U}^\dagger$ have excessive magnitude (Step 3). These two conditions will imply that $U^\dagger$ and $\tilde{U}^\dagger$ are far, which then implies a lower bound on $U^\dagger L^\dagger$ and $\tilde{U}^\dagger L^\dagger$.

Because we do not want $U^\dagger$ and $\tilde{U}^\dagger$ share non-zero positions, we want to maximize the Hamming distance (in index representation) between $\vec{R}$ and $\vec{R}'$ (i.e., $\vec{R}_i = \vec{R}'_i$ implies $U_{:,i}^\dagger$ and $\tilde{U}_{:,i}^\dagger$ share all non-zero positions, which is a bad event). We sample $\mathbb{S}$ randomly from $\mathbb{D}$ because random sample is a known procedure that generates "code" with large Hamming distance [30].

Before proceeding, we note one catch in the above explanation, i.e., different columns are of different importance. Specifically, $\|U^\dagger L^\dagger - \tilde{U}^\dagger L^\dagger\|_F^2 = \sum_{i \in [d]} (\sigma_i^\dagger)^2 \|U_{:,i}^\dagger - \tilde{U}_{:,i}^\dagger\|^2$. When $\vec{R}_i$ and $\vec{R}'_i$ collide for a large $(\sigma_i^\dagger)^2$, it makes more impact to the Frobenius norm. Thus, we define a weighted cost function that resembles the structure of Hamming distance.

$$\mathbf{c}(\vec{R}, \vec{R}') = \sum_{i \in [\underline{t}, \overline{t}]} (\sigma_i^\dagger)^2 I(\vec{R}_i = \vec{R}'_i). \tag{42}$$

Note that the direction we need for $\mathbf{c}(\vec{R}, \vec{R}')$ is opposite to Hamming distance. One usually maximizes Hamming distance whereas we need to minimize the weighted cost.

We need to develop a specialized technique to produce concentration behavior for $\mathbb{S}$ because $\mathbf{c}(\vec{R}, \vec{R}')$ is weighted.

**Lemma 12.** *Let $\rho$ and $\lambda$ be the parameters for producting $\mathbb{D}^{(i)}$. Let $\zeta < 1$ be a tunable parameter. Let $\mathbb{S}$ be a random subset of $\mathbb{D}$ of $\mathbb{D} \triangleq \bigotimes_{i \in [\underline{t}, \overline{t}]} \mathbb{D}^{(i)}$ such that*

$$|\mathbb{S}| = \exp\left( c_3 \frac{n\rho^{2\lambda + \zeta}}{\rho^2 \sigma_\epsilon^2} \left( \sum_{i \in [\underline{t}, \overline{t}]} (\sigma_i^\dagger)^2 \right) \right) \tag{43}$$

*for some constant $c_3$. With high probability at least $1 - \exp(-c_4 \rho^{2\lambda} d)$ ($c_4$ a constant), for any $\vec{R}$ and $\vec{R}'$ in $\mathbb{S}$, $\mathbf{c}(\vec{R}, \vec{R}') \leq \rho^\zeta (\sum_{i \in [\underline{t}, \overline{t}]} (\sigma_i^\dagger)^2)$.*

*Proof.* Let $\Psi = \sum_{i \in [\underline{t}, \overline{t}]} (\sigma_i^\dagger)^2$. Let $\vec{R}$ and $\vec{R}'$ be two different random elements in $\mathbb{D}$. We shall first compute that $\Pr\left[ \mathbf{c}(\vec{R}, \vec{R}') \geq \rho^\zeta \Psi \right]$. Here, we assume that $\vec{R}$ is an arbitrary fixed element and $\vec{R}'$ is random.

Recall that $\sigma_i^\dagger \in [0, \rho\sigma_\epsilon \sqrt{\frac{n}{d}}]$ for $i \in [\underline{t}, \overline{t}]$. We shall partition $[0, \rho\sigma_\epsilon \sqrt{\frac{n}{d}}]$ into subintervals and group $\sigma_i^\dagger$ by these intervals. Let $\mathcal{I}_t$ be the set of $\sigma_i^\dagger$ that are in $[2^{-t-1}\rho\sigma_\epsilon \sqrt{\frac{n}{d}}, 2^{-t}\rho\sigma_\epsilon \sqrt{\frac{n}{d}}]$ ($t \geq 0$). Let $T$ be the largest integer such that $\mathcal{I}_T$ is not empty. Let $L_t = |\mathcal{I}_t|$ and $\ell_t = \sum_{i \in \mathcal{I}_t} I(\vec{R}_i = \vec{R}'_i)$. We call $\{\ell_t\}_{t \leq T}$ the *overlapping coefficients* between $\vec{R}$ and $\vec{R}'$.

Note that $\mathbf{c}(\vec{R}, \vec{R}') \leq \sum_{t \leq T} \ell_t 2^{-2t} \rho^2 \sigma_\epsilon^2 d/n$. Therefore, a necessary condition for $\mathbf{c}(\vec{R}, \vec{R}') \geq \rho^\zeta \Psi$ is $\sum_{t \leq T} \frac{\ell_t 2^{-2t} \rho^2 \sigma_\epsilon^2 d}{n} \geq \rho^\eta \Psi$. Together with the requirement that $\sum_{t \leq T} \ell_t \geq 1$, we need

$$\sum_{t \leq T} \ell_t \geq \max\left\{ \frac{n\rho^\zeta \Psi}{d\rho^2 \sigma_\epsilon^2}, 1 \right\} \tag{44}$$

Recall that we assume that $\vec{R}$ is fixed and $\vec{R}'$ is random. When $\mathbf{c}(\vec{R}, \vec{R}') \geq \rho^\varsigma \Psi$, we say $\vec{R}'$ is bad. We next partition all bad $\vec{R}'$ into sets indexed by $\{\ell_t\}_{t \leq T}$. Let $\mathbb{C}(\{\ell_t\}_{t \leq T})$ be all the bad $\vec{R}'$ such that the overlapping coefficients between $\vec{R}$ and $\vec{R}'$ are $\{\ell_t\}_{t \leq T}$. We have

$$\Pr[\mathbf{c}(\vec{R}, \vec{R}') \geq \rho^\varsigma \Psi] = \Pr[\vec{R}' \text{ is bad}] = \Pr\left[\vec{R}' \in \bigcup_{\{\ell_t\}_{t \leq T}} \mathbb{C}(\{\ell_t\}_{t \leq T})\right] = \sum_{\substack{k \geq 1}} \sum_{\substack{\text{all } \mathbb{C}(\{\ell_i\}) \\ \text{s.t. } \sum_t \ell_t = k}} \Pr[\vec{R}' \in \mathbb{C}(\{\ell_t\}_{t \leq T})]$$

Next also note that

$$\Pr[\vec{R}' \in \mathbb{C}(\{\ell_t\}_{t \leq T})] \leq \prod_{t \leq T} \binom{L_t}{\ell_t} \left(\frac{1}{K}\right)^{\sum_{t \leq T} \ell_t},$$

where $K$ is the size of each $\mathbb{D}^{(i)}$.

The number of possible $\{\ell_i\}_{t \leq T}$ such that $\sum_{t \leq T} \ell_i = k$ is at most $\binom{d+k}{k}$. Therefore,

$$\sum_{\substack{k \geq 1}} \sum_{\substack{\text{all } \mathbb{C}(\{\ell_i\}) \\ \text{s.t. } \sum_t \ell_t = k}} \Pr\left[\vec{R}' \in \mathbb{C}(\{\ell_i\}_{i \leq T})\right]$$

$$\leq \sum_{\substack{k \geq 1}} \sum_{\substack{\text{all } \mathbb{C}(\{\ell_i\}) \\ \text{s.t. } \sum_t \ell_t = k}} \prod_{t \leq T} \binom{L_t}{\ell_t} \left(\frac{1}{K}\right)^{\ell_t}$$

$$\leq \sum_{\substack{k \geq 1}} \sum_{\substack{\text{all } \mathbb{C}(\{\ell_i\}) \\ \text{s.t. } \sum_t \ell_t = k}} \prod_{t \leq T} \left(\frac{eL_t}{K}\right)^{\ell_t} \quad \left(\text{using } \binom{L_t}{\ell_t} \leq \left(\frac{eL_t}{\ell_t}\right)^{\ell_t} \leq (eL_i)^{\ell_i}\right)$$

$$\leq \sum_{k \geq 1} \binom{d+k}{k} \prod_{t \leq T} \left(\frac{eL_t}{K}\right)^{\ell_t}$$

$$\leq d \max_{\substack{k \text{ s.t.} \\ \sum_t \ell_t = k}} \binom{d+k}{k} \prod_{i \leq T} \left(\frac{eL_i}{K}\right)^{\ell_t}$$

$$\leq d \left(\frac{e(d+k)}{k}\right)^k \prod_{t \leq T} \left(\frac{eL_t}{K}\right)^{\ell_t} \quad \left(\text{where } k = \textstyle\sum_{t \leq T} \ell_t \text{ from the previous line}\right)$$

$$\leq d(2ed)^k \prod_{t \leq T} \left(\frac{eL_t}{K}\right)^{\ell_t}$$

$$\leq d \prod_{t \leq T} \left(\frac{2e^2 d^2}{K}\right)^{\ell_t} \leq \exp\left(-c\rho^{2\lambda} d (\textstyle\sum_{t \leq T} \ell_t)\right) \quad (c \text{ is a suitable constant; using } K \text{ from Lemma } 11)$$

$$\leq \exp\left(-c\rho^{2\lambda} d \max\left\{\frac{n\rho^\varsigma \Psi}{d\rho^2 \sigma_\epsilon^2}, 1\right\}\right)$$

By using a union bound on all pairs of $\vec{R}$ and $\vec{R}'$ in $\mathbb{S}$, we have for sufficiently small $c_3$, there exists a $c_4$ such that

$$\Pr\left[\exists \vec{R}, \vec{R}' \in \mathbb{S} : \mathbf{c}(\vec{R}, \vec{R}') \geq \rho^\varsigma \Psi\right] \leq \exp\left(-c_4 \rho^{2\lambda} d \max\left\{\frac{n\rho^\varsigma \Psi}{d\rho^2 \sigma_\epsilon^2}, 1\right\}\right) \leq \exp(-c_4 \rho^{2\lambda} d).$$

$\square$

### 4.2.3   Step 3. Building up unitary matrices

We next construct a set of unitary matrices in $\mathbf{R}^{d \times d}$ based on elements in $\mathbb{S}$. We shall refer to the procedure to generate a unitary $U$ from an element $\vec{R} \in \mathbb{S}$ as $q(\vec{R})$.

**Procedure** $q(\vec{R})$**:** Let $U \in \mathbf{R}^{d \times d}$ be the matrix $q(\vec{R})$ aims to fill in. Let $\mathbf{v}^{(1)}, \mathbf{v}^{(2)}, \ldots, \mathbf{v}^{(\underline{t}-1)} \in \mathbf{R}^d$ be an arbitrary set of orthonormal vectors. $q(\vec{R})$ partitions the matrix $U$ into three regions of columns and it fills different regions with different strategies. See also three regions illustrated by matrices in Fig. 3.

*Region 1:* $U_{:,i}$ *for* $i < \underline{t}$. We set $U_{:,i} = \mathbf{v}^{(i)}$ when $i < \underline{t}$. This means all the unitary matrices we construct share the first $\underline{t} - 1$ columns.

*Region 2:* $U_{:,i}$ *for* $i \in [\underline{t}, \overline{t}]$. We next fill in non-zero entries of each $U_{:,i}$ by $\vec{R}_i$ for $i \in [\underline{t}, \overline{t}]$. We fill in each $U_{:,i}$ sequentially from left to right (from small $i$ to large $i$). We need to make sure that *(i)* it is feasible to construct $U_{:,i}$ that is orthogonal to all $U_{:,j}$ ($j < i$) by using only entries specified by $R_i$. *(ii)* there is a way to fill in $U_{:,i}$ so that not too many entries are excessively large. (ii) is needed because for any $\vec{R}$ and $\vec{R}'$, $\vec{R}_i$ and $\vec{R}'_i$ still share a small number of non-zero positions (whp $|\vec{R}_i \cap \vec{R}'_i| = O(\rho^{2\lambda}d)$, according to Lemma 11). When the mass in $|\vec{R}_i \cap \vec{R}'_i|$ is large, the distance between $U$ and $U'$ is harder to control.

*Region 3:* $U_{:,i}$ *for* $i > \overline{t}$. $q(\vec{R})$ fills in the rest of the vectors arbitrarily so long as $U$ is unitary. Unlike the first $\underline{t} - 1$ columns, these columns depend on $\vec{R}$ so each $U \in \mathbb{U}$ has a different set of ending column vectors.

**Analysis for region 2.** Our analysis focuses on two asserted properties for Region 2 are true. We first show (i) is true and describe a procedure to make sure (ii) happens.

For $j \leq i - 1$, let $\mathbf{w}^{(j)} \in \mathbf{R}^{\rho^\lambda d}$ be the projection of $U_{:,j}$ onto the coordinates specified by $\vec{R}_i$. See Fig. 3(d) for an illustration. Note that $\overline{t} = \frac{\rho^\lambda d}{2}$, the dimension of the subspace spanned by $\mathbf{w}^{(1)}, \ldots \mathbf{w}^{(i-1)}$ is at most $\rho^\lambda d/2$. Therefore, we can find a set of orthonormal vectors $\{\mathbf{u}^{(1)}, \ldots \mathbf{u}^{(\kappa)}\} \subseteq \mathbf{R}^{\rho^\lambda d}$ ($\kappa \geq \frac{\rho^\lambda d}{2}$) that are orthogonal to $\mathbf{w}^{(j)}$ ($j \leq i - 1$). To build a $U_{:,i}$ that's orthogonal to all $U_{:,j}$ ($j \leq i - 1$), we can first find a $\mathbf{u} \in \mathbf{R}^{\rho^\lambda d}$ that is a linear combination of $\{\mathbf{u}^{(j)}\}_{j \leq \kappa}$, and then "inflate" $\mathbf{u}$ back to $\mathbf{R}^d$, i.e., the $k$-th non-zero coordinate of $U_{:,i}$ is $\mathbf{u}_k$. One can see that

$$\langle U_{:,j}, U_{:,i} \rangle = \langle \mathbf{w}^{(j)}, \mathbf{u} \rangle = 0$$

for any $j < i$.

We now make sure (ii) happens. We have the following Lemma.

**Lemma 13.** *Let* $\{\mathbf{u}^{(1)}, \ldots, \mathbf{u}^{(\kappa)}\}$ *(*$\kappa \geq \rho^\lambda d/2$*) be a collection of orthonormal vectors in* $\mathbf{R}^{\rho^\lambda d}$*. Let* $\eta$ *be a small tunable parameter. There exists a set of coefficients* $\beta_1, \ldots, \beta_\kappa$ *such that* $\mathbf{u} = \sum_{i=1}^{\kappa} \beta_i \mathbf{u}^{(i)}$ *is a unit vector and there exist constant* $c_5$*,* $c_6$*, and* $c_7$ *such that*

$$\sum_{i \leq \rho^\lambda d} \mathbf{u}_i^2 I\left(\mathbf{u}_i \geq \frac{c_5}{\sqrt{\rho^{\lambda+\eta}d}}\right) \leq c_7 \xi, \tag{45}$$

*where* $\xi = \exp(-\frac{c_6}{\rho^\eta})$.

*Proof of Lemma 13.* We use a probabilistic method to find $\mathbf{u}$. Let $z_i \sim N(0, 1/\sqrt{\rho^\lambda d})$ for $i \in [\rho^\lambda d]$. Let $S = \sqrt{\sum_{i \leq \rho^\lambda d} z_i^2}$. We shall set $\beta_i = z_i/S$. One can check that $\mathbf{u} = \sum_{i \in [\rho^\lambda d]} \beta_i \mathbf{u}^{(i)}$ is a unit vector. We then examine whether (45) is satisfied for these $\beta_i$'s we created. If not, we re-generate a new set of $z_i$'s and $\beta_i$'s. We repeat this process until (45) is satisfied.

Because $\beta_i$'s are normalized, setting the standard deviation of $z_i$ is unnecessary. We nevertheless do so because $S$ will be approximately a constant, which helps us simplify the calculation.

We claim that there exists a constant $c$ such that for any $\ell$,

$$\mathbb{E}\left[\mathbf{u}_\ell^2 I\left(\mathbf{u}_\ell \geq \frac{c_5}{\sqrt{\rho^{\lambda+\eta}d}}\right)\right] \leq \frac{c\xi}{\rho^\lambda d}. \tag{46}$$

We first show that (46) implies Lemma 13. Then we will show (46).

By linearity of expectation, (46) implies

$$\mathbb{E}\left[\sum_{\ell \leq \rho^\lambda d} \mathbf{u}_\ell^2 I\left(\mathbf{u}_\ell \geq \frac{c_5}{\sqrt{\rho^{\lambda+\eta}d}}\right)\right] \leq c\xi.$$

Then we use a Markov inequality and obtain

$$\Pr\left[\left(\sum_{\ell \leq \rho^\lambda d} \mathbf{u}_\ell^2 I\left(\mathbf{u}_\ell \geq \frac{c_5}{\sqrt{\rho^{\lambda+\eta}d}}\right)\right) \geq 2c\xi\right] \leq \frac{1}{2}$$

So our probabilistic method described above is guaranteed to find a $\mathbf{u}$ that satisfies (45)

We next move to showing (46). Recall that

$$\mathbf{u}_\ell = \frac{1}{S}(\sum_{i \leq \kappa} z_i \mathbf{u}_\ell^{(i)}).$$

We also let $Z_\ell = \sum_{i \leq \kappa} z_i \mathbf{u}_\ell^{(i)}$. We can see that $Z_\ell$ is a Gaussian random variable with a standard deviation $\sqrt{\frac{1}{\rho^\lambda d} \sum_{i \leq \kappa}(\mathbf{u}_\ell^{(i)})^2} \leq \sqrt{\frac{1}{\rho^\lambda d}}$. The inequality uses the fact that $\mathbf{u}^{(i)}$'s are orthonormal to each other and therefore $\sum_{i \leq \kappa}(\mathbf{u}_\ell^{(i)})^2 \leq 1$.

On the other hand, one can see that

$$\mathbb{E}[S] = \mathbb{E}[\sum_{i \leq \kappa} z_i^2] = \left(\frac{1}{\sqrt{\rho^\lambda d}}\right)^2 \cdot \kappa \geq \frac{1}{2}.$$

Therefore, by a standard Chernoff bound, $\Pr\left[S \leq \frac{1}{4}\right] \leq \exp(-\Theta(\kappa))$.

Next, because $\{\mathbf{z}_i\}_{i \leq \kappa}$ collectively form a spherical distribution, we have $\left\{\frac{z_i}{S}\right\}_{i \leq \kappa}$ is independent to $S$. Therefore,

$$\mathbb{E}\left[\mathbf{u}_\ell^2 I\left(\mathbf{u}_\ell \geq \frac{c_5}{\sqrt{\rho^{\lambda+\eta}d}}\right)\right] = \mathbb{E}\left[\mathbf{u}_\ell^2 I\left(\mathbf{u}_\ell \geq \frac{c_5}{\sqrt{\rho^{\lambda+\eta}d}}\right) \mid S \geq \frac{1}{4}\right] \tag{47}$$

Conditioned on $S \geq \frac{1}{4}$, we use the fact that $\mathbf{u}_\ell = Z_\ell/S$ to get $\mathbf{u}_\ell \leq 4Z_\ell$ and

$$I\left(\mathbf{u}_\ell \geq \frac{c_5}{\sqrt{\rho^{\lambda+\eta}d}}\right) = I\left(Z_\ell \geq \frac{c_5}{\sqrt{\rho^{\lambda+\eta}d}}S\right) \leq I\left(Z_\ell \geq \frac{c_5}{4\sqrt{\rho^{\lambda+\eta}d}}\right). \tag{48}$$

Therefore,

(47)

$$\leq 16\mathbb{E}\left[Z_\ell^2 I\left(Z_\ell \geq \frac{c_5}{4\sqrt{\rho^{\lambda+\eta}d}}\right) \mid S \geq \frac{1}{4}\right]$$

$$= \frac{16}{\Pr\left[S \geq \frac{1}{4}\right]}\left(\mathbb{E}\left[Z_\ell^2 I\left(Z_\ell \geq \frac{c_5}{4\sqrt{\rho^{\lambda+\eta}d}}\right)\right] - \mathbb{E}\left[Z_\ell^2 I\left(Z_\ell \geq \frac{c_5}{4\sqrt{\rho^{\lambda+\eta}d}}\right) \mid S < \frac{1}{4}\right]\Pr\left[S \leq \frac{1}{4}\right]\right)$$

$$\leq 16(1 + \exp(-\Theta(\kappa)))\left(\mathbb{E}\left[Z_\ell^2 I\left(Z_\ell \geq \frac{c_5}{4\sqrt{\rho^{\lambda+\eta}d}}\right)\right]\right)$$

$$\leq 16(1 + \exp(-\Theta(\kappa)))\mathbb{E}\left[Z_\ell^2 I\left(Z_\ell \geq \frac{2c_5}{\sqrt{\rho^{\lambda+\eta}d}}\right)\right]$$

$$\leq \frac{16(1 + \exp(-\Theta(\kappa)))}{\sqrt{2\pi}\sigma_{Z_\ell}}\int_{\frac{2c_5}{\sqrt{\rho^{\lambda+\eta}d}}}^{\infty} z^2 \exp\left(\frac{z^2}{\sigma_{Z_\ell}^2}\right)dz$$

$$\leq \frac{c}{\rho^{\lambda+\eta}d}\exp\left(-\Theta(\rho^{-\eta})\right) \quad \text{(using } \sigma_{Z_\ell} \leq \sqrt{\frac{1}{\rho^\lambda d}}\text{)}$$

$$\leq O(\frac{\xi}{\rho^\lambda d})$$

$\square$

#### 4.2.4 Proof of Proposition 3

Now we are ready to use Theorem 2 to prove Proposition 3. Let $\mathbb{S}$ be the set constructed from Step 1 and $\mathbb{U}^\dagger = \{U^\dagger : U^\dagger = q(\vec{R}), \vec{R} \in \mathbb{S}\}$. Let $\mathbb{N}^\dagger = \{U^\dagger L^\dagger : U^\dagger \in \mathbb{U}^\dagger\}$. Let also $\mathbf{P}_{N^\dagger}$ be the distribution of $(\mathbf{y}, \mathbf{z}^\dagger)$ generated from the model $\mathbf{y} = N^\dagger\mathbf{z}^\dagger + \epsilon$. Let $\mathbf{P}_{N^\dagger|\mathbf{z}^\dagger}$ be the distribution of $\mathbf{y}$ from the normalized model when $\mathbf{z}^\dagger$ is given. Let $\mathbf{P}_{N^\dagger,n}$ be the product distribution when we observe $n$ samples from the model $\mathbf{y} = N^\dagger\mathbf{z}^\dagger + \epsilon$. Let $f_{N^\dagger}(\mathbf{y}, \mathbf{z}^\dagger)$ be the pdf of for $\mathbf{P}_{N^\dagger}$, $f_{N^\dagger}(\mathbf{y} \mid \mathbf{z}^\dagger)$ be the pdf of $\mathbf{y}$ given $\mathbf{z}^\dagger$, and $f(\mathbf{z}^\dagger)$ be the pdf of $\mathbf{z}^\dagger$.

We need to show that for any $N^\dagger, \tilde{N}^\dagger \in \mathbb{N}$ *(i)* $\|N^\dagger - \tilde{N}^\dagger\|_F$ is large, and *(ii)* the KL-divergence between $\mathbf{P}_{N^\dagger,n}$ and $\mathbf{P}_{N^\dagger,n}$ is bounded.

**Lemma 14.** *Let $\mathbb{S}$, $\mathbb{U}^\dagger$, and $\mathbb{N}^\dagger$ be generated by the parameters $\lambda, \eta$, and $\zeta$ (see Steps 1 to 3). For any $N^\dagger$ and $\tilde{N}^\dagger$ in $\mathbb{N}$, we have*

$$\|N^\dagger - \tilde{N}^\dagger\|_F^2 \geq \sum_{i\in[\underline{t},\overline{t}]}\sigma_i^\dagger(2 - c_8\rho^{\lambda-\eta} - c_9\rho^\zeta) \tag{49}$$

*for some constant $c_8$ and $c_9$.*

*Proof.* Let $U^\dagger, \tilde{U}^\dagger \in \mathbb{U}^\dagger$ such that $N^\dagger = U^\dagger L^\dagger$ and $\tilde{N}^\dagger = \tilde{U}^\dagger L^\dagger$; let $\vec{R}, \vec{R}' \in \mathbb{S}$ be that $U^\dagger = q(\vec{R})$ and $\tilde{U}^\dagger = q(\vec{R}')$. Also, recall that we let $\Psi = \sum_{i\in[\underline{t},\overline{t}]}(\sigma_i^\dagger)^2$.

We have

$$\|N^\dagger - \tilde{N}^\dagger\|_F^2 = \sum_{i\in[d]}\|(U_{:,i}^\dagger - \tilde{U}_{:,i}^\dagger)\sigma_i^\dagger\|_2^2 \geq \sum_{i\in[\underline{t},\overline{t}]}\|(U_{:,i}^\dagger - \tilde{U}_{:,i}^\dagger)\sigma_i^\dagger\|_2^2$$

Let also $H = \{\vec{R}_i = \vec{R}_i', i \in [\underline{t}, \overline{t}]\}$, i.e., the set of coordinates that $\vec{R}$ and $\vec{R}'$ agree. Whp, we have

$$\Psi = \sum_{i\in H}(\sigma_i^\dagger)^2 + \sum_{\substack{i\in[\underline{t},\overline{t}]\\i\notin H}}(\sigma_i^\dagger)^2 = \mathbf{c}(\vec{R}, \vec{R}') + \sum_{\substack{i\in[\underline{t},\overline{t}]\\i\notin H}}(\sigma_i^\dagger)^2 \leq \rho^\zeta\Psi + \sum_{\substack{i\in[\underline{t},\overline{t}]\\i\notin H}}(\sigma_i^\dagger)^2 \tag{50}$$

The last inequality holds because of Lemma 12. Therefore, we have $\sum_{\substack{i\in[\underline{t},\overline{t}]\\i\notin H}}(\sigma_i^\dagger)^2 \geq (1 - \rho^\zeta)\Psi$.

Now we have

$$\sum_{i\in[\underline{t},\bar{t}]} \|(U^\dagger_{:,i} - \tilde{U}^\dagger_{:,i})\sigma^\dagger_i\|^2_2 \geq \sum_{\substack{i\in[\underline{t},\bar{t}]\\ i\notin H}} \|(U^\dagger_{:,i} - \tilde{U}^\dagger_{:,i}\|^2_2 (\sigma^\dagger_i)^2.$$

Next we bound $\|U^\dagger_{:,i} - \tilde{U}^\dagger_{:,i}\|^2_2$ when $\vec{R}_i \neq \vec{R}'_i$. We have

$$\|U^\dagger_{:,i} - \tilde{U}^\dagger_{:,i}\|^2_2 \geq 2 - \sum_{j\in\vec{R}_i\cap\vec{R}'_i}\left((U^\dagger_{j,i})^2 + (\tilde{U}^\dagger_{j,i})^2\right). \tag{51}$$

By Lemma 11, whp $|\vec{R}_i \cap \vec{R}'_i| \leq c_2\rho^{2\lambda}d$. Next, we give a bound for $\sum_{j\in\vec{R}_i\cap\vec{R}'_i}(U^\dagger_{j,i})^2$. The bound for $\sum_{j\in\vec{R}_i\cap\vec{R}'_i}(\tilde{U}^\dagger_{j,i})^2$ can be derived in a similar manner.

For each $j \in |\vec{R}_i \cap \vec{R}'_i|$, we check whether $U^\dagger_{j,i} \geq \frac{c_5}{\sqrt{\rho^{\lambda+\eta}d}}$:

$$\sum_{j\in\vec{R}_i\cap\vec{R}'_i}(U^\dagger_{j,i})^2 = \sum_{j\in\vec{R}_i\cap\vec{R}'_i}\left[(U^\dagger_{j,i})^2 I(U^\dagger_{j,i} \leq \frac{c_5}{\sqrt{\rho^{\lambda+\eta}d}}) + (U^\dagger_{j,i})^2 I(U^\dagger_{j,i} > \frac{c_5}{\sqrt{\rho^{\lambda+\eta}d}})\right]$$

$$\leq O\left(\frac{\rho^{2\lambda}d}{\rho^{\lambda+\eta}d} + \exp(-\Theta(1/\rho^\eta))\right) = O(\rho^{\lambda-\eta}).$$

Therefore,

$$\sum_{i\in[\underline{t},\bar{t}]} \|(U^\dagger_{:,i} - \tilde{U}^\dagger_{:,i})(\sigma^\dagger_i)^2\|^2_2 \geq \sum_{i\in[\underline{t},\bar{t}]}(\sigma^\dagger_i)^2(2 - O(\rho^{\lambda-\eta})(1-\rho^\zeta)) \geq \sum_{i\in[\underline{t},\bar{t}]}(\sigma^\dagger_i)^2(2 - c_8\rho^{\lambda-\eta} - c_9\rho^\zeta).$$

$\square$

We need two additional building blocks.

**Lemma 15.** *Consider the regression problem* $\mathbf{y} = M\mathbf{x} + \epsilon$, *where* $\|M\| \leq \Upsilon = O(1)$ *and the eigenvalues* $\sigma_i(\mathbb{E}[\mathbf{x}\mathbf{x}^T])$ *of the features follow a power law distribution with exponent* $\omega$. *Consider the problem* $\mathbf{y} = N\mathbf{z} + \epsilon$ *in orthogonalized form. Let* $\sigma^N_i$ *be the $i$-th singular value of $N$. Let $t$ be an arbitrary value in* $[0, \min\{d_1, d_2\}]$. *There exists a constant $c_{10}$ such that*

$$\sum_{i\geq t}\sigma^2_i(N) \leq \frac{c_{10}}{t^{\omega-1}}.$$

*Proof of Lemma 15.* Without loss of generality, assume that $d_1 \geq d_2$. We first split the columns of $N$ into two parts, $N = [N_+, N_-]$, in which $N_+ \in \mathbf{R}^{d_2\times t}$ consists of the first $t$ columns of $N$ and $N_- \in \mathbf{R}^{d_2\times(d_1-t)}$ consists of the remaining columns. Let $\mathbf{0}$ be a zero matrix in $\mathbf{R}^{d_2\times(d_1-t)}$. $[N_+, \mathbf{0}]$ is a matrix of rank at most $t$.

Let us split other matrices in a similar manner.

- $M = [M_+, M_-]$, where $M_+ \in \mathbf{R}^{d_2\times t}$ and $M_- \in \mathbf{R}^{d_2\times(d_1-t)}$,

- $V^* = [V^*_+, V^*_-]$, where $V^*_+ \in \mathbf{R}^{d_1\times t}$, and $V^*_- \in \mathbf{R}^{d_1\times(d_1-t)}$, and

- $\Lambda^* = [\Lambda^*_+, \Lambda^*_-]$, where $\Lambda^*_+ \in \mathbf{R}^{d_1\times t}$ and $\Lambda^*_- \in \mathbf{R}^{d_1\times(d_1-t)}$.

We have

$$\sum_{i\geq t}(\sigma^N_i)^2 = \|N - \mathbf{P}_t(N)\|^2_F$$

$$\leq \|N - [N_+, \mathbf{0}]\|^2_F \quad (\mathbf{P}_t(N) \text{ gives an optimal rank-}t \text{ approximation of } N).$$

$$= \|N_-\|^2_F \leq \|M_-V^*_-\|^2\|(\Lambda^*_-)^{\frac{1}{2}}\|^2_F \leq \frac{c_{10}}{t^{\omega-1}}$$

$\square$

We next move to our second building block.

**Fact 4.1.**

$$\mathrm{KL}(\mathbf{P}_{N^\dagger,n}, \mathbf{P}_{\tilde{N}^\dagger,n}) = \frac{n\|N^\dagger - \tilde{N}^\dagger\|_F^2}{2\sigma_\epsilon^2}. \tag{52}$$

*Proof of Fact 4.1.*

$$
\begin{aligned}
& \mathrm{KL}(\mathbf{P}_{N^\dagger,n}, \mathbf{P}_{\tilde{N}^\dagger,n}) \\
=\ & n\mathrm{KL}(\mathbf{P}_{N^\dagger}, \mathbf{P}_{\tilde{N}^\dagger}) \\
=\ & n\mathbb{E}_{\mathbf{y}=N^\dagger \mathbf{z}^\dagger+\epsilon}\left[\log\left(\frac{f_{N^\dagger}(\mathbf{y},\mathbf{z}^\dagger)}{f_{\tilde{N}^\dagger}(\mathbf{y},\mathbf{z}^\dagger)}\right)\right] \\
=\ & n\mathbb{E}_{\mathbf{z}^\dagger}\left[\mathbb{E}_{\mathbf{y}=N^\dagger \mathbf{z}^\dagger+\epsilon}\left[\log\left(\frac{f_{N^\dagger}(\mathbf{y},\mathbf{z}^\dagger)}{f_{\tilde{N}^\dagger}(\mathbf{y},\mathbf{z}^\dagger)}\right)\mid \mathbf{z}^\dagger\right]\right] \\
=\ & n\mathbb{E}_{\mathbf{z}^\dagger}\left[\mathbb{E}_{\mathbf{y}=N^\dagger \mathbf{z}^\dagger+\epsilon}\left[\log\left(\frac{f_{N^\dagger}(\mathbf{y}\mid\mathbf{z}^\dagger)f(\mathbf{z}^\dagger)}{f_{\tilde{N}^\dagger}(\mathbf{y}\mid\mathbf{z}^\dagger)f(\mathbf{z}^\dagger)}\right)\mid \mathbf{z}^\dagger\right]\right] \\
=\ & n\mathbb{E}_{\mathbf{z}}\mathbb{E}_{\mathbf{y}=N^\dagger \mathbf{z}^\dagger+\epsilon}\left[\left[\log\left(\frac{f_{N^\dagger}(\mathbf{y}\mid\mathbf{z}^\dagger)}{f_{\tilde{N}^\dagger}(\mathbf{y}\mid\mathbf{z}^\dagger)}\right)\mid \mathbf{z}^\dagger\right]\right] \\
=\ & n\mathbb{E}_{\mathbf{z}}\left[\mathrm{KL}(\mathbf{P}_{N^\dagger\mid\mathbf{z}^\dagger}, \mathbf{P}_{\tilde{N}^\dagger\mid\mathbf{z}^\dagger})\right] = \frac{n\|N^\dagger - \tilde{N}^\dagger\|_F^2}{2\sigma_\epsilon^2}
\end{aligned}
$$

$\square$

We now complete the proof for Proposition 3. First, define $\psi \triangleq \frac{\sum_{i\geq\bar{t}+1}(\sigma_i^\dagger)^2}{\sum_{i\in[\underline{t},\bar{t}]}(\sigma_i^\dagger)^2}$. For any $N^\dagger$ and $\tilde{N}^\dagger$ in $\mathbb{N}$, we have

$$\|N^\dagger - \tilde{N}^\dagger\|_F^2 = \sum_{i\geq\underline{t}}(\sigma_i^\dagger)^2 = (1+\psi)\Psi,$$

where recall that $\Psi = \sum_{i\in[\underline{t},\bar{t}]}(\sigma_i^\dagger)^2$. Using Lemma 14, we have $\mathrm{KL}(\mathbf{P}_{N^\dagger,n}, \mathbf{P}_{\tilde{N}^\dagger,n}) = n(1+\psi)\Psi$. Next, we find a smallest $\alpha$ such that

$$\max_{N^\dagger,\tilde{N}^\dagger} \mathrm{KL}(\mathbf{P}_{N^\dagger,n}, \mathbf{P}_{\tilde{N}^\dagger,n}) \leq \alpha\log|\mathbb{N}|. \tag{53}$$

By Lemma 12, we have $|\mathbb{N}| = \exp(c_3\frac{n\rho^{2\lambda+\zeta-2}}{\sigma_\epsilon^2}\Psi)$. (53) is equivalent to requiring

$$\frac{n(1+\psi)\Psi}{2\sigma_\epsilon^2} \leq \frac{\alpha c_3 n\rho^{2\lambda+\zeta-2}\Psi}{\sigma_\epsilon^2}.$$

We may thus set $\alpha = O(\rho^{2-2\lambda+\zeta}(1+\psi))$. Now we may invoke Theorem 2 and get

$$
\begin{aligned}
\mathbf{r}(\mathbf{x}, M, n, \sigma_\epsilon) &\geq \Psi\left(1 - \frac{1}{\sqrt{|\mathbb{N}|}}\right)\underbrace{(2 - c_8\rho^{\lambda-\eta} - c_9\rho^\zeta)}_{\text{Lemma 14}}\left(1 - O(\rho^{2-2\lambda-\zeta})(1+\psi)\right) \\
&\geq \Psi(1 - O(\rho^{\lambda-\eta} + \rho^\zeta + \rho^{2-2\lambda-\zeta})) - \underbrace{\Psi\psi}_{=\sum_{i>\bar{t}}(\sigma_i^\dagger)^2}\rho^{2-2\lambda-\zeta} \\
&\geq \Psi(1 - O(\rho^{\lambda-\eta} + \rho^\zeta + \rho^{2-2\lambda-\zeta})) - O\left(\frac{1}{(\rho^\lambda d)^{\omega-1}}\right)\rho^{2-2\lambda-\zeta} \\
&\qquad \text{(Use Lemma 15 to bound } \sum_{i>\bar{t}}(\sigma_i^\dagger)^2 )
\end{aligned}
$$

We shall set $\xi = \eta$ be a small constant (say 0.001), $\lambda = \frac{1}{2} + \xi$, and $\zeta = \frac{1}{2}$. This gives us

$$\mathbf{r}(\mathbf{x}, M, n, \sigma_\epsilon) \geq \Psi(1 - O(\rho^{\frac{1}{2}} + \rho^{\frac{1}{2}-2\xi})) - \frac{\rho^{\frac{1}{2}-2\xi-(\omega-1)(\frac{1}{2}+\xi)}}{d^{\omega-1}}$$

Together with the fact that $\sum_{i>\bar{t}}(\sigma_i^\dagger)^2 = O\left(\frac{1}{(\rho^\lambda\omega)^{\omega-1}}\right)$, we complete the proof of Proposition 3.

# 5 Related work and comparison

In this section, we compare our results to other regression algorithms that make low rank constraints on $M$. Most existing MSE results are parametrized by the rank or spectral properties of $M$, e.g. [33]defined a generalized notion of rank

$$\mathbb{B}_q(R_q^A) \in \left\{ A \in \mathbf{R}^{d_2 \times d_1} : \sum_{i=1}^{\min\{d_1,d_2\}} |\sigma_i^A|^q \le R_q \right\}, \quad q \in [0,1], A \in \{N, M\}, \qquad (54)$$

i.e. $R_q^N$ characterizes the generalized rank of $N$ whereas $R_q^M$ characterizes that of $M$. When $q = 0$, $R_q^N = R_q^M$ is the rank of the $N$ because $\operatorname{rank}(N) = \operatorname{rank}(M)$ in our setting. In their setting, the MSE is parametrized by $R^M$ and is shown to be $O\left( R_q^M \left( \frac{\sigma_\epsilon^2 \lambda_1^*(d_1+d_2)}{(\lambda_{\min}^*)^2 n} \right)^{1-q/2} \right)$. In the special case when $q = 0$, this reduces to $O\left( \frac{\sigma_\epsilon^2 \lambda_1^* \operatorname{rank}(M)(d_1+d_2)}{(\lambda_{\min}^*)^2 \cdot n} \right)$. On the other hand, the MSE in our case is bounded by (cf. Thm. 1). We have $\mathbb{E}[\|\hat{\mathbf{y}} - \mathbf{y}\|_2^2] = O\left( R_q^N (\frac{\sigma_\epsilon^2 d_2}{n})^{1-q/2} + n^{-c_0} \right)$. When $q = 0$, this becomes $O\left( \frac{\sigma_\epsilon^2 \operatorname{rank}(M) d_2}{n} + n^{-c_0} \right)$.

The improvement here is twofold. First, our bound is directly characterized by $N$ in orthogonalized form, whereas result of [33]needs to examine the interaction between $M$ and $C^*$, so their MSE depends on both $R_q^M$ and $\lambda_{\min}^*$. Second, our bound no longer depends on $d_1$ and pays only an additive factor $n^{-c_0}$, thus, when $n < d_1$, our result is significantly better. Other works have different parameters in the upper bounds, but all of these existing results require that $n > d_1$ to obtain non-trivial upper bounds [27, 8, 11, 27]. Unlike these prior work, we require a stochastic assumption on $\mathbf{X}$ (the rows are i.i.d.) to ensure that the model is identifiable when $n < d_1$, e.g. there could be two sets of disjoint features that fit the training data equally well. Our algorithm produces an adaptive model whose complexity is controlled by $k_1$ and $k_2$, which are adjusted dynamically depending on the sample size and noise level. [8] and [11] also point out the need for adaptivity; however they still require $n > d_1$ and make some strong assumptions. For instance, [8]assumes that there is a gap between $\sigma_i(\mathbf{X}M^{\mathrm{T}})$ and $\sigma_{i+1}(\mathbf{X}M^{\mathrm{T}})$ for some $i$. In comparison, our sufficient condition, the decay of $\lambda_i^*$, is more natural. Our work is not directly comparable to standard variable selection techniques such as LASSO [48] because they handle univariate $\mathbf{y}$. Column selection algorithms [14] generalize variable selection methods for vector responses, but they cannot address the identifiability concern.

## 5.1 Missing proof in comparison

This section proves the following corollary.

**Corollary 2.** *Use the notation appeared in Theorem 1. Let the ground-truth matrix $N \in \mathbb{B}_q(R_q^N)$ for $q \in [0, 1]$. We have whp*

$$\mathbb{E}[\|\hat{\mathbf{y}} - \mathbf{y}\|_2^2] = O\left( R_q^N \left( \frac{\sigma_\epsilon^2 d_2}{n} \right)^{1-q/2} + n^{-c_0} \right). \qquad (55)$$

*Proof.* We first prove the case $q = 0$ as a warmup. Observe that

$$\|N\|_F^2 - \sum_{i \le \ell^*} (\sigma_i^N)^2 = \sum_{\ell^* < i \le r} (\sigma_i^N)^2 \le \frac{\theta^2 \sigma_\epsilon^2 d_2 (r-\ell)}{n}.$$

The last inequality uses $\sigma_i^N \le \theta \sigma_\epsilon \sqrt{\frac{d_2}{n}}$ for $i > \ell^*$. Therefore, we have

$$\mathbb{E}[\|\hat{\mathbf{y}} - \mathbf{y}\|^2)2] \le O\left( \frac{(r-\ell^*)\theta^2 \sigma_\epsilon^2 d_2}{n} + \frac{\ell^* d_2 \theta^2 \sigma_\epsilon^2}{n} + n^{-c_0} \right) = O\left( \frac{r\theta^2 \sigma_\epsilon^2 d_2}{n} + n^{-c_0} \right).$$

Next, we prove the general case $q \in (0, 1]$. We can again use an optimization view to give an upper bound of the MSE. We view $\sum_{i \le d_2} (\sigma_i^N)^q \le R_q$ as a constraint. We aim to maximize the uncaptured signals, i.e., solve the following optimization problem

maximize: $\quad \sum_{i>\ell^*}(\sigma_i^N)^2$

subject to: $\quad \sum_{i>\ell}(\sigma_i^N)^q \leq R_-, \quad$ where $R_- = R_q - \sum_{i \leq \ell^*}(\sigma_i^N)^q$

$$\sigma_i^N \leq \frac{\theta^2 \sigma_\epsilon^2 d_2}{n}, \quad \text{for } i \geq \ell^*.$$

The optimal solution is achieved when $(\sigma_i^N)^2 = \frac{\theta^2 \sigma_\epsilon^2 d_2}{n}$ for $\ell^* < i \leq \ell^* + k$, where $k = \frac{R_-}{\left(\frac{\theta^2 \sigma_\epsilon^2 d_2}{n}\right)^{\frac{q}{2}}}$,

and $\sigma_i^N = 0$ for $i > \ell^* + k$. We have

$$\mathbb{E}[\|\hat{\mathbf{y}} - \mathbf{y}\|_2^2]$$

$$\leq \sum_{i>\ell^*}(\sigma_i^N)^2 + O\left(\frac{\ell^* d_2 \theta^2 \sigma_\epsilon^2}{n} + n^{-c_0}\right)$$

$$= \left(R_q - \sum_{i \leq \ell^*}(\sigma_i^N)^q\right)\left(\frac{\theta^2 \sigma_\epsilon^2 d_2}{n}\right)^{1-q/2} + O\left(\frac{\ell^* d_2 \theta^2 \sigma_\epsilon^2}{n} + n^{-c_0}\right) \quad (56)$$

We can also see that

$$\sum_{i \leq \ell^*}(\sigma_i^N)^q \left(\frac{\theta^2 \sigma_\epsilon^2 d_2}{n}\right)^{1-q/2} \geq \sum_{i \leq \ell^*}\left(\frac{\theta^2 \sigma_\epsilon^2 d_2}{n}\right)^{q/2}\left(\frac{\theta^2 \sigma_\epsilon^2 d_2}{n}\right)^{1-q/2} = \ell^*\left(\frac{\theta^2 \sigma_\epsilon^2 d_2}{n}\right).$$

Therefore,

$$(56) \leq O\left(R_q \left(\frac{\theta^2 \sigma_\epsilon^2 d_2}{n}\right)^{1-q/2} + n^{-c_0}\right).$$

$\square$

## 6 Experiments

| Model | $\text{MSE}_{out}$ | $\text{MSE}_{in}$ | $\text{MSE}_{out-in}$ | $R_{out}^2$(bps) | $R_{in}^2$ (bps) | Sharpe | $t$-statistic |
|---|---|---|---|---|---|---|---|
| ARRR, N= 983 | **0.9935** | 1.0140 | **-0.0205** | **46.3761** | 158.2564 | **2.4350** | **8.3268** |
| Lasso | 1.1158 | 0.3953 | 0.7205 | 6.6049 | 7147.0116 | 2.1462 | 0.0601 |
| Ridge | 1.2158 | 0.1667 | 1.0491 | 9.8596 | 8511.9076 | 0.6603 | -0.0497 |
| Reduced ridge | 1.0900 | 0.8687 | 0.2213 | 13.0321 | 1555.5136 | 0.3065 | -0.3275 |
| RRR | 1.2200 | 0.5867 | 0.6332 | 7.0830 | 4121.2548 | 0.3647 | -0.6626 |
| Nuclear norm | 1.2995 | 0.12078 | 1.1787 | 4.7297 | 8789.0625 | 0.6710 | 0.2340 |
| PCR | 1.0259 | 0.8456 | 0.1802 | 1.1278 | 1544.7258 | 1.8070 | 0.3947 |
| ARRR, N= 2838 | **1.0056** | 0.9050 | **0.1006** | **18.5761** | 689.0625 | **1.6239** | **15.4134** |
| Lasso | 1.0625 | 0.5286 | 0.5339 | 1.1236 | 6029.5225 | 0.5954 | 0.0179 |
| Ridge | 1.0289 | 0.6741 | 0.3548 | 0.2116 | 5342.1481 | 0.5739 | 0.0670 |
| Reduced ridge | 1.9722 | 0.7373 | 1.2349 | 1.0816 | 2416.7056 | 1.5482 | 0.0619 |
| RRR | 1.0873 | 0.61376 | 0.4735 | 4.5795 | 3844.124 | -0.477 | 0.6399 |
| Nuclear norm | 1.1086 | 0.15346 | 0.9551 | 2.2097 | 8461.2402 | -0.3698 | -0.8986 |
| PCR | 1.0263 | 0.5336 | 0.4927 | 5.233 | 4653.9684 | 1.2799 | 0.6990 |

**Table 1:** Summary of results for equity return forecasts. $R^2$ are measured by basis points (bps). $1\text{bps} = 10^{-4}$. Bold font denotes the best *out-of-sample* results and smallest gap.

We apply our algorithm on an equity market and a social network dataset to predict equity returns and user popularity respectively. Our baselines include ridge regression ("Ridge"), reduced rank ridge regression [32] ("Reduced ridge"), LASSO ("Lasso"), nuclear norm regularized regression ("Nuclear norm"), reduced rank regression [51] ("RRR"), and principal component regression [1] ("PCR").

**Predicting equity returns.** We use a stock market dataset from an emerging market that consists of approximately 3600 stocks between 2011 and 2018. We focus on predicting the *next 5-day returns*.

| Model | $\text{MSE}_{in}$ | $\text{MSE}_{out}$ | $\text{MSE}_{out-in}$ | $\text{Corr}_{in}$ | $\text{Corr}_{out}$ |
|---|---|---|---|---|---|
| ARRR | $5.0104 \pm 0.38$ | $\mathbf{9.4276} \pm 2.31$ | $\mathbf{4.4172}$ | $0.7425 \pm 0.07$ | $\mathbf{0.6730} \pm 0.13$ |
| Lasso | $2.3755 \pm 1.95$ | $14.8279 \pm 4.81$ | $12.4524$ | $0.9171 \pm 0.09$ | $0.4754 \pm 0.15$ |
| Ridge | $1.3974 \pm 0.53$ | $13.6244 \pm 4.39$ | $12.2270$ | $0.9555 \pm 0.04$ | $0.4742 \pm 0.17$ |
| Reduced ridge | $4.5260 \pm 1.93$ | $12.2339 \pm 2.70$ | $7.7079$ | $0.7905 \pm 0.09$ | $0.4972 \pm 0.18$ |
| RRR | $4.3456 \pm 0.47$ | $13.0768 \pm 2.63$ | $8.7313$ | $0.7725 \pm 0.12$ | $0.3820 \pm 0.22$ |
| Nuclear norm | $4.9190 \pm 2.04$ | $13.0532 \pm 4.38$ | $8.6677$ | $0.7872 \pm 0.10$ | $0.4869 \pm 0.16$ |
| PCR | $6.4037 \pm 1.99$ | $13.0847 \pm 4.19$ | $8.8892$ | $0.7199 \pm 0.05$ | $0.4861 \pm 0.15$ |

**Table 2:** Average results for Twitter dataset from 10 random samples. Bold font denotes the best *out-of-sample* results and smallest gap

For each asset in the universe, we compute its past 1-day, past 5-day, and past 10-day returns as features. We use a standard approach to translate forecasts into positions [3, 53]. We examine two universes in this market: (i) *Universe 1* is equivalent to S&P 500 and consists of 983 stocks, and (ii) *Full universe* consists of all stocks except for illiquid ones.

**Predicting equity returns.** We use a stock market dataset from an emerging market that consists of approximately 3600 stocks between 2011 and 2018. We focus on predicting the *next 5-day returns*. For each asset in the universe, we compute its past 1-day, past 5-day and past 10-day returns as features. We use a standard approach to translate forecasts into positions [3, 53]. We examine two universes in this market: (i) *Universe 1* is equivalent to S&P 500 and consists of 983 stocks, and (ii) *Full universe* consists of all stocks except for illiquid ones.

*Results.* Table 1 reports the forecasting power and portfolio return for *out-of-sample* periods in two universes. We observe that *(i)* The data has a low signal-to-noise ratio. The out-of-sample $R^2$ values of all the methods are close to 0. *(ii)* ADAPTIVE-RRR has the highest forecasting power. *(iii)* ADAPTIVE-RRR has the smallest in-sample and out-of-sample gap (see column $\text{MSE}_{out-in}$), suggesting that our model is better at avoiding spurious signals.

**Predicting user popularity in social networks.** We collected tweet data on political topics from October 2016 to December 2017. Our goal is to predict a user's *next 1-day* popularity, which is defined as the sum of retweets, quotes, and replies received by the user. There are a total of 19 million distinct users, and due to the huge size, we extract the subset of 2000 users with the most interactions for evaluation. For each user in the 2000-user set, we use its past 5 days' popularity as features. We further randomly sample 200 users and make predictions for them, i.e., setting $d_2 = 200$ to make $d_2$ of the same magnitude as $n$.

*Results.* We randomly sample users for 10 times and report the average MSE and correlation (with standard deviations) for both *in-sample* and *out-of-sample* data. In Table 2 we can see results consistent with the equity returns experiment: *(i)* ADAPTIVE-RRR yields the best performance in out-of-sample MSE and correlation. *(ii)* ADAPTIVE-RRR achieves the best generalization error by having a much smaller gap between training and test metrics.

## 6.1 Setup of experiments

### 6.1.1 Equity returns

We use daily historical stock prices and volumes from an emerging market to build our model. Our license agreement prohibits us to redistribute the data so we include only a subset of samples. Full datasets can be purchased by standard vendors such as quandl or algoseek. Our dataset consists of approximately 3,600 stocks between 2011 and 2018.

**Universes.** We examine two different universes in this market *(i)* *Universe 1* is equivalent to S&P 500. It consists of 800 stocks at any moment. Similar to S&P 500, the list of stocks appeared in universe 1 is updated every 6 months. A total number of 983 stocks have appeared in universe 1 at least once. *(ii)* *Universe 2* consists of all stocks except for illiquid ones. This excludes the smallest 5% stocks in capital and the smallest 5% stocks in trading volume. Standard procedure is used to avoid universe look-ahead and survival bias [53].

**Returns.** We use return information to produce both features and responses. Our returns are the "log-transform" of all open-to-open returns [53]. For example, the next 5-day return of stock $i$ is

$\log(p_{i,t+5}/p_{i,t})$, where $p_{i,t}$ is the open price for stock $i$ on trading day $t$. Note that all non-trading days need to be removed from the time series $p_{i,t}$. Similarly, the past 1-day return is $\log(p_{i,t}/p_{i,t-1})$.

**Model.** We focus on predicting the *next 5-day returns* for different universes. Let $\mathbf{r}_t = (r_{1,t}, r_{2,t}, \ldots, r_{d_2,t})$, where $r_{i,t}$ is the next 5-day return of stock $i$ on day $t$. Our regression model is

$$\mathbf{r}_{t+1} = M\mathbf{x}_t + \epsilon. \tag{57}$$

The features consist of the past 1-day, past 5-day, and past 10-day returns of all the stocks in the same universe. For example, in *Universe 1*, the number of responses is $d_2 = 800$. The number of features is $d_1 = 800 \times 3 = 2,400$. We further optimize the hyperparameters $k_1$ and $k_2$ by using a validation set because our theoretical results are asymptotic ones (with unoptimized constants). Baseline models use the same set of features and the same hyper-parameter tuning procedure.

We use three years of data for training, one year for validation, and one year for testing. The model is re-trained every test year. For example, the first training period is May 1, 2011 to May 1, 2014. The corresponding validation period is from June 1, 2014 to June 1, 2015. We use the validation set to determine the hyperparameters and build the model, and then we use the trained model to forecast returns of equity in the same universe from July 1, 2015 to July 1, 2016. Then the model is retrained by using data in the second training period (May 1, 2012 to May 1, 2015). This workflow repeats. To avoid looking-ahead, there is a gap of one month between training and validation periods, and between validation and test periods.

We use standard approach to translate forecasts into positions [23, 37, 3, 53]. Roughly speaking, the position is proportional to the product of forecasts and a function of average dollar volume. We allow short-selling. We do not consider transaction cost and market impact. We use Newey-West estimator to produce $t$-statistics of our forecasts.

### 6.1.2 User popularity

We use Twitter dataset to build models for predicting a user's *next 1-day* popularity, which is defined as the sum of retweets, quotes, and replies received by the user.

**Data collection.** We collected 15 months Twitter data from October 01, 2016 to December 31, 2017, using the Twitter streaming API. We tracked the tweets with topics related to politics with keywords "trump", "clinton", "kaine", "pence", and "election2016". There are a total of 804 million tweets and 19 million distinct users. User $u$ has one interaction if and only if he or she is retweeted/replied/quoted by another user $v$. Due to the huge size, we extract the subset of 2000 users with the most interactions for evaluation.

**Model.** Our goal is to forecast the popularity of a random subset of 200 users. Let $\mathbf{y}_t = (y_{1,t}, \ldots, y_{d_2,t})$, where $d_2 = 200$ and $y_{i,t}$ is the popularity of user $i$ at time $t$. Our regression model is

$$\mathbf{y}_{t+1} = M\mathbf{x}_t + \epsilon. \tag{58}$$

**Features.** For each user, we compute his/her daily popularity for 5 days prior to day $t$. Therefore, the total number of features is $d_1 = 2000 \times 5 = 10,000$.

We remark that there are $n = 240$ observations. This setup follows our assumption $d_1 \gg d_2 \approx n$.

**Training and hyper-parameters.** We use the period from October 01, 2016 to June 30, 2017 as the training dataset, the period from July 01, 2017 to October 30 as validation dataset to optimize the hyper-parameters, and the rest of the period, from September 10, 2017 to December 31, 2017, is used for the performance evaluation.

## 7 Conclusion

This paper examines the low-rank regression problem under the high-dimensional setting. We design the first learning algorithm with provable statistical guarantees under a mild condition on the features' covariance matrix. Our algorithm is simple and computationally more efficient than low rank methods based on optimizing nuclear norms. Our theoretical analysis of the upper bound and lower bound can be of independent interest. Our preliminary experimental results demonstrate the efficacy of our algorithm. The full version explains why our (algorithm) result is unlikely to be known or trivial.

## Broader Impact

The main contribution of this work is theoretical. Productionizing downstream applications stated in the paper may need to take six months or more so there is no immediate societal impact from this project.

## Acknowledgement

We thank anonymous reviewers for helpful comments and suggestions. Varun Kanade is supported in part by the Alan Turing Institute under the EPSRC grant EP/N510129/1. Yanhua Li was supported in part by NSF grants IIS-1942680 (CAREER), CNS-1952085, CMMI-1831140, and DGE-2021871. Qiong Wu and Zhenming Liu are supported by NSF grants NSF-2008557, NSF-1835821, and NSF-1755769. The authors acknowledge William & Mary Research Computing for providing computational resources and technical support that have contributed to the results reported within this paper.