[Reviews · NeurIPS 2020]

Review 1

Summary and Contributions: This paper studies the low-rank regression problem in a high-dimensional setting. It provides an efficient algorithm with small mean squared error under an eigenvalue decay assumption about the covariance matrix of the features.

Strengths: The paper provides a simple two-stage algorithm with provable guarantees under a power-law eigenvalue decay assumption. The algorithm is natural and seems to perform well experimentally. The theoretical claims appear to be sound.

Weaknesses: It is not clear to me how restrictive the assumption on the covariance matrix is.

Correctness: The theoretical claims appear to be correct and the experimental methodology is sound.

Clarity: Overall, the paper is decently written.

Relation to Prior Work: I am not an expert on this part of the literature. The authors argue that their results are distinct from prior literature on the topic.

Reproducibility: Yes

Additional Feedback: Thanks to the authors for their response. I have decided to keep the same score.


Review 2

Summary and Contributions: This paper studies low rank regression problem: given observed responses Y in R^{d_2 x n} and features X in R^{d_1 x n}, the goal is to recover a low rank matrix M such that y = Mx + eps. Unlike previous literature, this paper gives the first result with theoretical guarantee when n << d_1, i.e., the number of observations is much less than the dimension of the features x.

Strengths: This paper introduce a simple and clean algorithm. The algorithm is easy to implement in practice. The analysis for the error bound is not trivial. Authors give a bunch of new views of PCA method. In particular, authors give a clever way to first choose a rank k_1 to reduce the features to a rank k_1 subspace. They show that the samples can give a good estimation to these features. Then authors apply a relatively standard denoise step to find a rank k_2 result.

Weaknesses: My first concern is that the authors assume that a feature x is a multivariate Gaussian. This is usually not the case in practice. I am not sure whether this is a standard assumption in the literature.Though authors claim in line 86 that they can relax the assumption for most the result, it is not clear to me how to extend. My second concern is that since n << d_1, I am curious why the following simpler algorithm could not work:compute a rank-k PCA Y' of Y where k is a tunable parameter. Then solve the Equation Y'=MX to find M. Notice that since n << d_1, the solution of the equation exists. I would appreciate if author gives more intuition why this simple denoise approach for Y does not work.

Correctness: I did not find obvious issues in their proof and the experiments.

Clarity: The paper is well written in general.

Relation to Prior Work: Authors in Section 9 gave a clear discussion to compare with previous work.

Reproducibility: Yes

Additional Feedback: Some minor comments are as follows: - It would be good to mention the observations are n response vectors corresponding to n feature vectors at the beginning of the paper. - Line 88: it would be good if authors justify the assumption d_2 ~= n. Post Rebuttal: The authors addressed my questions.


Review 3

Summary and Contributions: This paper studies the problem of low rank regression where the number of samples is much less than the number of features. The authors give an algorithm achieving improved results in some parameter regimes, and the algorithm is arguably simpler than the previous state of the art (requiring two applications of PCA rather than solving an SDP). They also give lower bounds suggesting that their algorithm is essentially optimal in their setting. Experiments bear out that their algorithm is competitive.

Strengths: The algorithm described here is very simple, and solves a foundational problem nearly optimally. It's great that the authors are able to say something new just by analyzing PCA.

Weaknesses: I would be happy to see more compelling experimental datasets. What about predicting height based on genome, which seems well-suited for this algorithm? It would also be kind to provide one or two simplified corollaries to theorem 1 that state the result without so many parameters (e.g., under a rank assumption).

Correctness: I believe the technical results are correct, but I take issue with some of the expository claims made by the authors. For example, they claim in 9.2(b) (in the full version) that the power law decay assumption is "the weakest possible assumption for non-identity covariance matrices". I don't have any problem with the assumption, but there are clearly non-identity matrices that don't satisfy it. They also state that in 9.1.1 that their analysis works for X which are not near low rank, contrasting with other analyses of PCA. But while their theorem may hold for X which are not close to low rank, it isn't obvious to me that it gives an effective bound, or an interesting statement, in such cases. I am also skeptical of the authors claim of novelty in view PCA as a regularizer. As evidence, they quote Andrew Ng's intro machine learning course where he advises students not to use PCA to prevent overfitting, in cases where doing so throws away a substantial amount of the variance. I don't think it is charitable for the authors to equate simple advice for new ML students with researchers views about possible uses of PCA.

Clarity: The paper could be better written. The exposition is generally clear, and a great deal of intuition is provided. But the authors could give clearer statements of their theorems in particular. For example, in theorem 1, there is an excessive amount of notation, with for example \epsilon and \varepsilon meaning different things, and y apparently defined twice (in fact the second definition of y is a definition of N).

Relation to Prior Work: As mentioned above, the novelty justifications in section 9 of the full version are in my opinion overstated. But the discussion in Section 5 is good.

Reproducibility: Yes

Additional Feedback: I thank the authors for responding to my feedback. Yes, the UK Biobank data would be a great dataset to try your algorithm on, but I think you have to jump through some hoops to get access. I think I remember seeing some smaller human dataset somewhere, or maybe there is a non-human dataset that is easier to access. I urge the authors to purge from their paper any claim to novelty in using PCA for regularization. In particular, rules of thumb given to students in intro ML courses cannot be used to make any claims about the state of the art of ML research. In particular, if you claim that Andrew Ng has a certain opinion, then either present a *research* paper of his that explicitly makes that view, or email him and ask for a direct quote. Better, just get rid of this distraction from your paper! I also would also encourage you to clean up the notation in Thm 1, not just add an extra pointer afterward. \epsilon and \varepsilon shouldn't be used together, there are many greek letters available! If you want to include all those parameters, please put some thought into how to make the theorem as readable as possible. I have slightly lowered the score from 8 to 7, because I'm not sure the authors are serious enough about addressing my expository concerns.


Review 4

Summary and Contributions: This paper suggests a reduced-rank regression (RRR) estimator suitable for the high-dimensional n<<p setting. The estimator is very simple and consists of two steps: (1) reduce X with PCA to Z; (2) do SVD on cross-covariance between Z and Y. The paper claims that this procedure has good statistical guarantees and outpeforms all existing competitors.

Strengths: The paper addresses an important problem of estimating regression coefficients of multivariate regression in n<<p setting. It develops a detailed mathematical treatment (mostly in the Appendix) to provide some statistical guarantees on the performance.

Weaknesses: That said, I am not convinced that this paper provides a contribution of NeurIPS level. The suggested estimator is extremely simple; one could even say "naive". Both ingredients are standard in statistics and machine learning: step (1) is the same as in principal component regression (PCR); step (2) is the same as in partial least squares (PLS). Together, this method is something like a PCR-PLS, with some singular value thresholding. The authors claim that it's very novel and outperforms all the competitors, but I remain not entirely convinced by this (see below). Disclaimer: I did not attempt to follow the mathematical proofs in the Appendix.

Correctness: Likely yes, but I did not attempt to follow the mathematical proofs.

Clarity: Okay-ish.

Relation to Prior Work: Not clearly enough.

Reproducibility: Yes

Additional Feedback: This paper suggests a reduced-rank regression (RRR) estimator suitable for the high-dimensional n<<p setting. The estimator is very simple and consists of two steps: (1) reduce X with PCA to Z; (2) do SVD on cross-covariance between Z and Y. The paper claims that this procedure has good statistical guarantees and outpeforms all existing competitors. The paper addresses an important problem of estimating regression coefficients of multivariate regression in n<<p setting. It develops a detailed mathematical treatment (mostly in the Appendix) to provide some statistical guarantees on the performance. That said, I am not convinced that this paper provides a contribution of NeurIPS level. The suggested estimator is extremely simple; one could even say "naive". Both ingredients are standard in statistics and machine learning: step (1) is the same as in principal component regression (PCR); step (2) is the same as in partial least squares (PLS). Together, this method is something like a PCR-PLS, with some singular value thresholding. The authors claim that it's very novel and outperforms all the competitors, but I remain not entirely convinced by this (see below). Disclaimer: I did not attempt to follow the mathematical proofs in the Appendix. Major issues * The algorithm consists of applying PCA to X as in PCR and then doing SVD of cross-covariance between Y and PCs of X as in PLS. The entire method could be called PCR-PLS. Of course PLS is related to RRR but they are not equivalent. For example, the proposed algorithm does not seem to converge to the standard RRR for infinite data, or when the Step-1-PCA is modified to keep all PC components of X. Doing SVD of ZY where Z are standardized PCs of X is not equivalent to RRR. This is confusing. * I suspect that the authors already had to deal with similar criticim (that their method is not very novel) because the Appendix contains a dedicated section about why this approach is novel. One part of claims that PCA is traditionally *not* seen as a regularization method. It cites Andrew Ng saying that PCA is used to speed up the computations but cannot prevent overfitting. Whatever the views of Andrew Ng might be (and with all due respect), it is standard textbook knowlegde that PCA in principal component regression (PCR) can very effectively prevent overfitting and in fact is closely related to ridge regression. See Hastie et al., The Elements of Statistical Learning. It is strange to claim that PCA regularization is a novel idea. * The experimental results in Table 1 are not convincing -- perhaps because they are presented in an unclear way: (a) What exactly is R^2_out? If it actually is R^2 then it cannot be above 1. Unclear what R^2=18 means. (b) Many comparison methods have hyper-parameters (regularization strength, i.e. penalty in ridge regression, number of PCs in PCR, number of components in RRR, etc.). How were they set? It only makes sense to use cross-validation to set the optimal parameters. Was CV used here? It is not mentioned. (c) I cannot believe that the method suggested here outperforms PCR, RR, RRR, etc. by more than 3-fold (as in Table 1). This is an enormous difference that makes one suspect that other methods were not applied correctly. Unfortunately, the manuscript does not convince the reader that the comparisons are sound. I want to stress that this comparison table is absolutely crucial for the paper. Medium issues * The motivation in section 2 is slightly confusing because "our model" in line 84-88 does not mention reduced-rank. It seems that the reduced-rank regression here is not the goal per se but rather a regularization method to estimate multivariate regression coefficeints (matrix M). Maybe this can be pointed out more explicitly. * line 119: "PCA reduction outputs P_k(X)" where P_k is a rank-k approximation to X -- why?? PCA reduction can mean different things, including extracting k left singular vectors of X. Minor issues * line 32: what is "sample complexity n"? --------------- POST-REBUTTAL: I have increased my score from 3 to 4. The authors addressed some of the issues I raised above, but I am still not really convinced by the experiments and do not understand how PCA+RRR can outperform e.g. ridge RRR by such a large margin. Regarding "When all PCs of X is kept, the problem reduces to RRR" -- I wish the authors point this out and ideally prove in the revised paper. Regarding the discussion around Ng's quote: I do not understand what the authors mean by "but the analysis is possible only under factor models". Not sure what "under factor models" means here. I think "View 1" is simply that PCA can serve as a regularization. This is what PCR is all about. I teach this in the first Intro course to Machine Learning. And I do not mention any "factor models". I think the authors are actually doing themselves a disservice by arguing this point.

[Author Response · NeurIPS 2020]

**To Reviewer 1:** Thanks for the review. **To Reviewer 2: (1) Relaxing Gaussianity**: Our analysis used Gaussian con-
ditions in two places. (i) Estimation of the covariance matrix (Prop 1) requires only matrix concentration inequalities.
We can replace the Gaussian assumption by moments and/or boundedness assumptions. Sometimes a log factor needs
to be paid. (ii) Step 2&3 (in appendix) requires the sum of a sufficiently large subset of coordinates of $z$ to concentrate
around the expectation. We can replace Gaussians by sub-Gaussian random variables. Further relaxation is possible by
deriving *ab initio* concentration bounds. **(2) Suggested algorithm**: The proposed formulation can be exactly solved
(zero MSE) with multiple solutions of $M$. The solution under many circumstances (e.g., $M$ has low rank) can have
the noise term removed but may still be unsuitable for the application context. If the reviewer can elaborate on the
idea, we can properly discuss potential barriers of implementing the idea.

**To Reviewer 3: (1) Corollaries to Thm 1 (many parameters).** Sec. 5 discussed simplified corollaries in narrative,
including the rank assumption suggested (L315&317). We will give pointers to the Cor's after Thm 1. **(2) Power**
**Law.** Thank you for pointing this out. We will fix it. **(3) Effective bound (related to $X$).** Lower bounds for solving
the regression problem is in Sec. 4; A lower bound for the sub-procedure of covariance matrix estimation is indeed
open. We suspect proving this lower bound requires some heavy anti-concentration inequalities for matrices because
we need to show the gap paid by the Davis-Kahan theorem is inevitable. **(4) New data (height based on genome).**
Thank you for the suggestion. Is UKBB (used in Lello et. al 18) the dataset in your mind? We will look into it (our
method appears to be applicable). **(5) A. Ng's quote.** See the end of the page; we will contextualize this better.

**To Reviewer 4:** We comment on the reproducibility issue. **(1) Baseline implementation details (cross validation):**
Cross validation was used for baselines. More specifically, in p. 36 of App, "We use three years of data for train-
ing, one year for validation, and one year for testing. The model is re-trained every test year..." See Fig. 1 for the
sample code for parameter sweeping for RRR. **(2) Metrics &amp; $R^2$.** $R^2$ is measured in basis points ($10^{-4}$). See Ta-
ble 1's caption in the main text. The test $R^2$ (18bps) is within plausible range (E. Chan 17, Zhou & Jain 14). We
believe using multiplicative metrics (3-fold improvement) is inappropriate because the baselines' $R^2$ are too close
to 0 in equity datasets (suggesting they overfit). **(3) Intuition of performance gap.** The performance gap comes
from the overfitting of baselines (recall that we proved near-optimality of our algorithm's prediction power). Take
for example PCR, which can be considered a special case of ARRR with $k_1 = k_2$. For the equity dataset, the
$k_2$ found through cross validation is around 1/3 of $k_1$. This means PCR has effectively 3x more parameters than
ARRR. When the models fit against approximately 750 training points (750 trading days in training), PCR experi-
ences a more pronounced overfitting problem. Note that the severity of overfitting depends on the signal-to-noise
ratio (as predicted by Thm 1). Therefore, because the Twitter dataset has higher signal-to-noise ratio, the perfor-
mance gap between PCR and ARRR is less pronounced. Other baselines can be analyzed in a similar manner.
**(4) See Table 1 for a point-to-point response.** Blue text is non-opt statements (we do not disagree); red text is
   problematic ones (factual error or major misunderstanding); green text is clarified (mostly already in appendix).

| Review | Response |
|---|---|
| The algorithm consists of applying PCA to X as in PCR and then doing SVD of cross-covariance between Y and PCs of X as in PLS. The entire method could be called PCR-PLS. Of course PLS is related to RRR but they are not equivalent. For example, the proposed algorithm does not seem to converge to the standard RRR for infinite data. | In high-dim, we consider $n$ and $d_1$ grow together and $n < d_1$ (L88). $n$ grows with $d_1$ and $d_2$ fixed is a **non-scope. L109: "We examine the setting where existing algorithms fail to deliver non-trivial MSE'.** |
| , or when the Step-1-PCA is modified to keep all PC components of X. Doing SVD of ZY where Z is standardized PCs of X is not equivalent to RRR. This is confusing. | This statement is incorrect. When all PCs of X is kept, the problem reduces to RRR. See Reinsel & Velu 98. |
| I suspect that the authors already had to deal with similar criticim (that their method is not very novel) because the Appendix contains a dedicated section about why this approach is novel. One part of claims that PCA is traditionally *not* seen as a regularization method....it is standard textbook knowledge that PCA in principal component regression (PCR) can very effectively prevent overfitting and in fact is closely related to ridge regression. See Hastie et al., The Elements of Statistical Learning. It is strange to claim that PCA regularization is a novel idea. | See below for clarification. We remark that **Hastie et al's view** (Sec. 18.6 PCR in high dim) belongs to "view 1" (analysis is possible only under factor models). **References cited therein (e.g., Bair et al 06) in fact re-confirm the accuracy of our discussion.** |
| What exactly is $R^2_{out}$? If it actually is $R^2$ then it cannot be above 1. Unclear what $R^2$=18 means. | Explained above (see also Table 1's caption in the main text.) |
| Many comparison methods have hyper-parameters … How were they set? It only makes sense to use cross-validation to set the optimal parameters. Was CV used here? It is not mentioned. I cannot believe that the method suggested here outperforms PCR, RR, RRR, etc. by more than 3-fold (as in Table 1). This is an enormous difference that makes one suspect that other methods were not applied correctly. | See implementation details and intuition of the performance gap. |
| I am not convinced that this paper provides a contribution of NeurIPS level. The suggested estimator is extremely simple; one could even say "naive". | That an estimator is naive does not mean the analysis is trivial. Proving that simple estimators are theoretically sound and work in practice is an important research direction in CS and Statistics. |

Table 1: Specific questions and responses.

**A. Ng's view.** One view (view 2; Ng) claims PCA should
not be used for preventing overfitting whereas the other
view (view 1) claims PCA can tackle overfitting prob-
lems but the analysis is possible only under factor mod-
els. We will change the wording on view 1 to avoid
confusion. Our view is new: it can tackle overfitting
problems (different from v2) and can do so even without
factor models (strengthened v1) but this needs to be ana-
lyzed in a specific high-dim setting (not mentioned in v1
or v2).

```
function  rrr_rank = find_optima_rrr(Xtrain, ytrain, Xtest, ytest)
    rgbeta = 1:1:50;
    errorMatrix = zeros(1,length(rgbeta));
    for j = 1:length(rgbeta)
            keep_beta = rgbeta(j);
            [~, predY_test, ~] = f_rrr(Xtrain, ytrain, Xtest, keep_beta);
            err = norm(ytest(:)- predY_test(:));
            errorMatrix(j) = err;
    end
    min_ie = find(errorMatrix == min(errorMatrix(:)));
    rrr_rank = rgbeta(min_ie(1));
end
```

Figure 1: Sample code for parameter sweeping for RRR.

[Meta-Review · NeurIPS 2020]

The paper studies the low rank regression problem with a very small number of samples and a large number of features. For certain settings the paper gives improved results, and the algorithm is also arguably much simpler than existing algorithms. The authors also prove lower bounds showing optimality of their algorithms, and convincing experimental results. The algorithm is in fact so simple and for an important problem, that the reviewers were worried they might be missing some existing work, though we did not find any. Overall it's nice that the authors are able to say something fundamental about PCA.